# TRAINING MULTI-LAYER TRANSFORMERS IN ALMOST LINEAR TIME

## ABSTRACT

The computational complexity of the self-attention mechanism in popular transformer architectures poses significant challenges for training and inference, and becomes the bottleneck for long inputs. Is it possible to significantly reduce the quadratic time complexity of computing the gradients in multi-layer transformer models? This paper proves that a novel fast approximation method can calculate the gradients in almost linear time $n^{1+o(1)}$ where $n$ is the input sequence length, while it maintains a polynomially small approximation error $1/\mathrm{poly}(n)$ across the entire model. Our theory holds for general loss functions and when the multi-layer transformer model contains many practical sub-modules, such as residual connection, causal mask, and multi-head attention. We further validate our approach through numerical experiments, demonstrating both its high approximation fidelity and substantial speedups in practice. By improving the efficiency of gradient computation, we hope that this work will facilitate more effective training and deployment of long-context language models based on our theoretical results.

## 1 INTRODUCTION

Large Language Models (LLMs), such as ChatGPT (Schulman et al., 2022), GPT-4 (Achiam et al., 2023), Claude 3.5 (Anthropic, 2024), Llama 3.1 (Llama Team, 2024), DeepSeek R1 (Guo et al., 2025) and others, have demonstrated immense potential to enhance various aspects of our daily lives, e.g., conversation AI (Liu et al., 2024), AI agent (Xi et al., 2023; Chen et al., 2024c), search AI (OpenAI, 2024), AI assistant (Mahmood et al., 2023; Zhang et al., 2023) and many so on. One of LLMs' most emergent abilities is working with long-context information, a format crucial for recording material such as academic papers, official reports, legal documents, and so on. LLMs have proven adept at tackling long-context tasks, including Retrieval Augmented Generation (RAG) (Lewis et al., 2020; Gao et al., 2023d), zero-shot summarization (Liu et al., 2023; Zhang et al., 2024c), and maintaining very long-term conversations (Xu et al., 2021b; 2022), and so on. This proficiency has necessitated the development of long-context modeling capabilities within LLMs.

The self-attention mechanism is crucial for the success of LLMs since LLMs are mainly based on Transformer architecture, whose key module is attention. In attention computation, we will compute the attention score between each pair of tokens, which is the complexity bottleneck during long context training and inference. In detail, we need to spend $O(n^2d)$ running time for each self-attention block, which is quadratic in $n$, where $n$ is the length of the context input and $d$ is the hidden feature dimension of the model. For example, LLaMA 3.1 405B (Llama Team, 2024), one of the cutting-edge LLMs, supports $n =$128k and $d = 4096$, while taking 30.84M GPU training hours, which underscores the need for more efficient training processes for such extensive context models. Given the extensive context lengths of LLMs, this quadratic time complexity results in critical challenges: ($i$) a marked decrease in training efficiency (He et al., 2023; Lv et al., 2023); and ($ii$) significant energy usage, which in turn contributes to higher carbon dioxide emissions (Samsi et al., 2023; Stojkovic et al., 2024).

One seminal work (Alman & Song, 2023) showed that the self-attention inference can be approximated in almost linear time. However, this result is for the *inference* time (forward pass), but does not address the main challenge, which is the expensive computation in the *training* time (backward pass). In this work, we address this main challenge by proving that the gradient computation in the

back-propagation of self-attention can be approximated in almost linear time. This suggests we may be able to save the substantial resources required for training LLMs.

## 1.1 KEY BACKGROUND

We first introduce some basic background, starting with defining the softmax function and the self-attention module.

**Definition 1.1** (Softmax). *Let $z \in \mathbb{R}^n$. We define* Softmax $: \mathbb{R}^n \to \mathbb{R}^n$ *satisfying*

$$\mathsf{Softmax}(z) := \exp(z)/\langle \exp(z), \mathbf{1}_n \rangle.$$

*Here, we apply* exp *to a vector entry-wise.*

**Definition 1.2** (Self-attention module). *Let $X \in \mathbb{R}^{n \times d}$ denote the input sequence, where $n$ is the number of input tokens and $d$ is the hidden dimension size. Let $W_Q, W_K, W_V \in \mathbb{R}^{d \times d}$ be the query, key and value weight matrix. The self-attention function* Attn$(X)$ *with weights is:*

$$\mathsf{Attn}(X) = \mathsf{Softmax}(XW_QW_K^\top X^\top/d) \cdot XW_V.$$

*where* Softmax *is applied to each row of its input matrix. The attention can be re-written as:*

$$\mathsf{Attn}(X) = f(X) \cdot XW_V,$$

*where (1) $A := \exp(XW_QW_K^\top X^\top/d) \in \mathbb{R}^{n \times n}$ and* exp *is applied element-wise, (2) $D := \mathrm{diag}(A\mathbf{1}_n) \in \mathbb{R}^{n \times n}$, and (3) $f(X) := D^{-1}A \in \mathbb{R}^{n \times n}$ is the attention matrix.*

In contemporary LLMs, the architecture typically incorporates multiple layers of attention. Consequently, in order to design a fast training algorithm for the entire model, it is imperative to examine self-attention within the multi-layer transformer structure formally defined as follows.

**Definition 1.3** (Multi-layer transformer). *Let $m$ denote the number of transformer layers in the model. Let $X$ be the input sequence. Let $g_i$ denote components other than self-attention in the $i$-th transformer layer, and assume its forward and backward computations can be run in time linear in its input sequence length. Let* Attn$_i$ *denote the self-attention module in the $i$-th transformer layer with weights $W_{Q_i}, W_{K_i}, W_{V_i}$ (see also Definition 1.2). We define an $m$-layer transformer as*

$$\mathsf{F}_m(X) := g_m \circ \mathsf{Attn}_m \circ g_{m-1} \circ \mathsf{Attn}_{m-1} \circ \cdots \circ g_1 \circ \mathsf{Attn}_1 \circ g_0(X),$$

*where $\circ$ denotes function composition.*

In Definition 1.3, the $g_i$ includes the layer norm, MLP, residual connection, dropout, positional encoding, multi-head concatenation, and other operations. All forward and backward computations of these practical modules can be run in linear time with respect to $n$. Thus, in this work, we mainly focus on the acceleration of the self-attention module. Specifically, as shown in Definition 1.2, the $n \times n$ attention matrix $f(X)$ dominates the computational complexity, introducing a quadratic bottleneck. In the exact computation case, if the attention matrix is full rank, no acceleration is possible. However, by compromising negligible accuracy, designing a fast sub-quadratic algorithm becomes feasible. Fortunately, by employing the polynomial kernel approximation method from (Aggarwal & Alman, 2022), we can approximate the attention matrix and achieve an almost linear time $n^{1+o(1)}$ algorithm, effectively breaking the quadratic bottleneck.

## 1.2 OUR CONTRIBUTIONS

We now state our main result as follows:

**Theorem 1.4** (Main result, informal version of Theorem 3.2). *Let $n$ be the number of tokens and $d$ the hidden dimension size. We assume $d = O(\log n)$ and each number in matrices can be written using $O(\log n)$ bits. Assume the number of layers $m$ is constant. There exists an algorithm (Algorithm 1) that can compute the gradient of multi-layer self-attention (see also Definition 1.3) in almost linear time $n^{1+o(1)}$, where the approximation error of the algorithm that computes the gradient of the entire model can be bounded by $1/\mathrm{poly}(n)$.*

Our assumption is mild when the context length $n$ is large, as the feature dimension $d$ is usually regarded as a constant, which is also used in (Aggarwal & Alman, 2022); similarly, the number of

layers is usually much smaller than $n$ and regarded as a constant. Our results indicate that large language models (LLMs) can be trained in almost linear time $n^{1+o(1)}$ and maintain a robust approximation guarantee, while the traditional way takes $\Omega(n^2)$ time. This advancement is realized through the application of polynomial kernel approximation (Alman & Song, 2023; 2024a). To be more specific, by leveraging the inherent sparsity within the dense attention matrix, we perform efficient low-rank approximation, thereby significantly accelerating the computation of the dense matrices. Our framework is applicable to general loss functions, making it universally applicable. Furthermore, our analysis holds when the multi-layer transformer model contains many practical sub-modules, such as residual connection, causal mask, and multi-head attention (Section 5).

Numerous studies, including FlashAttention (Dao et al., 2022; Dao, 2023; Shah et al., 2024), quantization techniques (Hu et al., 2024a; Lin et al., 2024), and sparsity approaches (Han et al., 2024; Ma et al., 2024a), have empirically focused on accelerating attention mechanisms. However, theoretically, these methods are still constrained by quadratic time complexity. In this study, we introduce an innovative acceleration technique (Algorithm 1) that effectively overcomes this quadratic bottleneck, backed by solid theoretical foundations (Theorem 3.2). Moreover, this new method is designed to be seamlessly integrated with existing approaches to further enhance their performance (see Section 5).

Our contributions are as follows:

- We introduce a fast computation method that allows the gradient of each self-attention layer to be approximated in almost linear time $n^{1+o(1)}$ with $1/\operatorname{poly}(n)$ error, where $n$ is the input sequence length, breaking the quadratic time complexity bottleneck (Theorem 3.1).
- We extend our single-layer results to module-wise gradient computation so that our Algorithm 1 approximates gradient computation in $m \cdot n^{1+o(1)}$ time for $m$-layer transformer. Importantly, the approximation of the gradient diverges from the exact gradient by an error of $1/\operatorname{poly}(n)$ across the entire model (Theorem 3.2).
- Additionally, our analysis holds for the multi-layer transformer model contains residual connection, casual mask, and multi-head attention. Our results can be applied to any gradient-based algorithm, e.g., training, full fine-tuning, prompt-tuning, and so on (Section 5).

**Roadmap.** Our paper is organized as follows. Section 2 provides essential concepts and key definitions across the whole paper. Section 3 presents our primary findings, where we articulate our novel algorithm that is capable of calculating gradients across the entire model in almost linear time. In Section 4, we explain the techniques we employ, including low-rank approximation, techniques for accelerating the computation of gradients, and an analysis of the approximation error. Section 5 provides various extensions of our algorithm. Lastly, we conclude this paper in Section 6.

## 2 PRELIMINARY

### 2.1 LOSS FUNCTION

The loss function is the optimization objective in the training of LLMs, and we define it as follows.

**Definition 2.1** (Loss function $L(X)$). *For some input matrix $X \in \mathbb{R}^{n \times d}$, we define the one-unit differentiable loss function $\ell(X)_{j,k} : \mathbb{R}^{n \times d} \to \mathbb{R}$, for any $j \in [n], k \in [d]$, and assume differentiability. Furthermore, we define the overall loss function $L(X)$, such that $L(X) = \sum_{j=1}^{n} \sum_{k=1}^{d} \ell(X)_{j,k}$.*

**Remark 2.2.** *Typically, the most widely used loss function in the LLM training procedure is the cross-entropy loss function, which can also be viewed as a summation of one unit loss function as in Definition 2.1. The output matrix of the multi-layer transformer needs to pass an additional linear layer to map the hidden dimension $d$ to the vocabulary size $d_{\text{voc}}$. Assuming $d_{\text{voc}}$ is a constant, the weight matrix dimensions for this additional MLP layer are $d \times d_{\text{voc}}$. The probability tensor $Y_{\text{pred}} \in \mathbb{R}^{n \times d_{\text{voc}}}$ is the final output. We denote the ground truth as $Y_{\text{gt}} \in \mathbb{R}^{n \times d_{\text{voc}}}$ corresponding to $Y_{\text{pred}}$. According to the cross-entropy loss definition, the formula is expressed as*

$$L_{\text{cross}-\text{entropy}}(X) = -\sum_{j=1}^{n} \sum_{k=1}^{d_{\text{voc}}} (Y_{\text{gt}})_{j,k} \log((Y_{\text{pred}})_{j,k}).$$

*where the summation iterates over all elements. The ground truth $(Y_{\text{gt}})_{j,k} = 1$ for the correct class and $0$ otherwise.*

## 2.2 Closed Forms of Gradient Components

In training large language models (LLMs), updating the model necessitates computing the gradient of weights for every layer. Consequently, it becomes essential to derive the closed-form expressions for all corresponding gradient components with respect to the weights of the query, key, and value matrices in the transformer model. We first define some intermediate variables before detailing these gradient components in each self-attention transformer layer.

**Definition 2.3** (Intermediate variables $T_i$). *Let $m$ denote the number of transformer layers in the model. Let $m$-layer self-attention transformer as defined in Definition 1.3. Let $d$ denote the hidden dimension. Let $n$ denote the sequence length. Let $X \in \mathbb{R}^{n \times d}$ be the input sentence. Let $g_i$ denote components other than self-attention in the $i$-th transformer layer. Let $\mathsf{Attn}_i$ denote the self-attention module in the $i$-th transformer layer (see also Definition 1.2).*

*For $i \in \{0, 1, 2, \cdots, m\}$, we define $T_i(X) \in \mathbb{R}^{n \times d}$ be the intermediate variable (hidden states) output by $i$-th layer self-attention transformer. Namely, we have*

$$T_i(X) = \begin{cases} g_0(X), & i = 0; \\ (g_i \circ \mathsf{Attn}_i)(T_{i-1}(X)), & i \in [m]. \end{cases}$$

*Here, we use $\circ$ to denote function composition.*

Then, we are ready to introduce the closed forms of the three gradient components in a single self-attention transformer layer. Notably, according to the chain rule, the gradient of the $k$-th transformer layer in LLMs depends on the gradient components from the $(k + 1)$-th transformer layer. The gradient can be calculated for every transformer layer by combining the upstream and local gradients. The closed forms of the gradients for each layer in multi-layer transformers are formalized in the following lemma (Lemma 2.4).

**Lemma 2.4** (Closed form of gradient components, informal version of Lemma C.4). *Let $L(X)$ as defined in Definition 2.1, and the $m$-layer transformer defined as in Definition 1.3. Let $W_{Q_i}, W_{K_i}, W_{V_i} \in \mathbb{R}^{d \times d}$ denote the attention weight in the $i$-th attention. Let $T_i(X)$ denote the intermediate variable output by $i$-th self-attention transformer layer (see Definition 2.3). Let $G_i \in \mathbb{R}^{n \times d}$ denote the gradient matrix resulting from the application of the chain rule up to the function $g_i$, i.e., $G_i = \frac{\mathrm{d}L(X)}{\mathrm{d}\mathsf{Attn}_i(T_{i-1}(X))}$. For $j \in [n], k \in [d]$, let $G_i(j, k)$ denote the $(j, k)$-th entry of $G_i$, let $\frac{\mathrm{d}\mathsf{Attn}_i(T_{i-1}(X))_{j,k}}{\mathrm{d}T_{i-1}(X)} \in \mathbb{R}^{n \times d}$ denote the gradient of $(j, k)$-th entry of $\mathsf{Attn}_i(T_{i-1}(X))$. Then, we can show that*

- **Part 1.** $\frac{\mathrm{d}L(X)}{\mathrm{d}T_{i-1}(X)} = \sum_{j=1}^{n} \sum_{k=1}^{d} G_i(j, k) \cdot \frac{\mathrm{d}\mathsf{Attn}_i(T_{i-1}(X))_{j,k}}{\mathrm{d}T_{i-1}(X)}.$

- **Part 2.** *Let $W_{*_i}$ be $W_{Q_i}, W_{K_i}$ or $W_{V_i}$, then*

$$\frac{\mathrm{d}L(X)}{\mathrm{d}W_{*_i}} = \sum_{j=1}^{n} \sum_{k=1}^{d} G_i(j, k) \cdot \frac{\mathrm{d}\mathsf{Attn}_i(T_{i-1}(X))_{j,k}}{\mathrm{d}W_{*_i}}.$$

Our main results are based on the above closed forms of four gradient components.

## 3 Main Results

In this section, we present our main findings. In Section 3.1, we delineate the computational efficiency of our gradient calculation methods in each single layer. Section 3.2 introduces our main theorem (Theorem 3.2) for multi-layer transformer by integrating the preceding results and providing our main algorithm (Algorithm 1). Section 3.3 discusses how we transcend the previous works.

## 3.1 Fast Computing for Single Layer

In the case of single-layer attention, we provide our theorem that states the three gradient components can be calculated in almost linear time with negligible error.

**Theorem 3.1** (Single-layer gradient approximation). *We assume $d = O(\log n)$ and each number in matrices can be written using $O(\log n)$ bits. Let $L(X)$ be defined as Definition 2.1. Suppose we have a single-layer self-attention transformer model ($m = 1$ in Definition 1.3). Assume $\|X\|_\infty, \|W_Q W_K^\top\|_\infty, \|W_V\|_\infty \le \mathrm{poly}(n)$. We can approximate one-layer self-attention for three gradient components, i.e. $\frac{dL(X)}{dX}$, $\frac{dL(X)}{dW_Q W_K^\top}$ and $\frac{dL(X)}{dW_V}$, in $n^{1+o(1)}$ time with $1/\mathrm{poly}(n)$ error.*

*Proof.* We finish the proof by combining Lemma 4.1, 4.2 and 4.3. $\qquad\qquad\square$

Next, we present the formal algorithm for our method, detailed in Algorithm 1. Our algorithm comprises two primary functions: SINGLEGRAD, which computes the gradient for a single transformer layer (Line 12), and MULTIGRAD, which calculates the gradient across an $m$-layer transformer (Line 26). SINGLEGRAD function computes each gradient component using the techniques described in the Appendix and subsequently integrates these approximated components into the gradients for $T_i$, $W_{Q_i} W_{K_i}^\top$, and $W_{V_i}$. MULTIGRAD function iterates through each layer, leveraging the gradient for $T_i$ from preceding layer to compute the gradients in the current layer.

---

**Algorithm 1** Almost Linear Time (ALT) Multi-layer Transformer Gradient Approximation

---

1: **datastructure** ALTGRAD $\qquad\qquad\qquad\qquad\qquad\qquad\qquad\qquad$ ▷ Theorem 3.1 and 3.2
2: **members**
3: $\qquad n \in \mathbb{R}$: the length of input sequence
4: $\qquad d \in \mathbb{R}$: the hidden dimension
5: $\qquad m \in \mathbb{R}$: the number of transformer layers
6: $\qquad L(X) \in \mathbb{R}$: the loss function $\qquad\qquad\qquad\qquad\qquad\qquad\qquad\qquad$ ▷ Definition 2.1
7: $\qquad T_i \in \mathbb{R}^{n \times d}$: the output of $i$-th transformer layer
8: $\qquad \mathsf{Attn}_i \in \mathbb{R}^{n \times d}$: the output that pass $i$-th attention layer
9: $\qquad W_{Q_i}, W_{K_i}, W_{V_i} \in \mathbb{R}^{d \times d}$ : the weight matrices in $i$-th transformer layer
10: **end members**
11:
12: **procedure** SINGLEGRAD($\frac{dL(X)}{dT_i}$) $\qquad\qquad\qquad\qquad\qquad\qquad\qquad$ ▷ Theorem 3.1
13: $\qquad$ Compute $G_i = \frac{dL(X)}{d\mathsf{Attn}_i}$ via Lemma 4.4 $\qquad\qquad\qquad\qquad$ ▷ $n^{1+o(1)}$ time
14: $\qquad$ Compute $\widetilde{D}_6, \widetilde{D}_7, \widetilde{D}_8, \widetilde{D}_2, \widetilde{D}_4$ via Lemma E.5, E.6, E.8, E.10 $\qquad$ ▷ $n^{1+o(1)}$ time
15: $\qquad$ /* Approximate $\frac{dL(X)}{dT_{i-1}}$, Lemma 4.1 */
16: $\qquad \widetilde{g}_t \leftarrow \widetilde{D}_6 + \widetilde{D}_7 + \widetilde{D}_8 + \widetilde{D}_2 + \widetilde{D}_4 \qquad\qquad\qquad\qquad\qquad$ ▷ $n^{1+o(1)}$ time
17: $\qquad$ /* Approximate $\frac{dL(X)}{dW_{Q_i} W_{K_i}^\top}$, Lemma 4.2 */
18: $\qquad$ Construct $U_3, V_3$ via Lemma 4.2 $\qquad\qquad\qquad\qquad\qquad\qquad$ ▷ $n^{1+o(1)}$ time
19: $\qquad \widetilde{g}_w \leftarrow (T_{i-1}^\top U_3) \cdot (V_3^\top T_{i-1}) \qquad\qquad\qquad\qquad\qquad\qquad$ ▷ $n^{1+o(1)}$ time
20: $\qquad$ /* Approximate $\frac{dL(X)}{dW_{V_i}}$, Lemma 4.3 */
21: $\qquad$ Construct $U_1, V_1$ via Lemma C.13 $\qquad\qquad\qquad\qquad\qquad\qquad$ ▷ $n^{1+o(1)}$ time
22: $\qquad \widetilde{g}_v \leftarrow (T_{i-1}^\top U_1) \cdot (V_1^\top G_i) \qquad\qquad\qquad\qquad\qquad\qquad$ ▷ $n^{1+o(1)}$ time
23: $\qquad$ **return** $\widetilde{g}_t, \widetilde{g}_w, \widetilde{g}_v \qquad$ ▷ $\widetilde{g}_t$ is the approximated $\frac{dL(X)}{dT_{i-1}}$ for back-propagation
24: **end procedure**
25:
26: **procedure** MULTIGRAD($L(X)$) $\qquad\qquad\qquad\qquad\qquad\qquad\qquad$ ▷ Theorem 3.2
27: $\qquad$ Compute $\frac{dL(X)}{dT_m}$ $\qquad\qquad\qquad\qquad\qquad\qquad\qquad\qquad\qquad$ ▷ $O(nd)$ time
28: $\qquad \widetilde{g}_t \leftarrow \frac{dL(X)}{dT_m}$
29: $\qquad$ **for** $i = m \to 1$ **do**
30: $\qquad\qquad \widetilde{g}_t, \widetilde{g}_w, \widetilde{g}_v \leftarrow$ SINGLEGRAD $(\widetilde{g}_t)$
31: $\qquad\qquad$ Optimize $W_{Q_i}, W_{K_i}$ via $\widetilde{g}_w$ using optimizer
32: $\qquad\qquad$ Optimize $W_{V_i}$ via $\widetilde{g}_v$ using optimizer
33: $\qquad$ **end for**
34: **end procedure**
35: **end datastructure**

---

## 3.2 Fast Computing for Multi-Layer Transformers

Based on the results demonstrated in previous sections, we are ready to introduce our main result: the gradients of the whole transformer model can be approximated in almost linear time.

**Theorem 3.2** (Main result, formal version of Theorem 1.4). *Let $m$ denote the number of transformer layers. Assume the number of layers $m$ is constant. We assume $d = O(\log n)$ and each number in matrices can be written using $O(\log n)$ bits. We can show that, for any $i \in [m]$, all the gradient components (see also Lemma 2.4) of the $i$-th layer can be computed by Algorithm 1 in almost linear time $n^{1+o(1)}$, and approximation error of the algorithm that computes the gradient of the entire $m$ layer transformer model can be bounded by $1/\operatorname{poly}(n)$.*

*Proof.* We prove the theorem by directly combining Theorem 3.1 and Lemma 4.5. $\qquad\square$

Theorem 3.2 demonstrates that, during the training of a multi-layer transformer model, at each training iteration, the gradient computation for the weight matrices of each layer can be performed in almost linear time $n^{1+o(1)}$. This result supports the feasibility of fast training for any transformer-based large language models (LLMs). Algorithm 1 highlights the significance of the gradient with respect to the intermediate variables $T_i(X)$. Due to the application of the chain rule in gradient computation, the gradient of $T_i(X)$ is indispensable for determining the gradients of the weight matrices $W_{Q_i}, W_{K_i}$ and $W_{V_i}$ at the $i$-th layer. Consequently, by iteratively computing the gradient for $T_i(X)$, we systematically propagate the gradient through to the initial transformer layer.

## 3.3 Beyond the Previous Work

Our algorithm exhibits significant advancements over two seminal prior studies, (Alman & Song, 2023) and (Alman & Song, 2024a). In (Alman & Song, 2023), the authors proposed an almost linear time algorithm for computing the forward process of the attention mechanism. In contrast, (Alman & Song, 2024a) introduced an almost linear time algorithm for the backward of attention mechanism. However, (Alman & Song, 2024a) has the following limitations: ($i$) only computing gradients for a single layer of the attention mechanism, which cannot extend to multiple layers; ($ii$) computing gradients only for the weight matrix $W_{Q_i}, W_{K_i}$ (as defined in Definition 1.2), but ignore other crucial components such as the MLP layer following attention computation and the activation function.

In our work, we have the following improvements beyond previous work: ($i$) we enable almost linear time gradient computation across an entire transformer layer, incorporating both the MLP layer and the activation function; ($ii$) we extend the gradient calculation to include not only $W_{Q_i}, W_{K_i}$ but also $T_i(X)$ and $W_{V_i}$. These advancements collectively demonstrate a substantial leap forward from the methodologies in (Alman & Song, 2023) and (Alman & Song, 2024a).

# 4 Technical Overview

The main new challenge for our setting is the gradient with respect to the intermediate variables $T_i$, which previous work (Alman & Song, 2024a) on a single layer does not require. However, it is an essential component for multi-layer gradient computation. The gradient computation w.r.t. $T_i$ are not the same as that for the gradient w.r.t. $W_{Q_i}, W_{K_i}, W_{V_i}$ in a single layer. We give more details in Section 4.2.

In this section, we provide a brief overview of the proof techniques used throughout this paper.

## 4.1 Low-Rank Approximation for Attention Matrix

In this section, we delve into the crucial techniques behind our work: the low-rank approximation of the attention matrix, which is achieved through the polynomial method (Alman et al., 2020; Aggarwal & Alman, 2022). Drawing inspiration from (Alman & Song, 2023), the intuition of this approximation lies in the fact that the attention matrix $f(X) \in \mathbb{R}^{n \times n}$ (as defined in Definition 1.2), also referred to as the similarity matrix in attention mechanism, can be effectively approximated by

low-rank matrices $U_1, V_1 \in \mathbb{R}^{n \times k_1}$, where $k_1 = n^{o(1)}$. The naive method for calculating the attention matrix $f(X)$ has a time complexity of $O(n^2)$, whereas the input data $X \in \mathbb{R}^{n \times d}$ contains only $d \cdot n = n^{1+o(1)}$ entries. This discrepancy suggests the potential of using low-rank representations of $f(X)$ to design a fast algorithm.

An example of how to use the low-rank representations is the attention forward. First, note that approximating $f(X)$ alone does not lead to a fast algorithm since $U_1 V_1^\top$ still requires $n \times n$ entries. But by using the structure of the attention $\mathsf{Attn}(X) := f(X)V$ where $V = X W_V$, we can do it faster. By expressing $f(X)$ as $U_1 V_1^\top$, the attention forward becomes $\underbrace{U_1}_{n \times k_1} \underbrace{V_1^\top}_{k_1 \times n} \underbrace{V}_{n \times d}$. It is well known that different multiplication sequences can require dramatically different numbers of operations, so the order of matrix multiplications matters, which is indeed the case here. We first perform $V_1^\top V \in \mathbb{R}^{k_1 \times d}$ and this cost $O(k_1 n d) = n^{1+o(1)}$ time. Then we can compute $U_1 V_1^\top V$ within $O(n k_1 d) = n^{1+o(1)}$ time.

This method significantly reduces the computation time of the attention forward from $O(n^2)$ to almost linear time, $n^{1+o(1)}$. Driven by this technique and analyzing the close forms of the gradients, we extend the acceleration to the gradient of the entire model.

## 4.2 Accelerating Gradient Computation of $T_i(X)$

Based on the low-rank approximation method mentioned in Section 4.1, we compute the gradient of $L(X)$ with respect to the intermediate variable $T_i(X)$, which denotes the output of the $i$-th transformer layer. This computation is critical because, thanks to the chain rule, it enables us to calculate gradients for other gradient components.

**Extending to general loss functions.** According to the findings in (Deng et al., 2023b), the gradient $\frac{\mathrm{d}L(X)}{\mathrm{d}T_i(X)}$ can be decomposed into five components, namely $C_2(X), C_4(X), C_6(X), C_7(X), C_8(X)$, as detailed in Lemma D.1. In this work, we introduce a comprehensive analysis framework (Definition 2.1) and we have demonstrated its applicability to the cross-entropy loss (Remark 2.2). Consequently, by utilizing this generalized analysis framework, we extend the notation $L(X)$ to include a wide range of general loss functions.

**Accelerating the gradient computation.** A crucial aspect of speeding up gradient computation for the entire multi-layer transformer model involves accelerating the calculation of gradients with respect to the intermediate variables $T_i(X)$. The main challenge lies in the fact that computing the gradient of $T_i(X)$ requires calculating the gradients for other components within a transformer layer, including the residual connection, multi-head attention, and causal attention mask (see Section 5). We have conducted an extensive analysis of these components within the transformer layer (see Section I, J, and K) and demonstrated that, through the application of low-rank approximation techniques, the gradient $\frac{\mathrm{d}L(X)}{\mathrm{d}T_i(X)}$ can be computed in almost linear time $n^{1+o(1)}$ (Lemma 4.1). In particular, we apply the low-rank approximation technique on the five terms $C_2(X), C_4(X), C_6(X), C_7(X), C_8(X)$ respectively, demonstrating that each term can be computed in almost linear time, $n^{1+o(1)}$, as shown in Section E. Then, we aggregate those terms, as described in Section E.6. Since all five terms are $n \times d$ matrices, the summation of these terms takes $O(nd)$ time. We then conclude that for any single-layer transformer, the gradient computation with respect to the input can be performed in almost linear time $n^{1+o(1)}$, as stated in Lemma 4.1.

The statement made for a single transformer layer can be readily generalized to any layer within an $m$-layer transformer model. For instance, consider the intermediate variables $T_i(X)$ and $T_{i-1}(X)$ (as defined in Definition 2.3), where $T_i(X) = (g_i \circ \mathsf{Attn}_i)(T_{i-1}(X))$. Given the gradient $\frac{\mathrm{d}L(X)}{\mathrm{d}T_i(X)}$, as established in the previous paragraph, we compute the gradient with respect to $T_{i-1}(X)$, namely $\frac{\mathrm{d}L(X)}{\mathrm{d}T_{i-1}(X)}$, in almost linear time $n^{1+o(1)}$. For a multi-layer transformer model, the above process can be conducted recursively. Thus, we can compute the gradient of the loss function $L(X)$ on *any* $T_i(X)$ in almost linear time $n^{1+o(1)}$.

**Lemma 4.1** (Fast computation for $\frac{dL(X)}{dT_i(X)}$, informal version of Lemma E.11). *Let $L(X)$ be defined as Definition 2.1. Let $m$ denote the number of self-attention transformer layers (see Definition 1.3). Let $T_i(X)$ denote the intermediate variable output by $i$-th self-attention transformer layer (see Definition 2.3). We show that $\frac{dL(X)}{dT_i(X)}$ can be approximated in $n^{1+o(1)}$ time, with $1/\operatorname{poly}(n)$ approximation error.*

*Proof sketch.* In Lemmas E.3, E.5, E.6, E.8, and E.10, we have delineated several essential gradient components, $D_6, D_7, D_8, D_2, D_4 \in \mathbb{R}^{n \times d}$. We have established that these components can be computed in almost linear time $n^{1+o(1)}$, with the approximation error bounded by $\epsilon/\operatorname{poly}(n)$. Moreover, Lemma D.9 illustrates that the gradient w.r.t. $T_i$ can be expressed as the sum of these gradient components. That is, $\frac{dL(X)}{dT_{i-1}(X)} = \sum_{i \in \{2,4,6,7,8\}} D_i$. Given that the computational complexity of the summation operation is $O(nd)$, the aggregate time complexity for approximating the gradient $\frac{dL(X)}{dT_{i-1}(X)}$ with $\widetilde{g}_t$ remains $n^{1+o(1)}$. For the approximation error, by setting $\epsilon$ to $1/\operatorname{poly}(n)$, we ensure that the error of the gradient approximation $\widetilde{g}_t$ is also $1/\operatorname{poly}(n)$. $\square$

### 4.3 Accelerating Gradient Computation of $W_i$ and $W_{V_i}$

Let $W_i := W_{Q_i} W_{K_i}^\top$, with $W_{Q_i}$ and $W_{K_i}$ representing the query and key weight matrices, respectively, the gradients of $W_i$ and $W_{V_i}$ represent *all* trainable weight matrices in a transformer layer. Consequently, by determining the gradients for $W_i$ and $W_{V_i}$ across each layer, we achieve almost linear time gradient back-propagation throughout multi-layer transformer models.

**Fast gradient computation.** The prior study in (Alman & Song, 2024a) demonstrated that the gradient of $W_i$ can be computed in almost linear time. We extend their findings by adapting their approach to accommodate general loss function $L(X)$ (as defined in Definition 2.1) and further generalize their results to include the gradient computation for both $W_i$ and $W_{V_i}$ in each transformer layer (Lemma 4.2 and 4.3).

**Lemma 4.2** (Fast computation for $\frac{dL(X)}{dW_i}$, informal version of Lemma F.5). *Let $L(X)$ be defined as Definition 2.1, and $m$ be the number of self-attention transformer layers (Definition 1.3). For any $i \in [m]$, let $W_i = W_{Q_i} W_{K_i}^\top$, $W_{V_i} \in \mathbb{R}^{d \times d}$ denote the attention weight in the $i$-th transformer layer. We show that $\frac{dL(X)}{dW_i}$ can be approximated in $n^{1+o(1)}$ time, with $1/\operatorname{poly}(n)$ approximation error.*

**Lemma 4.3** (Fast computation for $\frac{dL(X)}{dW_{V_i}}$, informal version of Lemma G.4). *Let $L(X)$ be defined as Definition 2.1, and $m$ be the number of self-attention transformer layers (Definition 1.3). For any $i \in [m]$, let $W_i = W_{Q_i} W_{K_i}^\top$, $W_{V_i} \in \mathbb{R}^{d \times d}$ denote the attention weight in the $i$-th transformer layer. We show that $\frac{dL(X)}{dW_{V_i}}$ can be approximated in $n^{1+o(1)}$ time, with $1/\operatorname{poly}(n)$ approximation error.*

### 4.4 Accelerating Gradient Computation for Multi-Layer Transformers

In this section, our focus turns to extending the single-layer transformer result from the previous section to a multi-layer transformer.

**Running time analysis.** We derive the closed-form gradient for the non-attention components within a transformer layer $g_i$ (Definition 1.3). With the closed-form gradient of $g_i$ established in Lemma H.1, we then demonstrate in Lemma 4.4 that the gradient computation for $g_i$ can also be achieved in $n^{1+o(1)}$ time. Given that the number of layers $m$ is constant and the computation time for gradients on each layer is $n^{1+o(1)}$, we iteratively repeat this procedure for $m$ times. Therefore, the overall running time for computing gradients across the entire model is $m \cdot n^{1+o(1)} = n^{1+o(1)}$.

**Lemma 4.4** (Computation time for $G_i$, informal version of Lemma H.2). *Let $T_i(X)$ be defined as Definition 2.3, i.e. $T_i(X) = (g_i \circ \mathsf{Attn}_i)(T_{i-1}(X))$. Let $G_i \in \mathbb{R}^{n \times d}$ denote the gradient matrix resulting from the application of the chain rule up to the function $g_i$, i.e., $G_i = \frac{dL(X)}{d\mathsf{Attn}_i(T_{i-1}(X))}$. Assume we already have $\frac{dL(X)}{dT_i(X)}$. Assuming for any $Z \in \mathbb{R}^{n \times d}$, we have $g_i(Z) \in \mathbb{R}^{n \times d}$, and $g_i(Z) = \phi(Z \cdot W_g)$, where $W_g \in \mathbb{R}^{d \times d}$ and $\phi : \mathbb{R} \to \mathbb{R}$ denotes any element-wise activation function. Let $\phi'$ denote the derivative of $\phi$. Then, we show that $G_i$ can be computed in $n^{1+o(1)}$ time.*

**Error propagation analysis.** Here, we consider the approximation error. The approximation error originates from the low-rank approximation of the attention matrix, as detailed in Lemma C.13. As discussed in previous sections, the approximation error in each layer can be bounded by $1/\operatorname{poly}(n)$. Then, we only need to focus on how error propagates in different layers.

We first prove that our $1/\operatorname{poly}(n)$ approximation error statement holds for one layer transformer, as evidenced in Lemma H.3. Subsequently, through mathematical induction and leveraging the results of error propagation over the gradient of $g_i$, we show that the approximation error can be bounded by $1/\operatorname{poly}(n)$ for any $m$-layer transformer (Lemma 4.5), where $m$ is considered as constant.

**Lemma 4.5** (Multi-layer transformer gradient approximation, informal version of Lemma H.4)**.** *Let $L(X)$ be defined as Definition 2.1. Let $X$ be defined as Definition 1.2. Suppose we have a $m$-layer transformer (see Definition 1.3). Then, for any $i \in [m]$, we can show that: (i) Running time: Our algorithm can approximate $\frac{\mathrm{d}L(X)}{\mathrm{d}T_{i-1}(X)}$, $\frac{\mathrm{d}L(X)}{\mathrm{d}W_i}$, and $\frac{\mathrm{d}L(X)}{\mathrm{d}W_{V_i}}$ in $n^{1+o(1)}$ time; (ii) Error bound: The approximation of the entire transformer model can be bounded by $1/\operatorname{poly}(n)$. Namely, our algorithm output $\widetilde{g}$ satisfies $\|\widetilde{g} - \frac{\mathrm{d}L(X)}{\mathrm{d}X}\|_\infty \leq 1/\operatorname{poly}(n)$.*

The rate of error accumulation in a transformer with $m$ layers grows exponentially as $n^m$. Namely, the error increases from $1/\operatorname{poly}(n)$ to $n^m/\operatorname{poly}(n)$. Nevertheless, because $m$ is a constant and the polynomial $\operatorname{poly}(n)$ has a high degree, the total error remains insignificant in practical scenarios.

## 5 EXTENSIONS

**Multi-head attention and residual connections.** Multi-head attention and residual connections are important components in attention mechanisms. These components were not involved in our initial analysis for simplicity. Incorporating them into our algorithm is straightforward. This suggests that our algorithm can be readily adapted to more practical transformer models. The detailed analysis for incorporating residual connection can be found in Section J and Lemma J.3. For the synergy with multi-head attention, we provide comprehensive analysis in Section K and Lemma K.2.

**Causal attention mask.** The causal attention mask is critical to prevent transformers from "cheating" during training by ensuring future information is not used. The full-rank characteristic of the causal attention mask poses challenges for low-rank approximations. Nevertheless, we have identified a method to accelerate the computation of causal masked attention by exploiting its inherent properties, showing almost linear time complexity. A comprehensive explanation is provided in Section B.3. More detailed analysis can be found in Section I and Lemma I.7 and I.8.

**Prompt tuning.** Prompt tuning is a prevalent approach in parameter-efficient fine-tuning (PEFT), which requires the calculation of gradients on input data $X$. Given our algorithm can compute gradients for intermediate variables $T_i$ in almost linear time, we can adapt this acceleration to the gradient for the input data $X$, thus enhancing the efficiency of the prompt tuning process. Additional details are provided in Section B.5.

**Synergy with system-level attention acceleration.** Many contemporary works focus on system-level acceleration of attention mechanisms, often by leveraging caching and mitigating I/O bottlenecks. Our algorithm has the potential to integrate with such advancements. By combining our theoretical improvements in computation time (from $O(n^2)$ to $n^{1+o(1)}$) with system-level optimizations, the overall efficiency of attention mechanism computation may improve further. We leave the implementation of our method on GPU as future work. More details can be found in Section B.4.

## 6 CONCLUSION

In this work, we proposed a novel algorithm (Algorithm 1), which can approximately train a multi-layer transformer model in almost linear time, introducing only a small error. Importantly, our algorithm is designed to be compatible with general loss functions, practical sub-modules (residual connection, casual mask, multi-head attention), and general gradient-based algorithms. It may be seamlessly integrated with other system-level acceleration techniques. With experimental support, we believe our finding is able to accelerate the training of LLMs in practice.

## ETHIC STATEMENT

This paper does not involve human subjects, personally identifiable data, or sensitive applications. We do not foresee direct ethical risks. We follow the ICLR Code of Ethics and affirm that all aspects of this research comply with the principles of fairness, transparency, and integrity.

## REPRODUCIBILITY STATEMENT

We ensure reproducibility of our theoretical results by including all formal assumptions, definitions, and complete proofs in the appendix. The main text states each theorem clearly and refers to the detailed proofs. No external data or software is required.

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

# Appendix

**Roadmap.** In Section A, we provide related works of this paper. In Section B, we provide a detailed discussion about several potential extensions of our framework. In Section C, we introduce basic notations and concepts used in our paper, along with the low-rank approximation technique introduced in (Alman & Song, 2023) and (Alman & Song, 2024a). In Section D, we provide details about how we integrate the gradient of $T_i(X)$ into matrix form. In Section E, we explain how to apply the low-rank approximation technique to accelerate the computation for the gradient on $T_i(X)$. In Section F, we extend the result of (Alman & Song, 2024a) to arbitrary loss functions and accelerate the computation of gradient on $W$ via the low-rank approximation technique. In Section G, we calculate the gradient on $W_V$ and accelerate the computation of the gradient on $W_V$. In Section H, with the help of math induction, we analyze the time complexity and the approximation error across the entire model. In Section I, we discuss how our framework can expand to an attention mechanism with a causal attention mask. In Section J, we provide details about how to integrate our framework with attention mechanism with the residual connection. In Section K, we argue that, with the addition of multi-head attention, our algorithm can still achieve almost linear time gradient computation. In Section **??**, we discuss the limitation of this work. Finally, in Section **??**, we provide a discussion about potential societal impact of this work.

## A  RELATED WORK

**Long-context modeling in LLMs.** As LLMs grow in size and capability, in-context learning (ICL) (Min et al., 2022; Shi et al., 2024b; Xu et al., 2024b; Chen et al., 2024a) has become a preferred method for directing these models to perform a variety of tasks, as opposed to the resource-intensive process of fine-tuning. Nonetheless, research has indicated that longer prompts can impair LLMs' performance due to the limitation on maximum sequence length during pre-training (Li et al., 2024b). Consequently, extending the maximum sequence length during pre-training and fine-tuning stages is imperative. Enhancing training efficiency is crucial given the prevalent use of the Transformer architecture in LLMs, which incurs a quadratic computational cost relative to sequence length. Addressing this challenge, some studies have explored continued fine-tuning of LLMs with extended context lengths (Tworkowski et al., 2024), while others have experimented with the interpolation and extrapolation capabilities of positional embedding (Chen et al., 2023). (Shi et al., 2024a) handles long context by compressing the input tokens. However, these approaches have not fundamentally addressed the core issue: the quadratic computational cost associated with sequence length in the attention mechanism (Keles et al., 2023; Fournier et al., 2023). In this study, we delve into accelerating the attention mechanism, thereby addressing the long-context modeling issue at its essence.

**Attention acceleration.** Attention mechanism has faced criticism due to its quadratic time complexity with respect to context length, a concern exacerbated by the increasing length in modern large language models (LLMs) such as GPT-4 (Achiam et al., 2023), Claude 3.5 (Anthropic, 2024), Llama 3.1 (Touvron et al., 2023; Llama Team, 2024), etc. Nevertheless, this limitation can be circumvented by employing polynomial kernel approximation techniques (Aggarwal & Alman, 2022), which enable the derivation of a low-rank representation of the attention matrix. This innovation significantly accelerates both the training and inference processes of a single attention layer, achieving almost linear time complexity (Alman & Song, 2023; 2024a), while our work supports both training and inference for any multi-layer transformer. The foundational concept underpinning the work of (Alman & Song, 2023; 2024a) is the extension of the notion that polynomials can effectively approximate exponential functions to the domain of matrices. Given that each entry of the attention matrix is activated by a softmax function, the author of (Alman & Song, 2023) proposed the use of a polynomial matrix to approximate the softmax-activated attention matrix. Additionally, they demonstrated that this polynomial matrix can be factorized into the product of two low-rank matrices. By strategically reordering the sequence of matrix multiplications, these low-rank matrices are employed to diminish the computational complexity of the attention mechanism's forward pass to almost linear time. For more details, please refer to Section 3 in (Alman & Song, 2023). Furthermore, this approach can be extended to higher-order attention mechanisms, i.e., tensor attention (Alman & Song, 2024b; Liang et al., 2024h). Moreover, there are other theoretical approaches. For instance, (Liang et al., 2024a) introduces the conv-basis method to accelerate attention computation.

(Han et al., 2024) proposes a near-linear time algorithm under the assumptions of uniform softmax column norms and sparsity.

**Attention mechanism.** Attention mechanisms, including self-attention and cross-attention, are pivotal techniques employed in state-of-the-art neural networks. Since it was introduced in (Vaswani et al., 2017), it has gained widespread adoption across various domains. In particular, it is integral to decoder-only LLMs (Radford et al., 2019) and the Vision Transformer (ViT) architecture (Dosovitskiy et al., 2020). The former has been instrumental in the remarkable success of LLMs, while the latter has significantly advanced the field of computer vision, encompassing applications such as image generation (Rombach et al., 2022; Wang et al., 2023c; 2024b), detection (Li et al., 2022), segmentation (Zhang et al., 2022), and layout generation (Gupta et al., 2021; Chai et al., 2023; Wang et al., 2023a). Moreover, attention mechanism can be integrated into multi-modal models (Xu et al., 2021a; Zhang et al., 2024a; Liang et al., 2024h; Wang et al., 2024a), math reasoning (Li et al., 2024a), diffusion models (Peebles & Xie, 2023; Liang et al., 2024f; Hu et al., 2024f; Esser et al., 2024; Ma et al., 2024b; Li et al., 2024g), differential privacy (Behnia et al., 2022; Shi et al., 2022; Wang et al., 2023b; Liang et al., 2024g; Singh et al., 2024; Chu et al., 2023; Liang et al., 2024c; Li et al., 2024d; Song et al., 2023a) and many other techniques (Liang et al., 2024d; Li et al., 2024f; Qin et al., 2023a;b;c; Song et al., 2023b; Xiao et al., 2024; Viswanathan et al., 2023).

**Attention theory.** (Bahdanau et al., 2014) introduced attention mechanisms in NLP, enhancing encoder-decoder architecture with variable-length vectors to improve machine translation. Building on this, (Luong et al., 2015) developed local and global attention variants, further refining NLP tasks. Recent Large Language Model research has focused extensively on attention computation (Deng et al., 2023a; Alman & Song, 2023; Zandieh et al., 2023). Studies by (Zandieh et al., 2023; Chen et al., 2020; Kitaev et al., 2020) use Locality Sensitive Hashing for attention approximation, with (Zandieh et al., 2023) offering efficient dot-product attention. (Brand et al., 2023) and (Alman & Song, 2023) explore static and dynamic attention calculations, while (Li et al., 2023b) investigates hyperbolic regression regularization. (Deng et al., 2023a) proposes algorithms for reducing attention matrix dimensionality in LLMs. Attention has also been examined from optimization and convergence perspectives (Li et al., 2023a; Gao et al., 2023a; Snell et al., 2021; Zhang et al., 2020), investigating word co-occurrence learning (Li et al., 2023a), regression problems with exponential activation functions (Gao et al., 2023a), attention mechanism evolution during training (Snell et al., 2021), and the impact of heavy-tailed noise on stochastic gradient descent (Zhang et al., 2020). Theoretical explorations of attention variants include quantum attention (Gao et al., 2023c), tensor attention (Alman & Song, 2024b; Liang et al., 2024h), and differentially private attention (Liang et al., 2024g; Gao et al., 2023b; Liang et al., 2024c).

**More methods for model acceleration.** Various techniques have been developed for model acceleration. One approach involves modifying model architectures to enable faster inference, such as Mamba (Gu & Dao, 2023), Linearizing Transformers (Zhang et al., 2024b), PolySketchFormer (Kacham et al., 2023), and the Hopfield Model (Hu et al., 2024b;a; Wu et al., 2024a; Xu et al., 2024a; Hu et al., 2024c; Wu et al., 2024b; Hu et al., 2023; 2024e) and so on. Another line of work is to prune the weights in a neural network to reduce running time and memory consumption (Hubara et al., 2021; Jin et al., 2022; Frantar & Alistarh, 2022; 2023; Sun et al., 2024; Li et al., 2024c; Liang et al., 2024b). In addition, specific techniques have been developed to accelerate LLM generation (Chen et al., 2024b;a; Song & Yang, 2023; Li et al., 2024e).

## B DISCUSSION AND EXTENSION DETAILS

In Section B.1, we argue that our framework can easily adapt to the multi-head attention mechanism. In Section B.2, we introduce how to integrate residual connection to our framework. In Section B.3, we detail the integration of the causal attention mask into our algorithm. In Section B.4, we discuss the possibility of the synergy between our theoretical side attention acceleration and the existing system-level attention acceleration mechanism. In Section B.5, we show how to expedite prompt tuning using our results.

## B.1 MULTI-HEAD ATTENTION

The multi-head attention mechanism was first introduced by (Vaswani et al., 2017). This innovation allows a token to simultaneously attend to multiple positions within the same layer, thereby enriching the model's capacity for capturing various dependencies. However, this enhanced capability comes with an increase in the size of the attention matrix $f(X)$ from $1 \times n \times n$ to $h \times n \times n$, where $h$ is the number of attention heads. To mitigate the computational burden, each head's vector is derived by splitting the original vector, reducing the dimensionality of each head to $d_h := d/h$. To summarize, the key distinctions between multi-head and single-head attention are (1) an enlarged attention matrix $f(X)$ and (2) a reduced dimensionality $d_h$ within each attention head.

**Enlarged attention matrix.** As previously discussed, the attention matrix's dimensionality increases with the number of heads, $h$. Despite this expansion, the application of the low-rank approximation technique, as outlined in Section 4.1, ensures that the computation time for the attention matrix remains almost linear. Specifically, for a constant number of heads $h$ in the multi-head mechanism, the time complexity for computing $f(X) \in \mathbb{R}^{h \times n \times n}$ is $h \cdot n^{1+o(1)} = n^{1+o(1)}$.

**Reduced dimensionality.** Another differentiating factor of multi-head attention is the lower dimensionality processed by each head, i.e. $d_h := d/h$, compared the full $d$ in single-head attention. This reduction ensures that the gradient computation time does not increase with the introduction of multiple attention heads.

We provide comprehensive analysis of the synergy of our algorithm with multi-head attention in Section K. We first prove in Lemma K.2, with the addition of multi-head attention, the gradient over the attention mechanism can be computed in almost linear time. Then, we further prove that for any multi-layer transformer, with multi-head attention, the gradient can be computed in almost linear time as well.

## B.2 RESIDUAL CONNECTION

Residual connection is a pivotal technique in deep neural network architectures, effectively addressing issues such as vanishing and exploding gradients during training process, and facilitating faster convergence of the model. Residual connection is also integrated into the standard attention mechanism. Formally, given the intermediate variable $T_i(X)$ output by the $i$-th transformer layer as defined in Definition 2.3, we provide the formal definition of residual connection in Definition J.1 and J.2. Since the residual connection only brings an additional add operation to each component and with $T_i(X)$ belonging to the space $\mathbb{R}^{n \times d}$, the residual connection introduces only a marginal computational overhead of $O(n \cdot d)$ per layer. Consequently, the total computational cost for each layer is $O(n \cdot d) + n^{1+o(1)} = n^{1+o(1)}$. Hence, by intuition, the inclusion of residual connections does not compromise the overall complexity of our method.

The detailed analysis is provided in Section J, where we first prove in Lemma J.3, that if the gradient over one structure can be computed in almost linear time, then with the addition of the residual connection, the gradient can also be computed in almost linear time. Then we use math induction to extend our result to the entire multi-layer transformer model.

## B.3 CAUSAL ATTENTION MASK

In transformer training, attention mask is a crucial component, designed to prevent a given token from attending to future tokens in the sequence. Causal attention mask is a widely used attention mask, which is configured as a lower triangular matrix, where elements on or below the main diagonal are ones, with all other entries being zeros.

Now we describe how to incorporate this into our algorithm. Let $M \in \{0,1\}^{n \times n}$ represent the causal attention mask (see Definition I.2). Let $\widehat{f}(X) := D^{-1}(M \odot A)$ where $A = \exp(XWX^\top/d)$ and $D := \text{diag}((M \odot A) \cdot \mathbf{1}_n)$. Lemma I.1 reveals that $A$ has a low-rank representation given by $U_0 V_0^\top$. Using Lemma I.3, we know $(M \odot (U_0 V_0^\top)) \cdot v$ for any vector $v \in \mathbb{R}^n$ can be computed in almost linear time.

To integrate the causal mask into the gradient computation within each transformer layer, we first find all instances that have the structure of $f(X) \cdot H$ or $(f(X) \odot (UV^\top)) \cdot H$, where $H, U, V$ are low rank matrices. Then, we replace $f(X)$ with $\widehat{f}(X)$ in these instances. More detailed analysis of causal attention can be found in Section I. To be more specific, we group the gradient components for $T_i, W_i, W_{V_i}$ into two categories, one for dot product (Lemma I.7), another for Hadamard product (Lemma I.8). After showing each component can be calculated in almost linear time, the overall gradient computation remains $n^{1+o(1)}$ time. Thus, our framework can seamlessly accommodate causal attention masks.

### B.4 System-level attention acceleration

The attention computing acceleration involves a two-pronged strategy that leverages both system-level improvements (e.g. Flash Attention (Dao et al., 2022; Dao, 2023; Shah et al., 2024)) and the theoretical time complexity improvements (e.g. our work and (Han et al., 2024)).

Numerous efforts have been made in the literature to accelerate attention calculations at the system level. For instance, Flash Attention (Dao et al., 2022; Dao, 2023; Shah et al., 2024) targets the I/O bottleneck inherent in attention mechanisms. Studies such as block-wise parallel decoding (Stern et al., 2018) focus on implementing parallel decoding within transformer models to enhance inference speed. Additionally, recent advancements in the field of speculative decoding, such as Medusa (Cai et al., 2024), leverage a smaller, more efficient model to generate predictions, with the larger model only responsible for validating, the smaller model's outputs (Leviathan et al., 2023).

Despite these innovations, the aforementioned methods do not address the fundamental quadratic time complexity $O(n^2)$ of the attention mechanisms. This presents an opportunity to complement our low-rank approximation technique, with these system-level optimizations, thereby achieving an even greater acceleration in attention computation. For instance, we could design an I/O-aware algorithm for Algorithm 1, similar to the approach taken by Flash Attention, to effectively leverage GPU acceleration.

To implement our algorithm practically on GPU, we have some coding challenges to fix: (1) we need to define some new tensor operations in PyTorch, e.g. Eq. (5), Eq. (8); (2) we need to systematically re-implement some back-propagation function of the current PyTorch function; (3) we need to implement some CUDA function to run our algorithm in parallel for the casual mask, see discussion in Section B.3. We may leave this as our future work.

### B.5 Prompt tuning

Prompt tuning, as introduced by various studies (Li & Liang, 2021; Lester et al., 2021; Liu et al., 2022; Mu et al., 2024; Hu et al., 2024d; Liang et al., 2024e), has emerged as a parameter-efficient fine-tuning strategy for large language models (LLMs). Specifically, prompt tuning involves adjusting "soft prompts" conditioned on frozen LLMs. This method requires relatively small number of tuneable parameters compared with fine-tuning the entire LLMs, making it a popular choice for conserving training resources, including data and computational power.

The analysis reveals that the essence of prompt tuning involves computing gradients with respect to the soft prompts $X_p$ across the entire model. In both prompt tuning and full fine-tuning, the quadratic $O(n^2)$ computational complexity of gradient calculation remains the same due to the self-attention mechanism inherent in LLMs.

In this work, leveraging the low-rank approximation technique discussed in Section 4.1, our algorithm (Algorithm 1) efficiently computes gradients on soft prompts $X_p$ over the entire model in almost linear time. This suggests that our method is universal and can also be applied within traditional prompt tuning frameworks.

## C Preliminary on Gradient Calculation

In Section C.1, we list several useful math facts used in the following sections of this paper. In Section C.2, we provide the close forms of the gradient components. In Section C.3, we introduce some mathematical definitions to facilitate understanding of gradient calculations. In Section C.4,

we list some low rank approximation technique introduced in (Alman & Song, 2023) and (Alman & Song, 2024a). In Section C.5, we demonstrate that the entries of matrices defined in Section C.3 are bounded.

**Notations.** For two vectors $x \in \mathbb{R}^n$ and $y \in \mathbb{R}^n$, we use $\langle x, y \rangle$ to denote the inner product between $x, y$. Namely, $\langle x, y \rangle = \sum_{i=1}^{n} x_i y_i$. We use $e_i$ to denote a vector where only $i$-th coordinate is 1, and other entries are 0. For each $a, b \in \mathbb{R}^n$, we use $a \odot b \in \mathbb{R}^n$ to denote the Hardamard product, i.e. the $i$-th entry of $(a \odot b)$ is $a_i b_i$ for all $i \in [n]$. We use $\mathbf{1}_n$ to denote a length-$n$ vector where all the entries are ones. We use $\|A\|_\infty$ to denote the $\ell_\infty$ norm of a matrix $A \in \mathbb{R}^{n \times d}$, i.e. $\|A\|_\infty := \max_{i \in [n], j \in [d]} |A_{i,j}|$. We use $\mathrm{poly}(n)$ to denote polynomial time complexity with respective to $n$.

## C.1 BASIC MATH FACTS

In this section, we provide some useful basic math facts,

**Fact C.1.** *Let $x, y, z \in \mathbb{R}^n$. Then we have*

- $\langle x \odot y, z \rangle = x^\top \mathrm{diag}(y) z$.

- $\langle x, (y \odot z) \rangle = \langle y, (x \odot z) \rangle = \langle z, (y \odot x) \rangle$

- $\langle x, y \rangle = \langle x \odot y, \mathbf{1}_n \rangle$.

Then, we introduce a classical folklore used for the Hadamard product of two matrices.

**Fact C.2** (Folklore, (Alman & Song, 2024a)). *Let $U_1, V_1 \in \mathbb{R}^{n \times k_1}$. Let $U_2, V_2 \in \mathbb{R}^{n \times k_2}$. Then we have*

$$
(\underbrace{U_1}_{n \times k_1} \underbrace{V_1^\top}_{k_1 \times n}) \odot (\underbrace{U_2}_{n \times k_2} \underbrace{V_2^\top}_{k_2 \times n}) = \underbrace{(U_1 \oslash U_2)}_{n \times k_1 k_2} \underbrace{(V_1 \oslash V_2)^\top}_{k_1 k_2 \times n}
$$

*Here, given $U_1 \in \mathbb{R}^{n \times k_1}$ and $U_2 \in \mathbb{R}^{n \times k_2}$, the $U_1 \oslash U_2 \in \mathbb{R}^{n \times k_1 k_2}$ is the row-wise Kronecker product, i.e., $(U_1 \oslash U_2)_{i, l_1 + (l_2 - 1)k_1} := (U_1)_{i, l_1} (U_2)_{i, l_2}$ for all $i \in [n]$, $l_1 \in [k_1]$ and $l_2 \in [k_2]$.*

## C.2 CLOSE FORM OF THREE GRADIENT COMPONENTS

We first restate the definition of self-attention, where we denote $W := W_Q W_K^\top \in \mathbb{R}^{d \times d}$ for simplicity.

**Definition C.3** (Self-attention module). *Let $X \in \mathbb{R}^{n \times d}$ denote the input sequence, where $n$ is the number of input tokens and $d$ is the hidden dimension size. Let $W_V \in \mathbb{R}^{d \times d}$ be the value weight matrix, and let $W := W_Q W_K^\top \in \mathbb{R}^{d \times d}$ be the key-query weight matrix. The self-attention function* $\mathsf{Attn}(X)$ *with weights $W, W_V$ is:*

$$
\mathsf{Attn}(X) = \mathsf{Softmax}(XWX^\top / d) \cdot X \cdot W_V.
$$

*where* $\mathsf{Softmax}$ *is applied to each row of its input matrix. The attention can be re-written as:*

$$
\mathsf{Attn}(X) = f(X) \cdot X \cdot W_V,
$$

*where (1) $A := \exp(XWX^\top / d) \in \mathbb{R}^{n \times n}$ and $\exp$ is applied element-wise, (2) $D := \mathrm{diag}(A\mathbf{1}_n) \in \mathbb{R}^{n \times n}$, and (3) $f(X) := D^{-1}A \in \mathbb{R}^{n \times n}$ is the attention matrix.*

Note that the gradient of $W_Q$ and $W_K$ can easily be calculated from the gradient of $W$, i.e.,

$$
\frac{\mathrm{d}L(X)}{\mathrm{d}W_Q} = \frac{\mathrm{d}L(X)}{\mathrm{d}W} \cdot \frac{\mathrm{d}W}{\mathrm{d}W_Q}
$$
$$
= \frac{\mathrm{d}L(X)}{\mathrm{d}W} \cdot W_K
$$

where the first step follows from the chain rule, and the second step follows from basic calculus.

Then, we show how to derive the close form for the gradient components within each layer of a multi-layer transformer.

**Lemma C.4** (Close form of gradient components, formal version of Lemma 2.4). *If we have the below conditions,*

- *Let $L(X)$ be defined as Definition 2.1.*

- *Let $W_i := W_{Q_i} W_{K_i}^\top \in \mathbb{R}^{d \times d}$ be the key-query weight matrix, $W_{V_i} \in \mathbb{R}^{d \times d}$ be the value weight matrix for the $i$-th transformer layer.*

- *Let $T_i(X)$ denote the intermediate variable output by $i$-th self-attention transformer layer (see Definition 2.3).*

- *Let $G_i \in \mathbb{R}^{n \times d}$ denote the gradient matrix resulting from the application of the chain rule up to the function $g_i$, i.e., $G_i = \frac{\mathrm{d}L(X)}{\mathrm{d}\mathsf{Attn}_i(T_{i-1}(X))}$.*

- *For $i_2 \in [n], j_2 \in [d]$, let $G_i(i_2, j_2)$ denote the $(i_2, j_2)$-th entry of $G_i$, let $\frac{\mathrm{d}\mathsf{Attn}_i(T_{i-1}(X))_{i_2,j_2}}{\mathrm{d}T_{i-1}(X)} \in \mathbb{R}^{n \times d}$ denote the gradient of $(i_2, j_2)$-th entry of $\mathsf{Attn}_i(T_{i-1}(X))$.*

*Then, we can show that*

- **Part 1.**

$$\frac{\mathrm{d}L(X)}{\mathrm{d}T_{i-1}(X)} = \sum_{i_2=1}^{n} \sum_{j_2=1}^{d} G_i(i_2, j_2) \cdot \frac{\mathrm{d}\mathsf{Attn}_i(T_{i-1}(X))_{i_2,j_2}}{\mathrm{d}T_{i-1}(X)}.$$

- **Part 2.**

$$\frac{\mathrm{d}L(X)}{\mathrm{d}W_i} = \sum_{i_2=1}^{n} \sum_{j_2=1}^{d} G_i(i_2, j_2) \cdot \frac{\mathrm{d}\mathsf{Attn}_i(T_{i-1}(X))_{i_2,j_2}}{\mathrm{d}W_i}.$$

- **Part 3.**

$$\frac{\mathrm{d}L(X)}{\mathrm{d}W_{V_i}} = \sum_{i_2=1}^{n} \sum_{j_2=1}^{d} G_i(i_2, j_2) \cdot \frac{\mathrm{d}\mathsf{Attn}_i(T_{i-1}(X))_{i_2,j_2}}{\mathrm{d}W_{V_i}}.$$

*Proof.* We have

- $L(X) \in \mathbb{R}$.

- $\mathsf{Attn}_i(T_{i-1}(X)) \in \mathbb{R}^{n \times d}, T_{i-1}(X) \in \mathbb{R}^{n \times d}$.

- $W_i \in \mathbb{R}^{d \times d}, W_{V_i} \in \mathbb{R}^{d \times d}$.

Therefore, we have

- $\frac{\mathrm{d}L(X)}{\mathrm{d}T_{i-1}(X)} \in \mathbb{R}^{n \times d}$, $\frac{\mathrm{d}\mathsf{Attn}_i(T_{i-1}(X))}{\mathrm{d}T_{i-1}(X)} \in \mathbb{R}^{(n \times d) \times (n \times d)}$.

- $\frac{\mathrm{d}L(X)}{\mathrm{d}W_i} \in \mathbb{R}^{d \times d}$, $\frac{\mathrm{d}\mathsf{Attn}_i(T_{i-1}(X))}{\mathrm{d}W_i} \in \mathbb{R}^{(n \times d) \times (d \times d)}$.

- $\frac{\mathrm{d}L(X)}{\mathrm{d}W_{V_i}} \in \mathbb{R}^{d \times d}$, $\frac{\mathrm{d}\mathsf{Attn}_i(T_{i-1}(X))}{\mathrm{d}W_{V_i}} \in \mathbb{R}^{(n \times d) \times (d \times d)}$.

Then, simply applying chain rule, we can get the final results. $\qquad\square$

## C.3 BASIC NOTATIONS FOR COMPUTING GRADIENTS

We remark that, in this section, for convenience of computing a closed form for the gradient, we ignore the $1/d$ factor in function Softmax. Since it is only a rescaling factor, it won't affect how we compute these matrices in general.

Before we move on to compute gradients, we need to define some useful notations.

We begin with introducing the index for a matrix.

**Definition C.5** (Simplified notations). *For any matrix $Z \in \mathbb{R}^{n \times d}$, for $i \in [n], j \in [d]$, we have following definitions:*

- *Let $\underbrace{Z_{i,j}}_{\text{scalar}}$ and $Z(i,j)$ denote the $(i,j)$-th entry of $Z$.*

- *Let $\underbrace{Z_{i,*}}_{d \times 1}$ and $Z(i,*)$ denote the $i$-th row of $Z$.*

- *Let $\underbrace{Z_{*,j}}_{n \times 1}$ and $Z(*,j)$ denote the $j$-th column of $Z$.*

Then, we define the exponential matrix in the attention mechanism.

**Definition C.6** (Exponential function $u$). *If we have the below conditions,*

- *Let $X \in \mathbb{R}^{n \times d}$*

- *Let $W := W_Q W_K^\top \in \mathbb{R}^{d \times d}$*

*We define $u(X) \in \mathbb{R}^{n \times n}$ as follows*

$$u(X) := \exp(XWX^\top)$$

Then, we introduce the summation vector of the aforementioned exponential matrix.

**Definition C.7** (Sum function of softmax $\alpha$). *If we have the below conditions,*

- *Let $X \in \mathbb{R}^{n \times d}$*

- *Let $u(X)$ be defined as Definition $C.6$*

*We define $\alpha(X) \in \mathbb{R}^n$ as follows*

$$\alpha(X) := u(X) \cdot \mathbf{1}_n$$

Then, with the help of the summation vector, we are ready to normalize the exponential matrix and get the softmax probability matrix.

**Definition C.8** (Softmax probability function $f$). *If we have the below conditions,*

- *Let $X \in \mathbb{R}^{n \times d}$*

- *Let $u(X) \in \mathbb{R}^{n \times n}$ be defined as Definition $C.6$*

- *Let $\alpha(X) \in \mathbb{R}^n$ be defined as Definition $C.7$*

*We define $f(X) \in \mathbb{R}^{n \times n}$ as follows*

$$f(X) := \mathrm{diag}(\alpha(X))^{-1} u(X)$$

*where we define $f(X)_{j_0}^\top \in \mathbb{R}^n$ is the $j_0$-th row of $f(X)$.*

Besides the probability matrix introduced above, we introduce the value matrix in the following definition.

**Definition C.9** (Value function $h$)**.** *If we have the below conditions,*

- *Let $X \in \mathbb{R}^{n \times d}$*

- *Let $W_V \in \mathbb{R}^{d \times d}$*

*We define $h(X) \in \mathbb{R}^{n \times d}$ as follows*

$$h(X) = XW_V$$

Then, we introduce $s(X)$ to represent the output of the attention mechanism.

**Definition C.10** (Self-attention output $s$)**.** *If we have the below conditions,*

- *Let $f(X)$ be defined as Definition $C.8$*

- *Let $h(X)$ be defined as Definition $C.9$*

*We define $s(X) \in \mathbb{R}^{n \times d}$ as follows*

$$s(X) = f(X)h(X)$$

Then, we introduce $q(X)$ and $p(X)$ to facilitate the calculation of the gradient on $W$.

**Definition C.11** (Definition of $q(X)$)**.** *If we have the below conditions,*

- *Let $h(X) \in \mathbb{R}^{n \times d}$ be defined as in Definition C.9.*

- *Let $G_i \in \mathbb{R}^{n \times d}$ denote the gradient matrix resulting from the application of the chain rule up to the function $g_i$, i.e., $G_i = \frac{\mathrm{d}L(X)}{\mathrm{d}\mathsf{Attn}_i(T_{i-1}(X))}$.*

- *For $i_2 \in [n], j_2 \in [d]$, let $G_i(i_2, j_2)$ denote the $(i_2, j_2)$-th entry of $G_i$.*

*We define $q(X) \in \mathbb{R}^{n \times n}$ as*

$$q(X) = \underbrace{G_i}_{n \times d} \underbrace{h(X)^\top}_{d \times n}.$$

*where we define $q(X)_{j_0}^\top \in \mathbb{R}^n$ is the $j_0$-th row of $q(X)$.*

**Definition C.12** (Definition of $p(X)$, Definition C.5 in (Alman & Song, 2024a))**.** *For every index $j_0 \in [n]$, we define $p(X)_{j_0} \in \mathbb{R}^n$ as*

$$p(X)_{j_0} := (\mathrm{diag}(f(X)_{j_0}) - f(X)_{j_0}f(X)_{j_0}^\top)q(X)_{j_0}$$

*where we have $p(X) \in \mathbb{R}^{n \times n}$ and we define $p(X)_{j_0}^\top \in \mathbb{R}^n$ is the $j_0$-th row of $p(X)$.*

*Furthermore, we define $p_1(X) = f(X) \odot q(X)$ and $p_2(X) = \mathrm{diag}(p_1(X) \cdot \mathbf{1}_n)f(X)$. Additionally, we can calculate $p(X)$ as*

$$p(X) = p_1(X) - p_2(X)$$

## C.4 LOW RANK REPRESENTATIONS

Using (Alman & Song, 2023)'s polynomial method techniques, we can obtain the following low-rank representation result.

**Lemma C.13** (Low rank representation to $f$, Section 3 of (Alman & Song, 2023), Lemma D.1 of (Alman & Song, 2024a))**.** *For any $R = o(\sqrt{\log n})$, there exists a $k_1 = n^{o(1)}$ such that: Let $X \in \mathbb{R}^{n \times d}$ and $W \in \mathbb{R}^{d \times d}$ be a square matrix. It holds that $\|XW\|_\infty \leq R, \|X\|_\infty \leq R$, then there are two matrices $U_1, V_1 \in \mathbb{R}^{n \times k_1}$ such that $\|U_1V_1^\top - f(X)\|_\infty \leq \epsilon/\mathrm{poly}(n)$. Here $f(X) = D^{-1}\exp(XWX^\top)$ (see also Definition C.8) and we define $D = \mathrm{diag}(\exp(XWX^\top)\mathbf{1}_n)$ (see also Definition C.7). Moreover, these matrices $U_1, V_1$ can be explicitly constructed in $n^{1+o(1)}$ time.*

A similar technique can be applied to $s(X)$.

**Lemma C.14** (Low rank representation to $s$). *Let $d = O(\log n)$. Assume that each number in the $n \times d$ matrices $h(X) \in \mathbb{R}^{n \times d}$ can be written using $O(\log n)$ bits. Let $n \times d$ matrix $s(X) \in \mathbb{R}^{n \times d}$ be defined as Definition C.10. Then, there are two matrices $U_1, V_1 \in \mathbb{R}^{n \times k_1}$ we have $\|U_1 V_1^\top h(X) - s(X)\|_\infty \le \epsilon / \operatorname{poly}(n)$.*

*Proof.* We can show that

$$\|U_1 V_1^\top h(X) - s(X)\|_\infty = \|U_1 V_1^\top h(X) - f(X)h(X)\|_\infty$$

$$= \|\underbrace{(U_1 V_1^\top}_{n \times n} - \underbrace{f(X))}_{n \times n} \underbrace{h(X)}_{n \times d}\|_\infty$$

$$\le n \|\underbrace{U_1 V_1^\top}_{n \times n} - \underbrace{f(X)}_{n \times n}\|_\infty \|\underbrace{h(X)}_{n \times d}\|_\infty$$

$$\le n \|\underbrace{U_1 V_1^\top}_{n \times n} - \underbrace{f(X)}_{n \times n}\|_\infty \cdot \operatorname{poly}(n)$$

$$\le \epsilon / \operatorname{poly}(n)$$

where the 1st step is from the choice of $s(X)$, the 2nd step comes from $AC - BC = (A - B)C$ holds for any matrices $A$, $B$, and $C$, the 3rd step is because of basic linear algebra, the 4th step is due to each number in $h(X)$ can be written using $O(\log(n))$ bits, the fifth step follows from $\|U_1 V_1^\top - f(X)\|_\infty \le \epsilon / \operatorname{poly}(n)$.

$\square$

We can also get a low-rank representation of $p_1(x)$ and $p_2(x)$.

**Lemma C.15** (Low rank representation to $p_1(X)$, Lemma D.4 of (Alman & Song, 2024a)). *Let $k_1 = n^{o(1)}$. Let $k_2 = n^{o(1)}$. Assume that $p_1(X) := f(X) \odot q(X)$. Assume $U_1, V_1 \in \mathbb{R}^{n \times k_1}$ approximates the $f(X)$ such that $\|U_1 V_1^\top - f(X)\|_\infty \le \epsilon / \operatorname{poly}(n)$. Assume $U_2, V_2 \in \mathbb{R}^{n \times k_2}$ approximates the $q(X) \in \mathbb{R}^{n \times n}$ such that $\|U_2 V_2^\top - q(X)\|_\infty \le \epsilon / \operatorname{poly}(n)$. Then there are matrices $U_3, V_3 \in \mathbb{R}^{n \times k_3}$ such that $\|U_3 V_3^\top - p_1(X)\|_\infty \le \epsilon / \operatorname{poly}(n)$. The matrices $U_3, V_3$ can be explicitly constructed in $n^{1+o(1)}$ time.*

**Lemma C.16** (Low rank representation $p_2(X)$, Lemma D.5 of (Alman & Song, 2024a)). *Let $k_1 = n^{o(1)}$. Let $k_2 = n^{o(1)}$. Let $k_4 = n^{o(1)}$. Assume that $p_2(X)$ is an $n \times n$ where $j_0$-th row $p_2(X)_{j_0} = f(X)_{j_0} f(X)_{j_0}^\top q(X)_{j_0}$ for each $j_0 \in [n]$. Assume $U_1, V_1 \in \mathbb{R}^{n \times k_1}$ approximates the $f(X)$ such that $\|U_1 V_1^\top - f(X)\|_\infty \le \epsilon / \operatorname{poly}(n)$. Assume $U_2, V_2 \in \mathbb{R}^{n \times k_2}$ approximates the $q(X) \in \mathbb{R}^{n \times n}$ such that $\|U_2 V_2^\top - q(X)\|_\infty \le \epsilon / \operatorname{poly}(n)$. Then there are matrices $U_4, V_4 \in \mathbb{R}^{n \times k_4}$ such that $\|U_4 V_4^\top - p_2(X)\|_\infty \le \epsilon / \operatorname{poly}(n)$. The matrices $U_4, V_4$ can be explicitly constructed in $n^{1+o(1)}$ time.*

## C.5 BOUNDED ENTRIES OF MATRICES

In this section, we provide proof that entries of matrices are bounded.

We begin with the exponential matrix $f(X)$.

**Lemma C.17** (Bounded entries of $f(X)$). *If we have the below conditions,*

- *Let $f(X) \in \mathbb{R}^{n \times n}$ be defined in Definition C.8.*

*Then, we can show that*

$$\|f(X)\|_\infty \le 1$$

*Proof.* By Definition C.8, we have

$$f(X) = \operatorname{diag}(\alpha(X))^{-1} u(X)$$

By Definition C.7, we have

$$\alpha(X) = u(X)\mathbf{1}_n$$

Combining above two equations, we have

$$\|f(X)\|_\infty \leq 1$$

$\square$

A similar analysis can be applied to $h(X)$ and $s(X)$ as well.

**Lemma C.18** (Bounded entries of $h(X)$)**.** *If we have the below conditions,*

- *Let $X \in \mathbb{R}^{n \times d}, W, W_V \in \mathbb{R}^{d \times d}$ be defined in Definition C.3.*

- *Assuming each entry of $X, W, W_V$ can be re represented using $O(\log(n))$ bits.*

- *Let $h(X) \in \mathbb{R}^{n \times d}$ be defined in Definition C.9.*

*Then, we can show that*

$$\|h(X)\|_\infty \leq \mathrm{poly}(n)$$

*Proof.* By Definition C.9, we have

$$h(X) := XW_V$$

Then, we have

$$
\begin{aligned}
\|h(X)\|_\infty &= \|XW_V\|_\infty \\
&\leq n\|X\|_\infty \|W_V\|_\infty \\
&\leq \mathrm{poly}(n)
\end{aligned}
$$

where the 1st step is from the definition of $h(X)$, the 2nd step comes from basic linear algebra, the 3rd step is because of each entry in $X$ and $W_V$ can be represented by $O(\log(n))$ bits. $\square$

**Lemma C.19** (Bounded entries of $s(X)$)**.** *If we have the below conditions,*

- *Let $X \in \mathbb{R}^{n \times d}, W, W_V \in \mathbb{R}^{d \times d}$ be defined in Definition C.3.*

- *Assuming each entry of $X, W, W_V$ can be re represented using $O(\log(n))$ bits.*

- *Let $s(X) \in \mathbb{R}^{n \times d}$ be defined in Definition C.10.*

*Then, we can show that*

$$\|s(X)\|_\infty \leq \mathrm{poly}(n)$$

*Proof.* By Definition C.10, we have

$$s(X) = \underbrace{f(X)}_{n \times d}\underbrace{h(X)}_{n \times n}\underbrace{}_{n \times d}$$

Then, we have

$$
\begin{aligned}
\|s(X)\|_\infty &= \|f(X)h(X)\|_\infty \\
&\leq n\|f(X)\|_\infty \|h(X)\|_\infty \\
&\leq \mathrm{poly}(n)
\end{aligned}
$$

where the 1st step is from the definition of $c(X)$, the 2nd step comes from basic linear algebra, the 3rd step is because of Lemma C.17, C.18. $\square$

# D    MATRIX VIEW

In this section, we dive into analyzing the gradient of $\frac{\mathrm{d}L(X)}{\mathrm{d}T_{i-1}(X)}$.

In Section D.1, we give the gradient of $s(X)$ with respective to $X$. In Section D.2, we show the close form of the gradient on $T_i(X)$ via the chain rule. In Section D.3, we integrate each $C_i(X)$ to its corresponding matrix term $B_i(X)$. In Section D.4, applying the similar technique used in the previous section, we integrate the gradient on $T_i(X)$ into its corresponding matrix view. In Section D.5, we further apply matrix integration on each matrix term in the gradient on $T_i(X)$ calculated in the previous section. In Section D.6, we give the matrix view of all gradient components.

## D.1    GRADIENT OF $s(X)$

In this section, we give the gradient of $s(X)$ with respective to $X$.

The results from (Deng et al., 2023b) give the gradient of $c(X)$. By chain rule, the gradient of $s(X)$ is equivalent to the gradient of $c(X)$ from (Deng et al., 2023b), since $c(X) = s(X) - B$ where $B$ is a constant matrix.

**Lemma D.1** (Gradient of $s(X)_{i_0,j_0}$, Lemma B.16 in (Deng et al., 2023b))**.** *If we have the below conditions,*

- *Let $s(X) \in \mathbb{R}^{n \times d}$ be defined as Definition C.10*

*Then, we have*

- **Part 1.** *For all $i_0 = i_1 \in [n]$, $j_0, j_1 \in [d]$,*

$$\frac{\mathrm{d}s(X)_{i_0,j_0}}{\mathrm{d}X_{i_1,j_1}} = C_1(X) + C_2(X) + C_3(X) + C_4(X) + C_5(X)$$

   *where we have definitions:*

   – $C_1(X) := -s(X)_{i_0,j_0} \cdot f(X)_{i_0,i_0} \cdot \langle W_{j_1,*}, X_{i_0,*} \rangle$
   – $C_2(X) := -s(X)_{i_0,j_0} \cdot \langle f(X)_{i_0,*}, XW_{*,j_1} \rangle$
   – $C_3(X) := f(X)_{i_0,i_0} \cdot h(X)_{i_0,j_0} \cdot \langle W_{j_1,*}, X_{i_0,*} \rangle$
   – $C_4(X) := \langle f(X)_{i_0,*} \odot (XW_{*,j_1}), h(X)_{*,j_0} \rangle$
   – $C_5(X) := f(X)_{i_0,i_0} \cdot (W_V)_{j_1,j_0}$

- **Part 2.** *For all $i_0 \neq i_1 \in [n]$, $j_0, j_1 \in [d]$,*

$$\frac{\mathrm{d}s(X)_{i_0,j_0}}{\mathrm{d}X_{i_1,j_1}} = C_6(X) + C_7(X) + C_8(X)$$

   *where we have definitions:*

   – $C_6(X) := -s(X)_{i_0,j_0} \cdot f(X)_{i_1,i_0} \cdot \langle W_{j_1,*}, X_{i_0,*} \rangle$
     * *This is corresponding to $C_1(X)$*
   – $C_7(X) := f(X)_{i_1,i_0} \cdot h(X)_{i_1,j_0} \cdot \langle W_{j_1,*}, X_{i_0,*} \rangle$
     * *This is corresponding to $C_3(X)$*
   – $C_8(X) := f(X)_{i_1,i_0} \cdot (W_V)_{j_1,j_0}$
     * *This is corresponding to $C_5(X)$*

## D.2    GRADIENT ON $T_i(X)$

In the Lemma D.2, we use the chain rule to calculate the close form of the gradient on $T_i(X)$.

**Lemma D.2** (Gradient for $T_i(X)$)**.** *If we have the below conditions,*

- *Let $\mathrm{Attn}_i$ be defined as Definition C.3.*

- *Let $T_i(X) \in \mathbb{R}^{n \times d}$ be defined as Definition 2.3.*

- *Let $s(X)$ be defined as Definition C.10.*

- *Let $G_i \in \mathbb{R}^{n \times d}$ denote the gradient matrix resulting from the application of the chain rule up to the function $g_i$, i.e., $G_i = \frac{\mathrm{d}L(X)}{\mathrm{d}\mathsf{Attn}_i(T_{i-1}(X))}$.*

- *For $i_2 \in [n], j_2 \in [d]$, let $G_i(i_2, j_2)$ denote the $(i_2, j_2)$-th entry of $G_i$.*

*Then, we can show that, for $i_1 \in [n], j_1 \in [d]$, we have*

$$\frac{\mathrm{d}L(X)}{\mathrm{d}T_{i-1}(X)_{i_1,j_1}} = \sum_{i_0=1}^{n} \sum_{j_0=1}^{d} G_i(i_0, j_0) \cdot \frac{\mathrm{d}s(X)_{i_0,j_0}}{\mathrm{d}X_{i_1,j_1}}$$

*Proof.* By Lemma C.4, we have

$$\frac{\mathrm{d}L(X)}{\mathrm{d}T_{i-1}(X)} = \sum_{i_2=1}^{n} \sum_{j_2=1}^{d} G_i(i_2, j_2) \cdot \frac{\mathrm{d}\mathsf{Attn}_i(T_{i-1}(X))_{i_2,j_2}}{\mathrm{d}T_{i-1}(X)}.$$

By Definition C.3 and Definition C.10, we have

$$\mathsf{Attn}_i(T_{i-1}(X)) = s(T_{i-1}(X))$$

Therefore, by combining above two equations and substituting variable $T_{i-1}(X) = X$, we have

$$\frac{\mathrm{d}L(X)}{\mathrm{d}T_{i-1}(X)_{i_1,j_1}} = \sum_{i_0=1}^{n} \sum_{j_0=1}^{d} G_i(i_0, j_0) \cdot \frac{\mathrm{d}s(X)_{i_0,j_0}}{\mathrm{d}X_{i_1,j_1}}$$

$\square$

### D.3 MATRIX VIEW OF $C(X)$

In this section, we will provide the matrix view of $C_i(X) \in \mathbb{R}$, for $i \in \{6, 7, 8, 2, 4\}$. We will consider each $C_i(X)$ one by one. We begin with $C_6(X)$.

**Lemma D.3** (Matrix view of $C_6(X)$). *If we have the below conditions,*

- *Let $C_6(X, i_1, j_1) := -s(X)_{i_0,j_0} \cdot f(X)_{i_1,i_0} \cdot \langle W_{j_1,*}, X_{i_0,*} \rangle$ be defined as in Lemma D.1.*

- *We define a matrix $B_6(X) \in \mathbb{R}^{n \times d}$. For all $i_1 \in [n], j_1 \in [d]$, let $B_6(i_1, j_1)$ denote the $(i_1, j_1)$-th entry of $B_6(X)$. We define $B_6(i_1, j_1) = C_6(X, i_1, j_1)$.*

*Then, we can show that*

$$\underbrace{B_6(X)}_{n \times d} = \underbrace{-s(X)_{i_0,j_0}}_{1 \times 1} \underbrace{f(X)_{*,i_0}}_{n \times 1} \underbrace{(W \cdot X_{i_0,*})^\top}_{1 \times d}$$

*Proof.* We have

$$\begin{aligned}
C_6(X, i_1, j_1) &= -s(X)_{i_0,j_0} \cdot f(X)_{i_1,i_0} \cdot \langle W_{j_1,*}, X_{i_0,*} \rangle \\
&= -s(X)_{i_0,j_0} \cdot f(X)_{i_1,i_0} \cdot X_{i_0,*}^\top W_{j_1,*}
\end{aligned}$$

where the 1st step is from the choice of $C_6(X)$, the 2nd step comes from $\langle a, b \rangle = a^\top b$ holds for any $a, b \in \mathbb{R}^d$.

We have

$$\underbrace{B_6(X)(i_1, *)}_{d \times 1} = -\underbrace{s(X)_{i_0,j_0}}_{1 \times 1} \underbrace{f(X)_{i_1,i_0}}_{1 \times 1} \underbrace{W}_{d \times d} \underbrace{X_{i_0,*}}_{d \times 1}$$

Then, we have

$$\underbrace{B_6(X)}_{n \times d} = \underbrace{-s(X)_{i_0,j_0}}_{1 \times 1} \underbrace{f(X)_{*,i_0}}_{n \times 1} \underbrace{(W \cdot X_{i_0,*})^\top}_{1 \times d}$$

$\square$

A similar analysis procedure can also be applied on $C_7(X)$.

**Lemma D.4** (Matrix view of $C_7(X)$). *If we have the below conditions,*

- *Let $C_7(X, i_1, j_1) := f(X)_{i_1,i_0} \cdot h(X)_{j_0,i_1} \cdot \langle W_{j_1,*}, X_{i_0,*} \rangle$ be defined as in Lemma D.1.*

- *We define a matrix $B_7(X) \in \mathbb{R}^{n \times d}$. For all $i_1 \in [n], j_1 \in [d]$, let $B_7(i_1, j_1)$ denote the $(i_1, j_1)$-th entry of $B_7(X)$. We define $B_7(i_1, j_1) = C_7(X, i_1, j_1)$.*

*Then, we can show that*

$$\underbrace{B_7(X)}_{n \times d} = \underbrace{(f(X)_{*,i_0} \odot h(X)_{*,j_0})}_{n \times 1} \cdot \underbrace{(W \cdot X_{i_0,*})^\top}_{1 \times d}$$

*Proof.* We have

$$C_7(X, i_1, j_1) = f(X)_{i_1,i_0} \cdot h(X)_{i_1,j_0} \cdot \langle W_{j_1,*}, X_{i_0,*} \rangle$$
$$= f(X)_{i_1,i_0} \cdot h(X)_{i_1,j_0} \cdot W_{j_1,*}^\top X_{i_0,*}$$

where the 1st step is from the choice of $C_7(X)$, the 2nd step comes from $\langle a, b \rangle = a^\top b$ holds for any $a, b \in \mathbb{R}^d$.

We have

$$B_7(X)(i_1, *) = f(X)_{i_1,i_0} \cdot h(X)_{i_1,j_0} \cdot W \cdot X_{i_0,*}$$

Then, we have

$$\underbrace{B_7(X)}_{n \times d} = \underbrace{(f(X)_{*,i_0} \odot h(X)_{*,j_0})}_{n \times 1} \cdot \underbrace{(W \cdot X_{i_0,*})^\top}_{1 \times d}$$

$\square$

Then, we provide an analysis of $C_8(X)$.

**Lemma D.5** (Matrix view of $C_8(X)$). *If we have the below conditions,*

- *Let $C_8(X, i_1, j_1) := f(X)_{i_1,i_0} \cdot (W_V)_{j_1,j_0}$ be defined as in Lemma D.1.*

- *We define a matrix $B_8(X) \in \mathbb{R}^{n \times d}$. For all $i_1 \in [n], j_1 \in [d]$, let $B_8(i_1, j_1)$ denote the $(i_1, j_1)$-th entry of $B_8(X)$. We define $B_8(i_1, j_1) = C_8(X, i_1, j_1)$.*

*Then, we can show that*

$$\underbrace{B_8(X)}_{n \times d} = \underbrace{f(X)_{*,i_0}}_{n \times 1} \underbrace{(W_V)_{*,j_0}^\top}_{1 \times d}$$

*Proof.* We have

$$C_8(X, i_1, j_1) = f(X)_{i_1,i_0} \cdot (W_V)_{j_1,j_0}$$

where the 1st step is from the choice of $C_7(X)$.

We have

$$B_8(X)(i_1, *) = f(X)_{i_1,i_0} \cdot (W_V)_{*,j_0}$$

Then, we have

$$\underbrace{B_8(X)}_{n \times d} = \underbrace{f(X)_{*,i_0}}_{n \times 1} \underbrace{(W_V)_{*,j_0}^\top}_{1 \times d}$$

$\square$

Now, we consider $C_2(X)$.

**Lemma D.6** (Matrix view of $C_2(X)$). *If we have the below conditions,*

- *Let $C_2(X, j_1) := -s(X)_{i_0,j_0} \cdot \langle f(X)_{i_0,*}, XW_{*,j_1} \rangle$ be defined as in Lemma D.1.*

- *We define a matrix $B_2(X) \in \mathbb{R}^d$. For all $j_1 \in [d]$, the $j_1$-th entry of $B_2(X)$ is defined as $C_2(X, j_1)$.*

*Then, we can show that*

$$\underbrace{B_2(X)}_{d \times 1} = \underbrace{-s(X)_{i_0,j_0}}_{1 \times 1} \underbrace{W^\top}_{d \times d} \underbrace{X^\top}_{d \times n} \underbrace{f(X)_{i_0,*}}_{n \times 1}$$

*Proof.* We have

$$\begin{aligned} C_2(X, j_1) &= -s(X)_{i_0,j_0} \cdot \langle f(X)_{i_0,*}, XW_{*,j_1} \rangle \\ &= -s(X)_{i_0,j_0} \cdot (XW_{*,j_1})^\top f(X)_{i_0,*} \\ &= \underbrace{-s(X)_{i_0,j_0}}_{1 \times 1} \underbrace{W_{*,j_1}^\top}_{1 \times d} \underbrace{X^\top}_{d \times n} \underbrace{f(X)_{i_0,*}}_{n \times 1} \end{aligned}$$

where the 1st step is from the choice of $C_2(X)$, the second step follows from $\langle a, b \rangle = a^\top b$, for any $a, b \in \mathbb{R}^n$.

Then, we have

$$\underbrace{B_2(X)}_{d \times 1} = \underbrace{-s(X)_{i_0,j_0}}_{1 \times 1} \underbrace{W^\top}_{d \times d} \underbrace{X^\top}_{d \times n} \underbrace{f(X)_{i_0,*}}_{n \times 1}$$

$\square$

Finally, we analyze $C_4(X)$, which is the last term we need to compute.

**Lemma D.7** (Matrix view of $C_4(X)$). *If we have the below conditions,*

- *Let $C_4(X, j_1) := \langle f(X)_{i_0,*} \odot (XW_{*,j_1}), h(X)_{*,j_0} \rangle$ be defined as in Lemma D.1.*

- *We define a matrix $B_4(X) \in \mathbb{R}^d$. For all $j_1 \in [d]$, the $j_1$-th entry of $B_4(X)$ is defined as $C_4(X, j_1)$.*

*Then, we can show that*

$$\underbrace{B_4(X)}_{d \times 1} = \underbrace{W^\top}_{d \times d} \underbrace{X^\top}_{d \times n} \underbrace{(f(X)_{i_0,*} \odot h(X)_{*,j_0})}_{n \times 1}$$

*Proof.* We have

$$\begin{aligned} C_4(X, j_1) &= \langle f(X)_{i_0,*} \odot (XW_{*,j_1}), h(X)_{*,j_0} \rangle \\ &= \langle f(X)_{i_0,*} \odot h(X)_{*,j_0}, (XW_{*,j_1}) \rangle \\ &= (XW_{*,j_1})^\top (f(X)_{i_0,*} \odot h(X)_{*,j_0}) \end{aligned}$$

where the 1st step is from the choice of $C_4(X)$, the 2nd step comes from Fact C.1, and the last step follows from basic linear algebra. $\square$

### D.4 MATRIX VIEW OF GRADIENT ON $T_i(X)$

Since we have got the matrix view of each $C_i(X)$ term in the previous section, we can get the matrix view of the gradient on $T_i(X)$ in Lemma D.8.

**Lemma D.8** (Matrix view of single entry of gradient). *If we have the below conditions,*

- *Let $s(X)$ be defined as Definition C.10.*

- *Let $G_i \in \mathbb{R}^{n \times d}$ denote the gradient matrix resulting from the application of the chain rule up to the function $g_i$, i.e., $G_i = \frac{\mathrm{d}L(X)}{\mathrm{d}\mathsf{Attn}_i(T_{i-1}(X))}$.*

- *For $i_2 \in [n], j_2 \in [d]$, let $G_i(i_2, j_2)$ denote the $(i_2, j_2)$-th entry of $G_i$.*

- *Let $B_6(X), B_7(X), B_8(X) \in \mathbb{R}^{n \times d}$ be defined in Lemma D.3, Lemma D.4, and Lemma D.5*

- *Let $B_2(X), B_4(X) \in \mathbb{R}^d$ be defined in Lemma D.6 and Lemma D.7.*

*For any $i_0 \in [n], j_0 \in [d]$, we have*

$$G_i(i_0, j_0) \cdot \frac{\mathrm{d}s(X)_{i_0, j_0}}{\mathrm{d}X} = \underbrace{G_i(i_0, j_0)}_{1 \times 1} \cdot (\underbrace{B_6(X) + B_7(X) + B_8(X)}_{n \times d} + \underbrace{e_{i_0}}_{n \times 1} \underbrace{(B_2(X) + B_4(X))^\top}_{1 \times d})$$

*Proof.* By Lemma D.1, we have

- **Part 1.** For all $i_0 = i_1 \in [n], j_0, j_1 \in [d]$,

$$\frac{\mathrm{d}s(X)_{i_0, j_0}}{\mathrm{d}X_{i_1, j_1}} = C_1(X) + C_2(X) + C_3(X) + C_4(X) + C_5(X) \tag{1}$$

- **Part 2.** For all $i_0 \neq i_1 \in [n], j_0, j_1 \in [d]$,

$$\frac{\mathrm{d}s(X)_{i_0, j_0}}{\mathrm{d}X_{i_1, j_1}} = C_6(X) + C_7(X) + C_8(X) \tag{2}$$

Since for any $i_1 \in [n], j_1 \in [d]$, let $G_i(i_0, j_0) \cdot \frac{\mathrm{d}s(X)_{i_0, j_0}}{\mathrm{d}X_{i_1, j_1}}$ denote the $(i_1, j_1)$-th entry of $G_i(i_0, j_0) \cdot \frac{\mathrm{d}s(X)_{i_0, j_0}}{\mathrm{d}X}$, we consider the following two cases:

- **Case 1.** The $i_0$-th row of $G_i(i_0, j_0) \cdot \frac{\mathrm{d}s(X)_{i_0, j_0}}{\mathrm{d}X}$.

- **Case 2.** The other $n - 1$ rows of $G_i(i_0, j_0) \cdot \frac{\mathrm{d}s(X)_{i_0, j_0}}{\mathrm{d}X}$ where $i_1 \neq i_0$.

We first consider **Case 1.**

Recall that the matrix view of $C_2(X), C_4(X) \in \mathbb{R}$ are $B_2(X), B_4(X) \in \mathbb{R}^d$, and the matrix view of $C_6(X), C_7(X), C_8(X) \in \mathbb{R}$ are $B_6(X), B_7(X), B_8(X) \in \mathbb{R}^{n \times d}$, respectively.

For $k \in \{6, 7, 8\}$, we use $B_k(X)(s, *) \in \mathbb{R}^d$ to denote the $s$-th row of $B_k(X)$.

We use $(G_i(i_0, j_0) \cdot \frac{\mathrm{d}s(X)_{i_0, j_0}}{\mathrm{d}X})(i_0, *) \in \mathbb{R}^d$ to denote the $i_0$-th row of $G_i(i_0, j_0) \cdot \frac{\mathrm{d}s(X)_{i_0, j_0}}{\mathrm{d}X}$.

Since $C_6(X), C_7(X), C_8(X)$ are the corresponding parts of $C_1(X), C_3(X), C_5(X)$, and by Eq. (1), then we can have the following

$$(G_i(i_0, j_0) \cdot \frac{\mathrm{d}s(X)_{i_0, j_0}}{\mathrm{d}X})(i_0, *)$$
$$= \underbrace{G_i(i_0, j_0)}_{1 \times 1} \cdot \underbrace{(B_6(X)(i_0, *) + B_7(X)(i_0, *) + B_8(X)(i_0, *) + B_2(X) + B_4(X))}_{d \times 1}$$

We then consider **Case 2.**

For $k \in \{6, 7, 8\}$, we use $B_k(X)(\neq s, *) \in \mathbb{R}^{(n-1) \times d}$ to denote the matrix $B_k(X)$ with the $s$-th row removed.

Similarly, we use $(G_i(i_0, j_0) \cdot \frac{\mathrm{d}s(X)_{i_0, j_0}}{\mathrm{d}X})(\neq i_0, *) \in \mathbb{R}^{(n-1) \times d}$ to denote the matrix $G_i(i_0, j_0) \cdot \frac{\mathrm{d}s(X)_{i_0, j_0}}{\mathrm{d}X}$ with the $i_0$-th row removed.

By Eq. (2), we have

$$
(G_i(i_0, j_0) \cdot \frac{\mathrm{d}s(X)_{i_0,j_0}}{\mathrm{d}X})(\neq i_0, *) = \underbrace{G_i(i_0, j_0)}_{1 \times 1} \cdot \underbrace{(B_6(X)(\neq i_0, *) + B_7(X)(\neq i_0, *) + B_8(X)(\neq i_0, *))}_{d \times (n-1)}
$$

Combining **Case 1** and **Case 2** together, we have

$$
G_i(i_0, j_0) \cdot \frac{\mathrm{d}s(X)_{i_0,j_0}}{\mathrm{d}X} = \underbrace{G_i(i_0, j_0)}_{1 \times 1} \cdot \underbrace{(B_6(X) + B_7(X) + B_8(X)}_{n \times d} + \underbrace{e_{i_0}}_{n \times 1} \underbrace{(B_2(X) + B_4(X))^{\top}}_{1 \times d})
$$

$\square$

Then, we have the matrix view of $T_i(X)$ gradient.

**Lemma D.9** (Matrix view of $T_i(X)$ gradient). *If we have the below conditions,*

- *Let $L(X)$ be defined as Definition 2.1.*

- *Let $T(X)$ be defined as Definition 2.3.*

- *Let $G_i \in \mathbb{R}^{n \times d}$ denote the gradient matrix resulting from the application of the chain rule up to the function $g_i$, i.e., $G_i = \frac{\mathrm{d}L(X)}{\mathrm{d}\mathsf{Attn}_i(T_{i-1}(X))}$.*

- *For $i_2 \in [n], j_2 \in [d]$, let $G_i(i_2, j_2)$ denote the $(i_2, j_2)$-th entry of $G_i$.*

- *Let $B_6(X), B_7(X), B_8(X) \in \mathbb{R}^{n \times d}$ be defined in Lemma D.3, Lemma D.4, and Lemma D.5*

- *Let $B_2(X), B_4(X) \in \mathbb{R}^d$ be defined in Lemma D.6 and Lemma D.7.*

*Then, we have*

$$
\frac{\mathrm{d}L(X)}{\mathrm{d}T_{i-1}(X)} = \sum_{i_0=1}^{n} \sum_{j_0=1}^{d} \underbrace{G_i(i_0, j_0)}_{1 \times 1} \cdot \underbrace{(B_6(X) + B_7(X) + B_8(X)}_{n \times d} + \underbrace{e_{i_0}}_{n \times 1} \underbrace{(B_2(X) + B_4(X))^{\top}}_{1 \times d})
$$

*Proof.* By Lemma D.8, we have

$$
G_i(i_0, j_0) \cdot \frac{\mathrm{d}s(X)_{i_0,j_0}}{\mathrm{d}X} = \underbrace{G_i(i_0, j_0)}_{1 \times 1} \cdot \underbrace{(B_6(X) + B_7(X) + B_8(X)}_{n \times d} + \underbrace{e_{i_0}}_{n \times 1} \underbrace{(B_2(X) + B_4(X))^{\top}}_{1 \times d})
$$

Then, by Lemma C.4 we have

$$
\frac{\mathrm{d}L(X)}{\mathrm{d}T_{i-1}(X)} = \sum_{i_2=1}^{n} \sum_{j_2=1}^{d} G_i(i_2, j_2) \cdot \frac{\mathrm{d}\mathsf{Attn}_i(T_{i-1}(X))_{i_2,j_2}}{\mathrm{d}T_{i-1}(X)}.
$$

After combining the above two equations, we are done. $\square$

### D.5 MATRIX VIEW OF EACH TERM IN GRADIENT ON $T_i(X)$

In this subsection, we reduce the double summation to a matrix product for easy and clear analysis.

We first work on the $B_6$ term.

**Lemma D.10** (Matrix view of $B_6(X)$ term). *If we have the below conditions,*

- *Let $\underbrace{B_6(X)}_{n \times d} = \underbrace{-s(X)_{i_0,j_0}}_{1 \times 1} \underbrace{f(X)_{*,i_0}}_{n \times 1} \underbrace{(W \cdot X_{i_0,*})^{\top}}_{1 \times d}$ be defined in Lemma D.3.*

- *We define $z_6(X) \in \mathbb{R}^{n \times n}$, which satisfies*

$$\underbrace{z_6(X)_{*,i_0}}_{n \times 1} = \underbrace{(G_i(i_0, *)^\top}_{1 \times d} \underbrace{s(X)_{i_0, *})}_{d \times 1} \underbrace{f(X)_{*,i_0}}_{n \times 1}$$

- *Let $f(X) \in \mathbb{R}^{n \times n}$ be defined in Definition C.8.*

- *Let $W \in \mathbb{R}^{d \times d}$ be defined in Definition C.3.*

- *Let $G_i \in \mathbb{R}^{n \times d}$ denote the gradient matrix resulting from the application of the chain rule up to the function $g_i$, i.e., $G_i = \frac{\mathrm{d}L(X)}{\mathrm{d}\mathrm{Attn}_i(T_{i-1}(X))}$.*

- *For $i_2 \in [n], j_2 \in [d]$, let $G_i(i_2, j_2)$ denote the $(i_2, j_2)$-th entry of $G_i$.*

*Then we have*

$$\sum_{i_0=1}^{n} \sum_{j_0=1}^{d} \underbrace{G_i(i_0, j_0)}_{1 \times 1} \underbrace{B_6(X)}_{n \times d} = -\underbrace{z_6(X)}_{n \times n} \underbrace{X}_{n \times d} \underbrace{W^\top}_{d \times d}$$

*Proof.*

$$\sum_{i_0=1}^{n} \sum_{j_0=1}^{d} G_i(i_0, j_0) B_6(X) = -\sum_{i_0=1}^{n} \sum_{j_0=1}^{d} \underbrace{G_i(i_0, j_0)}_{1 \times 1} \underbrace{s(X)_{i_0,j_0}}_{1 \times 1} \underbrace{f(X)_{*,i_0}}_{n \times 1} \underbrace{(W \cdot X_{i_0,*})^\top}_{1 \times d}$$

$$= -\sum_{i_0=1}^{n} \left( \sum_{j_0=1}^{d} \underbrace{G_i(i_0, j_0)}_{1 \times 1} \underbrace{s(X)_{i_0,j_0}}_{1 \times 1} \right) \underbrace{f(X)_{*,i_0}}_{n \times 1} \underbrace{(W \cdot X_{i_0,*})^\top}_{1 \times d}$$

$$= -\sum_{i_0=1}^{n} \underbrace{(G_i(i_0, *)^\top}_{1 \times d} \underbrace{s(X)_{i_0,*})}_{d \times 1} \underbrace{f(X)_{*,i_0}}_{n \times 1} \underbrace{(W \cdot X_{i_0,*})^\top}_{1 \times d}$$

$$= -\sum_{i_0=1}^{n} \underbrace{(G_i(i_0, *)^\top}_{1 \times d} \underbrace{s(X)_{i_0,*})}_{d \times 1} \underbrace{f(X)_{*,i_0}}_{n \times 1} \underbrace{X_{i_0,*}^\top}_{1 \times d} \underbrace{W^\top}_{d \times d}$$

where the 1st step is from the choice of $B_6(X)$, the 2nd step comes from basic algebra, the 3rd step is because of $a^\top b = \sum_{i=1}^{d} a_i \cdot b_i$ holds for any $a, b \in \mathbb{R}^d$, the 4th step is due to $(AB)^\top = B^\top A^\top$ for any matrices $A$ and $B$.

Recall that we have $\underbrace{z_6(X)_{*,i_0}}_{n \times 1} = \underbrace{(G_i(i_0, *)^\top}_{1 \times d} \underbrace{s(X)_{i_0,*})}_{d \times 1} \underbrace{f(X)_{*,i_0}}_{n \times 1}$.

Then, we have

$$-\sum_{i_0=1}^{n} \underbrace{(G_i(i_0, *)^\top}_{1 \times d} \underbrace{s(X)_{i_0,*})}_{d \times 1} \underbrace{f(X)_{*,i_0}}_{n \times 1} \underbrace{X_{i_0,*}^\top}_{1 \times d} \underbrace{W^\top}_{d \times d} = -\sum_{i_0=1}^{n} \underbrace{z_6(X)_{*,i_0}}_{n \times 1} \underbrace{X_{i_0,*}^\top}_{1 \times d} \underbrace{W^\top}_{d \times d}$$

$$= -\underbrace{z_6(X)}_{n \times n} \underbrace{X}_{n \times d} \underbrace{W^\top}_{d \times d}$$

where the 1st step is from the choice of $z_6(X)$, the 2nd step comes from basic linear algebra. $\square$

Then, we can get the matrix view of $B_7(X)$ term.

**Lemma D.11** (Matrix view of $B_7(X)$ term)**.** *If we have the below conditions,*

- *Let $\underbrace{B_7(X)}_{n \times d} = \underbrace{(f(X)_{*,i_0} \odot h(X)_{*,j_0})}_{n \times 1} \cdot \underbrace{(W \cdot X_{i_0,*})^\top}_{1 \times d}$ be defined in Lemma D.4.*

- *We define $z_7(X) \in \mathbb{R}^{n \times n}$, which satisfies*

$$\underbrace{z_7(X)_{*,i_0}}_{n \times 1} = \underbrace{f(X)_{*,i_0}}_{n \times 1} \odot (\underbrace{h(X)}_{n \times d} \underbrace{G_i(i_0,*)}_{d \times 1}).$$

- *Let $X \in \mathbb{R}^{n \times d}, W \in \mathbb{R}^{d \times d}$ be defined in Definition C.3.*

- *Let $G_i \in \mathbb{R}^{n \times d}$ denote the gradient matrix resulting from the application of the chain rule up to the function $g_i$, i.e., $G_i = \frac{\mathrm{d}L(X)}{\mathrm{d}\mathrm{Attn}_i(T_{i-1}(X))}$.*

- *For $i_2 \in [n], j_2 \in [d]$, let $G_i(i_2, j_2)$ denote the $(i_2, j_2)$-th entry of $G_i$.*

*Then we have*

$$\sum_{i_0=1}^{n} \sum_{j_0=1}^{d} \underbrace{G_i(i_0,j_0)}_{1 \times 1} \underbrace{B_7(X)}_{n \times d} = \underbrace{z_7(X)}_{n \times n} \underbrace{X}_{n \times d} \underbrace{W^\top}_{d \times d}$$

*Proof.* We have

$$\sum_{i_0=1}^{n} \sum_{j_0=1}^{d} \underbrace{G_i(i_0,j_0)}_{1 \times 1} \underbrace{B_7(X)}_{n \times d} = \sum_{i_0=1}^{n} \sum_{j_0=1}^{d} \underbrace{G_i(i_0,j_0)}_{1 \times 1} \underbrace{(f(X)_{*,i_0} \odot h(X)_{*,j_0})}_{n \times 1} \cdot \underbrace{(W \cdot X_{i_0,*})^\top}_{1 \times d}$$

$$= \sum_{i_0=1}^{n} \underbrace{(f(X)_{*,i_0}}_{n \times 1} \odot (\sum_{j_0=1}^{d} \underbrace{G_i(i_0,j_0)}_{1 \times 1} \underbrace{h(X)_{*,j_0}}_{n \times 1})) \cdot \underbrace{(W \cdot X_{i_0,*})^\top}_{1 \times d}$$

$$= \sum_{i_0=1}^{n} \underbrace{(f(X)_{*,i_0}}_{n \times 1} \odot (\underbrace{h(X)}_{n \times d} \underbrace{G_i(i_0,*)}_{d \times 1})) \cdot \underbrace{(X_{i_0,*}^\top W^\top)}_{1 \times d}$$

where the 1st step is from the choice of $B_7(X)$, the 2nd step comes from basic algebra, the 3rd step is because of basic linear algebra.

Recall that we have $\underbrace{z_7(X)_{*,i_0}}_{n \times 1} = \underbrace{f(X)_{*,i_0}}_{n \times 1} \odot (\underbrace{h(X)}_{n \times d} \underbrace{G_i(i_0,*)}_{d \times 1})$.

Then we have

$$\sum_{i_0=1}^{n} \underbrace{(f(X)_{*,i_0}}_{n \times 1} \odot (\underbrace{h(X)}_{n \times d} \underbrace{G_i(i_0,*)}_{d \times 1})) \cdot \underbrace{(X_{i_0,*}^\top W^\top)}_{1 \times d}$$

$$= \sum_{i_0=1}^{n} \underbrace{z_7(X)_{*,i_0}}_{n \times 1} \underbrace{X_{i_0,*}^\top}_{1 \times d} \underbrace{W^\top}_{d \times d}$$

$$= \underbrace{z_7(X)}_{n \times n} \underbrace{X}_{n \times d} \underbrace{W^\top}_{d \times d}$$

where the 1st step is from the choice of $z_7(X)$, the 2nd step comes from basic linear algebra. $\square$

Then, we consider $B_8(X)$.

**Lemma D.12** (Matrix view of $B_8(X)$ term)**.** *If we have the below conditions,*

- *Let $\underbrace{B_8(X)}_{n \times d} = \underbrace{f(X)_{*,i_0}}_{n \times 1} \underbrace{(W_V)_{*,j_0}^\top}_{1 \times d}$ be defined in Lemma D.5.*

- *Let $G_i \in \mathbb{R}^{n \times d}$ denote the gradient matrix resulting from the application of the chain rule up to the function $g_i$, i.e., $G_i = \frac{\mathrm{d}L(X)}{\mathrm{d}\mathrm{Attn}_i(T_{i-1}(X))}$.*

- *For $i_2 \in [n], j_2 \in [d]$, let $G_i(i_2, j_2)$ denote the $(i_2, j_2)$-th entry of $G_i$.*

*Then we have*

$$\sum_{i_0=1}^{n} \sum_{j_0=1}^{d} \underbrace{G_i(i_0, j_0)}_{1 \times 1} \underbrace{B_8(X)}_{n \times d} = \underbrace{f(X)}_{n \times n} \underbrace{G_i}_{n \times d} \underbrace{W_V^\top}_{d \times d}$$

*Proof.* We have

$$\sum_{i_0=1}^{n} \sum_{j_0=1}^{d} \underbrace{G_i(i_0, j_0)}_{1 \times 1} \underbrace{B_8(X)}_{n \times d} = \sum_{i_0=1}^{n} \sum_{j_0=1}^{d} \underbrace{G_i(i_0, j_0)}_{1 \times 1} \underbrace{f(X)_{*,i_0}}_{n \times 1} \underbrace{(W_V)_{*,j_0}^\top}_{1 \times d}$$

$$= \sum_{i_0=1}^{n} \underbrace{f(X)_{*,i_0}}_{n \times 1} (\sum_{j_0=1}^{d} \underbrace{G_i(i_0, j_0)}_{1 \times 1} \underbrace{(W_V)_{*,j_0}^\top}_{1 \times d})$$

$$= \sum_{i_0=1}^{n} \underbrace{f(X)_{*,i_0}}_{n \times 1} \underbrace{G_i(i_0, *)^\top}_{1 \times d} \underbrace{W_V^\top}_{d \times d}$$

$$= \underbrace{f(X)}_{n \times n} \underbrace{G_i}_{n \times d} \underbrace{W_V^\top}_{d \times d}$$

where the 1st step is from the choice of $B_8(X)$, the 2nd step comes from basic algebra, the 3rd step is because of basic linear algebra, the 4th step is due to basic linear algebra.

$\square$

Now, we can do the matrix view of $B_2(X)$ term.

**Lemma D.13** (Matrix view of $B_2(X)$ term). *If we have the below conditions,*

- *Let $\underbrace{B_2(X)}_{d \times 1} = -\underbrace{s(X)_{i_0,j_0}}_{1 \times 1} \underbrace{W^\top}_{d \times d} \underbrace{X^\top}_{d \times n} \underbrace{f(X)_{i_0,*}}_{n \times 1}$ be defined in Lemma D.6*

- *Let $G_i \in \mathbb{R}^{n \times d}$ denote the gradient matrix resulting from the application of the chain rule up to the function $g_i$, i.e., $G_i = \frac{\mathrm{d}L(X)}{\mathrm{d}\mathsf{Attn}_i(T_{i-1}(X))}$.*

- *For $i_2 \in [n], j_2 \in [d]$, let $G_i(i_2, j_2)$ denote the $(i_2, j_2)$-th entry of $G_i$.*

- *We define $z_2(X) \in \mathbb{R}^{n \times n}$, which satisfies*

$$\underbrace{z_2(X)_{i_0,*}}_{n \times 1} = (\underbrace{G_i(i_0, *)^\top}_{1 \times d} \underbrace{s(X)_{i_0,*}}_{d \times 1}) \underbrace{f(X)_{i_0,*}}_{n \times 1}$$

- *Let $X \in \mathbb{R}^{n \times d}, W \in \mathbb{R}^{d \times d}$ be defined in Definition C.3*

*Then we have*

$$\sum_{i_0=1}^{n} \sum_{j_0=1}^{d} \underbrace{G_i(i_0, j_0)}_{1 \times 1} \underbrace{e_{i_0}}_{n \times 1} \underbrace{B_2(X)^\top}_{1 \times d} = -\underbrace{z_2(X)}_{n \times n} \underbrace{X}_{n \times d} \underbrace{W}_{d \times d}$$

*Proof.* We have

$$\sum_{i_0=1}^{n} \sum_{j_0=1}^{d} \underbrace{G_i(i_0, j_0)}_{1 \times 1} \underbrace{e_{i_0}}_{n \times 1} \underbrace{B_2(X)^\top}_{1 \times d} = -\sum_{i_0=1}^{n} \sum_{j_0=1}^{d} \underbrace{G_i(i_0, j_0)}_{1 \times 1} \underbrace{s(X)_{i_0,j_0}}_{1 \times 1} \underbrace{e_{i_0}}_{n \times 1} \underbrace{f(X)_{i_0,*}^\top}_{1 \times n} \underbrace{X}_{n \times d} \underbrace{W}_{d \times d}$$

$$= -\sum_{i_0=1}^{n}(\sum_{j_0=1}^{d}\underbrace{G_i(i_0,j_0)}_{1\times1}\underbrace{s(X)_{i_0,j_0}}_{1\times1})\underbrace{e_{i_0}}_{n\times1}\underbrace{f(X)_{i_0,*}^{\top}}_{1\times n}\underbrace{X}_{n\times d}\underbrace{W}_{d\times d}$$

$$= -\sum_{i_0=1}^{n}\underbrace{(G_i(i_0,*)^{\top}}_{1\times d}\underbrace{s(X)_{i_0,*}}_{d\times1})\underbrace{e_{i_0}}_{n\times1}\underbrace{f(X)_{i_0,*}^{\top}}_{1\times n}\underbrace{X}_{n\times d}\underbrace{W}_{d\times d}$$

$$= -\sum_{i_0=1}^{n}\underbrace{e_{i_0}}_{n\times1}\underbrace{(G_i(i_0,*)^{\top}}_{1\times d}\underbrace{s(X)_{i_0,*}}_{d\times1})\underbrace{f(X)_{i_0,*}^{\top}}_{1\times n}\underbrace{X}_{n\times d}\underbrace{W}_{d\times d}$$

where the 1st step is from the choice of $B_2(X)$, the 2nd step comes from basic algebra, the 3rd step is because of $a^{\top}b = \sum_{i=1}^{d} a_i \cdot b_i$ holds for any $a, b \in \mathbb{R}^d$, the 4th step is due to $(AB)^{\top} = B^{\top}A^{\top}$ holds for any matrix $A, B$.

Recall that we have $\underbrace{z_2(X)_{i_0,*}}_{n\times1} = \underbrace{(G_i(i_0,*)^{\top}}_{1\times d}\underbrace{s(X)_{i_0,*})}_{d\times1}\underbrace{f(X)_{i_0,*}}_{n\times1}.$

Then, we have

$$-\sum_{i_0=1}^{n}\underbrace{e_{i_0}}_{n\times1}\underbrace{(G_i(i_0,*)^{\top}}_{1\times d}\underbrace{s(X)_{i_0,*})}_{d\times1}\underbrace{f(X)_{i_0,*}^{\top}}_{1\times n}\underbrace{X}_{n\times d}\underbrace{W}_{d\times d} = -\sum_{i_0=1}^{n}\underbrace{e_{i_0}}_{n\times1}\underbrace{z_2(X)_{i_0,*}^{\top}}_{1\times n}\underbrace{X}_{n\times d}\underbrace{W}_{d\times d}$$

$$= -\underbrace{z_2(X)}_{n\times n}\underbrace{X}_{n\times d}\underbrace{W}_{d\times d}$$

where the 1st step is from the choice of $z_2(X)$, the 2nd step comes from basic linear algebra. $\square$

Finally, we do a similar analysis for the term $B_4(X)$. Then, we get all the matrix views we need.

**Lemma D.14** (Matrix view of $B_4(X)$ term). *If we have the below conditions,*

- *Let* $\underbrace{B_4(X)}_{d\times1} = \underbrace{W^{\top}}_{d\times d}\underbrace{X^{\top}}_{d\times n}\underbrace{(f(X)_{i_0,*} \odot h(X)_{*,j_0})}_{n\times1}$ *be defined in Lemma D.7.*

- *Let* $G_i \in \mathbb{R}^{n\times d}$ *denote the gradient matrix resulting from the application of the chain rule up to the function* $g_i$, *i.e.,* $G_i = \frac{dL(X)}{d\mathsf{Attn}_i(T_{i-1}(X))}$.

- *For* $i_2 \in [n], j_2 \in [d]$, *let* $G_i(i_2, j_2)$ *denote the* $(i_2, j_2)$*-th entry of* $G_i$.

- *We define* $z_4(X) \in \mathbb{R}^{n\times n}$, *which satisfies*

$$\underbrace{z_4(X)_{i_0,*}}_{n\times1} = \underbrace{f(X)_{i_0,*}}_{n\times1} \odot \underbrace{(h(X)G_i(i_0,*))}_{n\times1}$$

*Then we have*

$$\sum_{i_0=1}^{n}\sum_{j_0=1}^{d}\underbrace{G_i(i_0,j_0)}_{1\times1}\underbrace{e_{i_0}}_{n\times1}\underbrace{B_4(X)^{\top}}_{1\times d} = \underbrace{z_4(X)}_{n\times n}\underbrace{X}_{n\times d}\underbrace{W}_{d\times d}$$

*Proof.* We have

$$\sum_{i_0=1}^{n}\sum_{j_0=1}^{d}\underbrace{G_i(i_0,j_0)}_{1\times1}\underbrace{e_{i_0}}_{n\times1}\underbrace{B_4(X)^{\top}}_{1\times d} = \sum_{i_0=1}^{n}\sum_{j_0=1}^{d}\underbrace{G_i(i_0,j_0)}_{1\times1}\underbrace{e_{i_0}}_{n\times1}\underbrace{(f(X)_{i_0,*}^{\top} \odot h(X)_{*,j_0}^{\top})}_{1\times n}\underbrace{X}_{n\times d}\underbrace{W}_{d\times d}$$

$$= \sum_{i_0=1}^{n}\underbrace{e_{i_0}}_{n\times1}\underbrace{(f(X)_{i_0,*}^{\top}}_{1\times n} \odot (\sum_{j_0=1}^{d}\underbrace{G_i(i_0,j_0)}_{1\times1}\underbrace{h(X)_{*,j_0}^{\top}))}_{1\times n}\underbrace{X}_{n\times d}\underbrace{W}_{d\times d}$$

$$= \sum_{i_0=1}^{n} \underbrace{e_{i_0}}_{n \times 1} \underbrace{(f(X)_{i_0,*}^{\top}}_{1 \times n} \odot \underbrace{(h(X)G_i(i_0,*))^{\top})}_{1 \times n} \underbrace{X}_{n \times d} \underbrace{W}_{d \times d}$$

$$= \sum_{i_0=1}^{n} \underbrace{e_{i_0}}_{n \times 1} \underbrace{z_4(X)_{i_0,*}^{\top}}_{1 \times n} \underbrace{X}_{n \times d} \underbrace{W}_{d \times d}$$

$$= \underbrace{z_4(X)}_{n \times n} \underbrace{X}_{n \times d} \underbrace{W}_{d \times d}$$

where the 1st step is from the choice of $B_4(X)$, the 2nd step comes from basic algebra, the 3rd step is because of basic linear algebra, the 4th step is due to the choice of $z_4(X)$, the 5th step follows from basic linear algebra. $\qquad\square$

## D.6 COMPONENTS OF GRADIENT ON $T_i(X)$

**Definition D.15** (Definition of $D_k$). *If we have the below conditions,*

- *For $k_1 \in \{6, 7, 8\}$, let $B_{k_1}(X) \in \mathbb{R}^{n \times d}$ be defined as Lemma D.3, D.4, and D.5, respectively.*

- *For $k_2 \in \{2, 4\}$, let $B_{k_2}(X) \in \mathbb{R}^{d \times 1}$ be defined as Lemma D.6 and D.7, respectively.*

- *Let $G_i \in \mathbb{R}^{n \times d}$ denote the gradient matrix resulting from the application of the chain rule up to the function $g_i$, i.e., $G_i = \frac{\mathrm{d}L(X)}{\mathrm{d}\mathsf{Attn}_i(T_{i-1}(X))}$.*

*We define $D_k \in \mathbb{R}^{n \times d}$ as follows:*

- *For $k_1 \in \{6, 7, 8\}$, we define*

$$D_{k_1} := \sum_{i_0=1}^{n} \sum_{j_0=1}^{d} \underbrace{G_i(i_0, j_0)}_{1 \times 1} \underbrace{B_{k_1}(X)}_{n \times d}$$

- *For $k_2 \in \{2, 4\}$, we define*

$$D_{k_2} := \sum_{i_0=1}^{n} \sum_{j_0=1}^{d} \underbrace{G_i(i_0, j_0)}_{1 \times 1} \underbrace{e_{i_0}}_{n \times 1} \underbrace{B_{k_2}(X)^{\top}}_{1 \times d}$$

**Definition D.16** (Definition of $K$). *If we have the below conditions,*

- *Let $s(X) \in \mathbb{R}^{n \times d}$ be defined as Definition C.10.*

- *Let $G_i \in \mathbb{R}^{n \times d}$ denote the gradient matrix resulting from the application of the chain rule up to the function $g_i$, i.e., $G_i = \frac{\mathrm{d}L(X)}{\mathrm{d}\mathsf{Attn}_i(T_{i-1}(X))}$.*

*We define $K \in \mathbb{R}^n$, where for each $i_0 \in [n]$, we define*

$$\underbrace{K_{i_0}}_{1 \times 1} = \underbrace{G_i(i_0, *)^{\top}}_{1 \times d} \underbrace{s(X)_{i_0,*}}_{d \times 1}$$

*Furthermore, we have*

$$\underbrace{K}_{n \times 1} = \underbrace{(G_i \odot s(X))}_{n \times d} \underbrace{\mathbf{1}_d}_{d \times 1}$$

**Lemma D.17** (Close form of $D_k$). *If we have the below conditions,*

- *Let $X \in \mathbb{R}^{n \times d}, W \in \mathbb{R}^{d \times d}$ be defined as Definition C.3.*

- *For $k \in \{6, 7, 8, 2, 4\}$, let $D_k \in \mathbb{R}^{n \times d}$ be defined as Definition D.15.*

- *For $k_3 \in \{6, 7, 2, 4\}$, let $z_{k_3}(X) \in \mathbb{R}^{n \times n}$ be defined as Lemma D.10, D.11, D.13, and D.14, respectively.*

- *Let $K \in \mathbb{R}^n$ be defined as Definition D.16.*

- *We define $z_6(X) \in \mathbb{R}^{n \times n}$, which satisfies*

$$
\underbrace{z_6(X)}_{n \times n} = \underbrace{f(X)}_{n \times n} \underbrace{\mathrm{diag}(K)}_{n \times n}.
$$

- *We define $z_7(X) \in \mathbb{R}^{n \times n}$, which satisfies*

$$
\underbrace{z_7(X)}_{n \times n} = \underbrace{f(X)}_{n \times n} \odot (\underbrace{h(X)}_{n \times d} \underbrace{G_i^\top}_{d \times n})
$$

- *We define $z_2(X) \in \mathbb{R}^{n \times n}$, which satisfies*

$$
\underbrace{z_2(X)}_{n \times n} = \underbrace{\mathrm{diag}(K)}_{n \times n} \underbrace{f(X)}_{n \times n}
$$

- *We define $z_4(X) \in \mathbb{R}^{n \times n}$, which satisfies*

$$
\underbrace{z_4(X)}_{n \times n} = \underbrace{f(X)}_{n \times n} \odot (\underbrace{G_i}_{n \times d} \underbrace{h(X)^\top}_{d \times n})
$$

*Then, we can show that the close forms of $D_k$ can be written as follows:*

- $D_6 = - \underbrace{z_6(X)}_{n \times n} \underbrace{X}_{n \times d} \underbrace{W^\top}_{d \times d}.$

- $D_7 = \underbrace{z_7(X)}_{n \times n} \underbrace{X}_{n \times d} \underbrace{W^\top}_{d \times d}.$

- $D_8 = \underbrace{f(X)}_{n \times n} \underbrace{G_i}_{n \times d} \underbrace{W_V^\top}_{d \times d}.$

- $D_2 = - \underbrace{z_2(X)}_{n \times n} \underbrace{X}_{n \times d} \underbrace{W}_{d \times d}.$

- $D_4 = \underbrace{z_4(X)}_{n \times n} \underbrace{X}_{n \times d} \underbrace{W}_{d \times d}.$

*Proof.* We finish the proof by parts.

- By Lemma D.10, we have the close form of $D_6$.

- By Lemma D.11, we have the close form of $D_7$.

- By Lemma D.12, we have the close form of $D_8$.

- By Lemma D.13, we have the close form of $D_2$.

- By Lemma D.14, we have the close form of $D_4$.

$\square$

# E  FAST COMPUTATION FOR GRADIENT ON $T(X)$

In this section, we give an almost linear time $n^{1+o(1)}$ algorithm for each $B_i(X)$ term. Namely, we consider $B_6(X), B_7(X), B_8(X), B_2(X), B_4(X)$ in Section E.1, E.2, E.3, E.4, and E.5, respectively.

## E.1  FAST COMPUTATION FOR $B_6(X)$ TERM

Before we introduce the almost linear time algorithm for $B_6(X)$ term, we need to introduce the accelerated algorithm for the key component term, $z_6(X)$, in Lemma E.2.

We first compute $K$, which is defined in Definition D.16

**Lemma E.1** (Computation time for $K$). *If we have the below conditions,*

- *Let $K \in \mathbb{R}^n$ be defined as Definition D.16.*

*Then, we can show that $K$ can be computed in $O(n \cdot d)$ time.*

*Proof.* Since for each $i_0 \in [n]$, we have

$$\underbrace{K_{i_0}}_{1 \times 1} = \underbrace{G_i(i_0, *)^\top}_{1 \times d} \underbrace{s(X)_{i_0,*}}_{d \times 1}$$

Then, we have that it takes $O(d)$ time for calculating each entry.

Since there are total $n$ entries in $K$, the overall computation time for $K$ is $O(n \cdot d)$. $\square$

We now compute $z_6(X)$.

**Lemma E.2** (Fast computation for $z_6(X)$). *If we have the below conditions,*

- *Let $X \in \mathbb{R}^{n \times d}, W, W_V \in \mathbb{R}^{d \times d}$ be defined in Definition C.3.*

- *Let $G_i \in \mathbb{R}^{n \times d}$ denote the gradient matrix resulting from the application of the chain rule up to the function $g_i$, i.e., $G_i = \frac{dL(X)}{d\mathsf{Attn}_i(T_{i-1}(X))}$.*

- *Assuming each entry of $X, W, W_V, G_i$ can be re represented using $O(\log(n))$ bits.*

- *Let $z_6(X) \in \mathbb{R}^{n \times n}$ be defined in Lemma D.10.*

*Then, for some $k_6 = n^{o(1)}$, there are matrices $U_6, V_6 \in \mathbb{R}^{n \times k_6}$ such that $\|U_6 V_6^\top - z_6(X)\|_\infty \leq \epsilon/\operatorname{poly}(n)$. The matrices $U_6, V_6$ can be constructed in $n^{1+o(1)}$ time.*

*Proof.* Recall in Lemma D.10, we have define $z_6(X)$ satisfying the following equation

$$\underbrace{z_6(X)_{*,i_0}}_{n \times 1} = \underbrace{(G_i(i_0, *)^\top}_{1 \times d} \underbrace{s(X)_{i_0,*})}_{d \times 1} \underbrace{f(X)_{*,i_0}}_{n \times 1} \tag{3}$$

Recall that $K \in \mathbb{R}^n$ has been defined in Definition D.16. By Lemma E.1, we have $K$ can be computed in $O(n \cdot d)$ time.

We also have

$$\underbrace{z_6(X)}_{n \times n} = \underbrace{f(X)}_{n \times n} \underbrace{\operatorname{diag}(K)}_{n \times n}$$

By Lemma C.13, we have $U_1, V_1 \in \mathbb{R}^{n \times k_1}$ such that

$$\|U_1 V_1^\top - f(X)\|_\infty \leq \epsilon/\operatorname{poly}(n)$$

Let $U_6 = U_1$, $V_6 = \text{diag}(K)V_1$.

We have $V_6 = \underbrace{\text{diag}(K)}_{n \times n} \underbrace{V_1}_{n \times k_1}$ can be computed in $nk_1$ time.

The overall running time for constructing $U_6$ and $V_6$ is $n^{1+o(1)}$.

Then, we consider the error bound.

We have

$$
\begin{aligned}
\|U_6 V_6^\top - z_6(X)\|_\infty &= \|U_1 V_1^\top \text{diag}(K) - f(X) \text{diag}(K)\|_\infty \\
&\leq n\|U_1 V_1^\top - f(X)\|_\infty \|\text{diag}(K)\|_\infty \\
&\leq n(\epsilon/\text{poly}(n))\|\text{diag}(K)\|_\infty \\
&\leq \epsilon/\text{poly}(n)
\end{aligned}
$$

where the 1st step is from the choice of $U_6$, $V_6$, the 2nd step comes from basic linear algebra, the 3rd step is because of Lemma C.13, the 4th step is due to $\|\text{diag}(K)\|_\infty \leq \text{poly}(n)$.

$\square$

Then, we are ready to introduce the almost linear time algorithm for $B_6(X)$ term.

**Lemma E.3** (Fast computation for $B_6(X)$ term). *If we have the below conditions,*

- *Let $X \in \mathbb{R}^{n \times d}$, $W, W_V \in \mathbb{R}^{d \times d}$ be defined in Definition C.3.*

- *Assuming each entry of $X, W, W_V, G_i$ can be re represented using $O(\log(n))$ bits.*

- *Let $B_6(X) \in \mathbb{R}^{n \times n}$ be defined in Lemma D.3.*

- *We define $D_6 \in \mathbb{R}^{n \times d}$, where $D_6 := \sum_{i_0=1}^n \sum_{j_0=1}^d G_i(i_0, j_0) B_6(X)$.*

- *Let $G_i \in \mathbb{R}^{n \times d}$ denote the gradient matrix resulting from the application of the chain rule up to the function $g_i$, i.e., $G_i = \frac{\mathrm{d}L(X)}{\mathrm{d}\text{Attn}_i(T_{i-1}(X))}$.*

- *For $i_2 \in [n], j_2 \in [d]$, let $G_i(i_2, j_2)$ denote the $(i_2, j_2)$-th entry of $G_i$.*

*Then, we can show that, there is an algorithm to approximate $D_6$ in $n^{1+o(1)}$ time, and it can achieve $\epsilon/\text{poly}(n)$ accuracy.*

*Namely, the algorithm output $\widetilde{D}_6$ satisfying*

$$
\|D_6 - \widetilde{D}_6\|_\infty \leq \epsilon/\text{poly}(n)
$$

*Proof.* Recall that in Lemma D.10, we have defined $z_6(X) \in \mathbb{R}^{n \times n}$, which satisfies

$$
\underbrace{z_6(X)_{*,i_0}}_{n \times 1} = \underbrace{(G_i(i_0, *))^\top}_{1 \times d} \underbrace{s(X)_{i_0,*}}_{d \times 1} \underbrace{f(X)_{*,i_0}}_{n \times 1}
$$

And, in that Lemma, we also have

$$
\sum_{i_0=1}^n \sum_{j_0=1}^d \underbrace{G_i(i_0, j_0)}_{1 \times 1} \underbrace{B_6(X)}_{n \times d} = -\underbrace{z_6(X)}_{n \times n} \underbrace{X}_{n \times d} \underbrace{W^\top}_{d \times d}
$$

Let $U_6, V_6 \in \mathbb{R}^{n \times k_6}$ be defined as Lemma E.2.

Let $\widetilde{z}_6(X) = U_6 V_6^\top$.

By Lemma E.2, we have

$$
\|\widetilde{z}_6(X) - z_6(X)\|_\infty \leq \epsilon/\text{poly}(n) \tag{4}
$$

**Proof of running time.**

We compute in the following way:

- Compute $\underbrace{V_6^\top}_{k_6 \times n} \underbrace{X}_{n \times d}$, which takes $n^{1+o(1)}$ time.

- Compute $\underbrace{V_6^\top X}_{k_6 \times d} \underbrace{W^\top}_{d \times d}$, which takes $n^{1+o(1)}$ time.

- Compute $\underbrace{U_6}_{n \times k_6} \underbrace{V_6^\top X W^\top}_{k_6 \times d}$, which takes $n^{1+o(1)}$ time.

Therefore, the overall running time is $n^{1+o(1)}$.

**Proof of error bound.**

We have

$$
\begin{aligned}
\|\widetilde{z}_6(X) X W^\top - z_6(X) X W^\top\|_\infty &\le d \cdot n \|\widetilde{z}_6(X) - z_6(X)\|_\infty \|X\|_\infty \|W\|_\infty \\
&\le d \cdot n (\epsilon / \operatorname{poly}(n)) \|X\|_\infty \|W\|_\infty \\
&\le \epsilon / \operatorname{poly}(n)
\end{aligned}
$$

where the 1st step is from basic linear algebra, the 2nd step comes from Eq.(4), the 3rd step is because of $\|W\|_\infty \le \operatorname{poly}(n)$ and $\|X\|_\infty \le \operatorname{poly}(n)$.

$\square$

### E.2 Fast computation for $B_7(X)$ term

Similar to the analysis process of $B_6(X)$ term, we first provide the almost linear time algorithm for $z_7(X)$, then provide that algorithm for $B_7(X)$.

**Lemma E.4** (Fast computation for $z_7(X)$). *If we have the below conditions,*

- *Let $z_7(X) \in \mathbb{R}^{n \times n}$ be defined in Lemma D.11.*

- *By Lemma C.13, let $U_1, V_1$ be the low rank approximation of $f(X)$, such that $\|U_1 V_1^\top - f(X)\|_\infty \le \epsilon / \operatorname{poly}(n)$.*

- *Let $X \in \mathbb{R}^{n \times d}, W, W_V \in \mathbb{R}^{d \times d}$ be defined in Definition C.3.*

- *Assuming each entry of $X, W, W_V, G_i$ can be re represented using $O(\log(n))$ bits.*

- *Let $G_i \in \mathbb{R}^{n \times d}$ denote the gradient matrix resulting from the application of the chain rule up to the function $g_i$, i.e., $G_i = \frac{dL(X)}{d\operatorname{Attn}_i(T_{i-1}(X))}$.*

- *For $i_2 \in [n], j_2 \in [d]$, let $G_i(i_2, j_2)$ denote the $(i_2, j_2)$-th entry of $G_i$.*

*Then, for some $k_7 = n^{o(1)}$, there are matrices $U_7, V_7 \in \mathbb{R}^{n \times k_7}$ such that $\|U_7 V_7^\top - z_7(X)\|_\infty \le \epsilon / \operatorname{poly}(n)$. The matrices $U_7, V_7$ can be constructed in $n^{1+o(1)}$ time.*

*Proof.* Recall that in Lemma D.11, we have defined $z_7(X) \in \mathbb{R}^{n \times n}$, where the $i_0$-th column of $z_7(X)$ satisfies

$$
\underbrace{z_7(X)_{*, i_0}}_{n \times 1} = \underbrace{f(X)_{*, i_0}}_{n \times 1} \odot (\underbrace{h(X)}_{n \times d} \underbrace{G_i(i_0, *)}_{d \times 1})
$$

which is equivalent to

$$
\underbrace{z_7(X)}_{n \times n} = \underbrace{f(X)}_{n \times n} \odot (\underbrace{h(X)}_{n \times d} \underbrace{G_i^\top}_{d \times n})
$$

By Lemma C.13, we know $\widetilde{f}(X) := U_1 V_1^\top$ is a good approximation for $f(X)$.

We choose $U_7 = U_1 \oslash h(X)$ and $V_7 = V_1 \oslash G_i$, where $U_7, V_7 \in \mathbb{R}^{n \times k_1 d}$.

**Proof of running time.**

For $U_7 = U_1 \oslash h(X)$, since $U_1 \in \mathbb{R}^{n \times k_1}, h(X) \in \mathbb{R}^{n \times d}$, constructing $U_7$ takes $O(ndk_1) = O(n^{1+o(1)})$ time.

Similarly, constructing $V_7$ takes $O(n^{1+o(1)})$ time.

**Proof of error bound.**

Using Fact C.2, we have

$$
\begin{aligned}
\|U_7 V_7^\top - z_7(X)\|_\infty &= \|U_7 V_7^\top - f(X) \odot (h(X) G_i^\top)\|_\infty \\
&= \|(U_1 \oslash h(X))(V_1 \oslash G_i)^\top - f(X) \odot (h(X) G_i^\top)\|_\infty \\
&= \|(U_1 V_1^\top) \odot (h(X) G_i^\top) - f(X) \odot (h(X) G_i^\top)\|_\infty \\
&= \|\widetilde{f}(X) \odot (h(X) G_i^\top) - f(X) \odot (h(X) G_i^\top)\|_\infty \\
&\leq d \|h(X)\|_\infty \|G_i\|_\infty \cdot \epsilon / \operatorname{poly}(n) \\
&\leq \epsilon / \operatorname{poly}(n)
\end{aligned}
\tag{5}
$$

where the 1st step is from the definition of $z_7(X)$, the 2nd step comes from the choice of $U_7$ and $V_7$, the 3rd step is because of Fact C.2, the 4th step is due to the definition of $\widetilde{f}(X)$, the 5th step follows from $\|\widetilde{f}(X) - f(X)\|_\infty \leq \epsilon / \operatorname{poly}(n)$, the sixth step follows from Lemma C.18 and $\|G_i\|_\infty \leq \operatorname{poly}(n)$.

$\square$

Then, we can do similarly fast computation for $B_7$ term.

**Lemma E.5** (Fast computation for $B_7(X)$ term)**.** *If we have the below conditions,*

- *Let $B_7(X) \in \mathbb{R}^{n \times d}$ be defined in Lemma D.4.*

- *We define $D_7 \in \mathbb{R}^{n \times d}$, where $D_7 := \sum_{i_0=1}^n \sum_{j_0=1}^d G_i(i_0, j_0) B_7(X)$.*

- *Let $X \in \mathbb{R}^{n \times d}, W, W_V \in \mathbb{R}^{d \times d}, B \in \mathbb{R}^{n \times d}$ be defined in Definition C.3.*

- *Assuming each entry of $X, W, W_V, G_i$ can be re represented using $O(\log(n))$ bits.*

- *Let $G_i \in \mathbb{R}^{n \times d}$ denote the gradient matrix resulting from the application of the chain rule up to the function $g_i$, i.e., $G_i = \frac{\mathrm{d}L(X)}{\mathrm{d}\mathsf{Attn}_i(T_{i-1}(X))}$.*

- *For $i_2 \in [n], j_2 \in [d]$, let $G_i(i_2, j_2)$ denote the $(i_2, j_2)$-th entry of $G_i$.*

*Then, we can show that, there is an algorithm to approximate $D_7$ in $n^{1+o(1)}$ time, and it can achieve $\epsilon / \operatorname{poly}(n)$ accuracy.*

*Namely, the algorithm output $\widetilde{D}_7$ satisfies*

$$
\|D_7 - \widetilde{D}_7\|_\infty \leq \epsilon / \operatorname{poly}(n)
$$

*Proof.* In Lemma D.11, we have

$$
\sum_{i_0=1}^n \sum_{j_0=1}^d \underbrace{G_i(i_0, j_0)}_{1 \times 1} \underbrace{B_7(X)}_{n \times d} = \underbrace{z_7(X)}_{n \times n} \underbrace{X}_{n \times d} \underbrace{W^\top}_{d \times d}
$$

Let $U_7, V_7 \in \mathbb{R}^{n \times k_7}$ be defined in Lemma E.4.

Let $\widetilde{z}_7(X) := U_7 V_7^\top$.

By Lemma E.4, we have

$$\|\widetilde{z}_7(X) - z_7(X)\|_\infty \le \epsilon/\operatorname{poly}(n) \tag{6}$$

**Proof of running time.**

We compute in the following way:

- Compute $\underbrace{V_7^\top}_{k_7 \times n} \underbrace{X}_{n \times d}$, which takes $n^{1+o(1)}$ time.

- Compute $\underbrace{V_7^\top X}_{k_7 \times d} \underbrace{W^\top}_{d \times d}$, which takes $n^{1+o(1)}$ time.

- Compute $\underbrace{U_7}_{n \times k_7} \underbrace{V_7^\top X W^\top}_{k_7 \times d}$, which takes $n^{1+o(1)}$ time.

Therefore, the overall running time is $n^{1+o(1)}$.

**Proof of error bound.**

We have

$$\begin{aligned}
\|\widetilde{z}_7(X) X W^\top - z_7(X) X W^\top\|_\infty &\le d \cdot n \|\widetilde{z}_7(X) - z_7(X)\|_\infty \|X\|_\infty \|W\|_\infty \\
&\le d \cdot n (\epsilon/\operatorname{poly}(n)) \|X\|_\infty \|W\|_\infty \\
&\le \epsilon/\operatorname{poly}(n)
\end{aligned}$$

where the 1st step is from basic linear algebra, the 2nd step comes from Eq. (6), the 3rd step is because of $\|W\|_\infty \le \operatorname{poly}(n)$ and $\|X\|_\infty \le \operatorname{poly}(n)$.

$\square$

### E.3 Fast computation for $B_8(X)$ term

Then, we can do fast computations on $B_8(X)$ term.

**Lemma E.6** (Fast computation for $B_8(X)$ term)**.** *If we have the below conditions,*

- *Let $B_8(X) \in \mathbb{R}^{n \times d}$ be defined in Lemma D.5.*

- *We define $D_8 \in \mathbb{R}^{n \times d}$, where $D_8 := \sum_{i_0=1}^n \sum_{j_0=1}^d G_i(i_0, j_0) B_8(X)$.*

- *Let $X \in \mathbb{R}^{n \times d}, W, W_V \in \mathbb{R}^{d \times d}$ be defined in Definition C.3.*

- *Assuming each entry of $X, W, W_V, G_i$ can be re represented using $O(\log(n))$ bits.*

- *Let $G_i \in \mathbb{R}^{n \times d}$ denote the gradient matrix resulting from the application of the chain rule up to the function $g_i$, i.e., $G_i = \frac{\mathrm{d}L(X)}{\mathrm{d}\mathsf{Attn}_i(T_{i-1}(X))}$.*

- *For $i_2 \in [n], j_2 \in [d]$, let $G_i(i_2, j_2)$ denote the $(i_2, j_2)$-th entry of $G_i$.*

*Then, we can show that, there is an algorithm to approximate $D_8$ in $n^{1+o(1)}$ time, and it can achieve $\epsilon/\operatorname{poly}(n)$ accuracy.*

*Namely, the algorithm output $\widetilde{D}_8$ satisfies*

$$\|D_8 - \widetilde{D}_8\|_\infty \le \epsilon/\operatorname{poly}(n)$$

*Proof.* Recall that in Lemma D.12, we have

$$\sum_{i_0=1}^n \sum_{j_0=1}^d \underbrace{G_i(i_0, j_0)}_{1 \times 1} \underbrace{B_8(X)}_{n \times d} = \underbrace{f(X)}_{n \times n} \underbrace{G_i}_{n \times d} \underbrace{W_V^\top}_{d \times d}$$

Let $\widetilde{f}(X) := U_1 V_1^\top$ denote the approximation of $f(X)$.

By Lemma C.13, we have

$$\|f(X) - \widetilde{f}(X)\|_\infty \leq \epsilon/\operatorname{poly}(n) \tag{7}$$

**Proof of running time.**

We compute in the following way:

- Compute $\underbrace{V_1^\top}_{k_1 \times n} \underbrace{G_i}_{n \times d}$, which takes $n^{1+o(1)}$ time.

- Compute $\underbrace{V_1^\top G_i}_{k_1 \times d} \underbrace{W_V^\top}_{d \times d}$, which takes $n^{1+o(1)}$ time.

- Compute $\underbrace{U_1}_{n \times k_1} \underbrace{V_1^\top G_i W_V^\top}_{k_1 \times d}$, which takes $n^{1+o(1)}$ time.

Therefore, the overall running time is $n^{1+o(1)}$.

**Proof of error bound.**

We have

$$\|\widetilde{f}(X) G_i W_V^\top - f(X) G_i W_V^\top\|_\infty$$
$$\leq d \cdot n \|\widetilde{f}(X) - f(X)\|_\infty \|G_i\|_\infty \|W_V\|_\infty$$
$$\leq d \cdot n (\epsilon/\operatorname{poly}(n)) \|G_i\|_\infty \|W_V\|_\infty$$
$$\leq \epsilon/\operatorname{poly}(n)$$

where the 1st step is from basic linear algebra, the 2nd step comes from Eq.(7), the 3rd step is because of $\|G_i\|_\infty \leq \operatorname{poly}(n)$ and $\|W_V\|_\infty \leq \operatorname{poly}(n)$.

$\square$

### E.4 Fast computation for $B_2(X)$ term

Then, we provide the proof of how to do fast computation on $B_2(X)$.

**Lemma E.7** (Fast computation for $z_2(X)$). *If we have the below conditions,*

- *Let $z_2(X) \in \mathbb{R}^{n \times n}$ be defined as in Lemma D.13.*

- *Let $X \in \mathbb{R}^{n \times d}, W, W_V \in \mathbb{R}^{d \times d}$ be defined in Definition C.3.*

- *Assuming each entry of $X, W, W_V, G_i$ can be re represented using $O(\log(n))$ bits.*

- *Let $G_i \in \mathbb{R}^{n \times d}$ denote the gradient matrix resulting from the application of the chain rule up to the function $g_i$, i.e., $G_i = \frac{\mathrm{d}L(X)}{\mathrm{d}\mathsf{Attn}_i(T_{i-1}(X))}$.*

- *For $i_2 \in [n], j_2 \in [d]$, let $G_i(i_2, j_2)$ denote the $(i_2, j_2)$-th entry of $G_i$.*

*Then, for some $k_9 = n^{o(1)}$, there are matrices $U_9, V_9 \in \mathbb{R}^{n \times k_9}$ such that $\|U_9 V_9^\top - z_2(X)\|_\infty \leq \epsilon/\operatorname{poly}(n)$. The matrices $U_9, V_9$ can be constructed in $n^{1+o(1)}$ time.*

*Proof.* Recall that in Lemma D.13, we have defined $z_2(X) \in \mathbb{R}^{n \times n}$, where the $i_0$-th row of $z_2(X)$ satisfies

$$\underbrace{z_2(X)_{i_0,*}}_{n \times 1} = \underbrace{(G_i(i_0,*))^\top}_{1 \times d} \underbrace{s(X)_{i_0,*}}_{d \times 1} \underbrace{f(X)_{i_0,*}}_{n \times 1}$$

Recall that $K \in \mathbb{R}^n$ has been defined in Definition D.16.

By Lemma E.1, we have $K$ can be computed in $O(n \cdot d)$ time.

We also have

$$\underbrace{z_2(X)}_{n \times n} = \underbrace{\mathrm{diag}(K)}_{n \times n} \underbrace{f(X)}_{n \times n}$$

By Lemma C.13, let $U_1, V_1$ be the low rank approximation of $f(X)$, such that $\|U_1 V_1^\top - f(X)\|_\infty \leq \epsilon/\mathrm{poly}(n)$.

Let $U_9 = \mathrm{diag}(K)U_1$, $V_6 = V_1$.

We have $U_9 = \underbrace{\mathrm{diag}(K)}_{n \times n} \underbrace{U_1}_{n \times k_1}$ can be computed in $nk_1$ time.

The overall running time for constructing $U_9$ and $V_9$ is $n^{1+o(1)}$.

Then, we consider the error bound.

We have

$$
\begin{aligned}
\|U_9 V_9^\top - z_2(X)\|_\infty &= \|\mathrm{diag}(K)U_1 V_1^\top - \mathrm{diag}(K)f(X)\|_\infty \\
&\leq n\|U_1 V_1^\top - f(X)\|_\infty \|\mathrm{diag}(K)\|_\infty \\
&\leq n(\epsilon/\mathrm{poly}(n))\|\mathrm{diag}(K)\|_\infty \\
&\leq \epsilon/\mathrm{poly}(n) \qquad\qquad (8)
\end{aligned}
$$

where the 1st step is from the choice of $U_6, V_6$, the 2nd step comes from basic linear algebra, the 3rd step is because of Lemma C.13, the 4th step is due to $\|\mathrm{diag}(K)\|_\infty \leq \mathrm{poly}(n)$.

$\square$

**Lemma E.8** (Fast computation for $B_2(X)$ term). *If we have the below conditions,*

- *Let $B_2(X) \in \mathbb{R}^{n \times d}$ be defined in Lemma D.6.*

- *We define $D_2 \in \mathbb{R}^{n \times d}$, where $D_2 := \sum_{i_0=1}^n \sum_{j_0=1}^d \underbrace{G_i(i_0, j_0)}_{1 \times 1} \underbrace{e_{i_0}}_{n \times 1} \underbrace{B_2(X)^\top}_{1 \times d}$.*

- *Let $X \in \mathbb{R}^{d \times n}, W, W_V \in \mathbb{R}^{d \times d}, B \in \mathbb{R}^{n \times d}$ be defined in Definition C.3.*

- *Assuming each entry of $X, W, W_V, B, G_i$ can be re represented using $O(\log(n))$ bits.*

- *Let $G_i \in \mathbb{R}^{n \times d}$ denote the gradient matrix resulting from the application of the chain rule up to the function $g_i$, i.e., $G_i = \frac{dL(X)}{d\mathrm{Attn}_i(T_{i-1}(X))}$.*

- *For $i_2 \in [n], j_2 \in [d]$, let $G_i(i_2, j_2)$ denote the $(i_2, j_2)$-th entry of $G_i$.*

*Then, we can show that, there is an algorithm to approximate $D_2$ in $n^{1+o(1)}$ time, and it can achieve $\epsilon/\mathrm{poly}(n)$ accuracy.*

*Namely, the algorithm output $\widetilde{D}_2$ satisfies*

$$\|D_2 - \widetilde{D}_2\|_\infty \leq \epsilon/\mathrm{poly}(n)$$

*Proof.* In Lemma D.13, we have

$$\sum_{i_0=1}^n \sum_{j_0=1}^d \underbrace{G_i(i_0, j_0)}_{1 \times 1} \underbrace{e_{i_0}}_{n \times 1} \underbrace{B_2(X)^\top}_{1 \times d} = -\underbrace{z_2(X)}_{n \times n} \underbrace{X}_{n \times d} \underbrace{W}_{d \times d}$$

Let $U_9, V_9 \in \mathbb{R}^{n \times k_9}$ be defined in Lemma E.7.

Let $\widetilde{z}_2(X) := U_9 V_9^\top$.

By Lemma E.7, we have

$$\|\widetilde{z}_2(X) - z_2(X)\|_\infty \le \epsilon/\operatorname{poly}(n) \tag{9}$$

**Proof of running time.**

We compute in the following way:

- Compute $\underbrace{V_9^\top}_{k_9 \times n} \underbrace{X}_{n \times d}$, which takes $n^{1+o(1)}$ time.

- Compute $\underbrace{V_9^\top X}_{k_9 \times d} \underbrace{W}_{d \times d}$, which takes $n^{1+o(1)}$ time.

- Compute $\underbrace{U_9}_{n \times k_9} \underbrace{V_9^\top X W}_{k_9 \times d}$, which takes $n^{1+o(1)}$ time.

Therefore, the overall running time is $n^{1+o(1)}$.

**Proof of error bound.**

We have

$$\begin{aligned}
\|\widetilde{z}_2(X) X W - z_2(X) X W\|_\infty &\le d \cdot n \|\widetilde{z}_2(X) - z_2(X)\|_\infty \|X\|_\infty \|W\|_\infty \\
&\le d \cdot n (\epsilon/\operatorname{poly}(n)) \|X\|_\infty \|W\|_\infty \\
&\le \epsilon/\operatorname{poly}(n)
\end{aligned}$$

where the 1st step is from basic linear algebra, the 2nd step comes from Eq.(9), the 3rd step is because of $\|W\|_\infty \le \operatorname{poly}(n)$ and $\|X\|_\infty \le \operatorname{poly}(n)$.

$\square$

### E.5 FAST COMPUTATION FOR $B_4(X)$ TERM

Finally, our analysis shows that we can do fast computations for $B_4(X)$ term. After that, we showed that all terms can be computed quickly.

**Lemma E.9** (Fast computation for $z_4(X)$). *If we have the below conditions,*

- *Let $z_4(X) \in \mathbb{R}^{n \times n}$ be defined in Lemma D.14.*

- *Let $X \in \mathbb{R}^{n \times d}, W, W_V \in \mathbb{R}^{d \times d}$ be defined in Definition C.3.*

- *Assuming each entry of $X, W, W_V, G_i$ can be re represented using $O(\log(n))$ bits.*

- *Let $G_i \in \mathbb{R}^{n \times d}$ denote the gradient matrix resulting from the application of the chain rule up to the function $g_i$, i.e., $G_i = \frac{\mathrm{d}L(X)}{\mathrm{d}\mathsf{Attn}_i(T_{i-1}(X))}$.*

- *For $i_2 \in [n], j_2 \in [d]$, let $G_i(i_2, j_2)$ denote the $(i_2, j_2)$-th entry of $G_i$.*

*Then, for some $k_{10} = n^{o(1)}$, there are matrices $U_{10}, V_{10} \in \mathbb{R}^{n \times k_{10}}$, let $\widetilde{z}_4(X) := U_{10} V_{10}^\top$, such that $\|\widetilde{z}_4(X) - z_4(X)\|_\infty \le \epsilon/\operatorname{poly}(n)$. The matrices $U_{10}, V_{10}$ can be constructed in $n^{1+o(1)}$ time.*

*Proof.* In Lemma D.14, we have defined $z_4(X) \in \mathbb{R}^{n \times n}$, where the $i_0$-th column of $z_4(X)$ satisfies

$$\underbrace{z_4(X)_{i_0,*}}_{n \times 1} = (\underbrace{f(X)_{i_0,*}}_{n \times 1} \odot \underbrace{(h(X) G_i(i_0, *))}_{n \times 1})$$

which is equivalent to

$$\underbrace{z_4(X)}_{n \times n} = (\underbrace{f(X)}_{n \times n} \odot \underbrace{G_i}_{n \times d} \underbrace{h(X)^\top}_{d \times n})$$

By Lemma C.13, let $U_1, V_1$ be the low rank approximation of $f(X)$, such that $\|U_1 V_1^\top - f(X)\|_\infty \le \epsilon / \operatorname{poly}(n)$.

We choose $U_{10} = U_1 \oslash G_i$ and $V_{10} = V_1 \oslash h(X)$, where $U_{10}, V_{10} \in \mathbb{R}^{n \times k_1 d}$.

**Proof of running time.**

For $U_{10} = U_1 \oslash G_i$, since $U_1 \in \mathbb{R}^{n \times k_1}, G_i \in \mathbb{R}^{n \times d}$, constructing $U_{10}$ takes $O(ndk_1) = O(n^{1+o(1)})$ time.

Similarly, constructing $V_{10}$ takes $O(n^{1+o(1)})$ time.

**Proof of error bound.**

Let $\widetilde{f}(X) := U_1 V_1^\top$.

Using Fact C.2, we have

$$
\begin{aligned}
\|\widetilde{z}_4(X) - z_4(X)\|_\infty &= \|U_{10} V_{10}^\top - f(X) \odot (G_i \cdot h(X)^\top)\|_\infty \\
&= \|(U_1 \oslash G_i)(V_1 \oslash h(X))^\top - f(X) \odot (G_i \cdot h(X)^\top)\|_\infty \\
&= \|(U_1 V_1^\top) \odot (G_i \cdot h(X)^\top) - f(X) \odot (G_i \cdot h(X)^\top)\|_\infty
\end{aligned}
$$

where the 1st step is from the definition of $\widetilde{z}_4(X), z_4(X)$, the 2nd step comes from the choice of $U_{10}$ and $V_{10}$, the 3rd step is because of Fact C.2.

$$
\begin{aligned}
\|(U_1 V_1^\top) \odot (G_i \cdot h(X)^\top) - f(X) \odot (G_i \cdot h(X)^\top)\|_\infty &= \|U_1 V_1^\top - f(X)\|_\infty \|G_i \cdot h(X)^\top\|_\infty \\
&\le d \cdot (\epsilon / \operatorname{poly}(n)) \|h(X)\|_\infty \|G_i\|_\infty \\
&\le \epsilon / \operatorname{poly}(n)
\end{aligned}
$$

where the 1st step is from basic linear algebra, the 2nd step comes from $\|U_1 V_1 - f(X)\|_\infty \le \epsilon / \operatorname{poly}(n)$, the 3rd step is because of Lemma C.18 and $\|G_i\|_\infty \le \operatorname{poly}(n)$.

$\square$

**Lemma E.10** (Fast computation for $B_4(X)$ term). *If we have the below conditions,*

- *Let $B_4(X) \in \mathbb{R}^{n \times d}$ be defined in Lemma D.7.*

- *We define $D_4 \in \mathbb{R}^{n \times d}$, where $D_4 := \sum_{i_0=1}^{n} \sum_{j_0=1}^{d} \underbrace{G_i(i_0, j_0)}_{1 \times 1} \underbrace{e_{i_0}}_{n \times 1} \underbrace{B_4(X)^\top}_{1 \times d}.$*

- *Let $X \in \mathbb{R}^{n \times d}, W, W_V \in \mathbb{R}^{d \times d}$ be defined in Definition C.3.*

- *Assuming each entry of $X, W, W_V, G_i$ can be re represented using $O(\log(n))$ bits.*

- *Let $G_i \in \mathbb{R}^{n \times d}$ denote the gradient matrix resulting from the application of the chain rule up to the function $g_i$, i.e., $G_i = \frac{\mathrm{d}L(X)}{\mathrm{d}\mathsf{Attn}_i(T_{i-1}(X))}$.*

- *For $i_2 \in [n], j_2 \in [d]$, let $G_i(i_2, j_2)$ denote the $(i_2, j_2)$-th entry of $G_i$.*

*Then, we can show that, there is an algorithm to approximate $D_4$ in $n^{1+o(1)}$ time, and it can achieve $\epsilon / \operatorname{poly}(n)$ accuracy.*

*Namely, the algorithm output $\widetilde{D}_4$ satisfies*

$$\|D_4 - \widetilde{D}_4\|_\infty \le \epsilon / \operatorname{poly}(n)$$

*Proof.* In Lemma D.14, we have

$$\sum_{i_0=1}^{n} \sum_{j_0=1}^{d} \underbrace{G_i(i_0, j_0)}_{1 \times 1} \underbrace{e_{i_0}}_{n \times 1} \underbrace{B_4(X)^{\top}}_{1 \times d} = \underbrace{z_4(X)}_{n \times n} \underbrace{X}_{n \times d} \underbrace{W}_{d \times d}$$

Let $\widetilde{z}_4(X) := U_{10} V_{10}^{\top}$.

By Lemma E.9, we have

$$\|\widetilde{z}_4(X) - z_4(X)\|_{\infty} \leq \epsilon / \operatorname{poly}(n) \tag{10}$$

**Proof of running time.**

We compute in the following way:

- Compute $\underbrace{V_{10}^{\top}}_{k_{10} \times n} \underbrace{X}_{n \times d}$, which takes $n^{1+o(1)}$ time.

- Compute $\underbrace{V_{10}^{\top} X}_{k_{10} \times d} \underbrace{W}_{d \times d}$, which takes $n^{1+o(1)}$ time.

- Compute $\underbrace{U_{10}}_{n \times k_{10}} \underbrace{V_{10}^{\top} X W}_{k_{10} \times d}$, which takes $n^{1+o(1)}$ time.

Therefore, the overall running time is $n^{1+o(1)}$.

**Proof of error bound.**

We have

$$\begin{aligned}
\|\widetilde{z}_4(X) X W - z_4(X) X W\|_{\infty} &\leq d \cdot n \|\widetilde{z}_4(X) - z_4(X)\|_{\infty} \|X\|_{\infty} \|W\|_{\infty} \\
&\leq d \cdot n (\epsilon / \operatorname{poly}(n)) \|X\|_{\infty} \|W\|_{\infty} \\
&\leq \epsilon / \operatorname{poly}(n)
\end{aligned}$$

where the 1st step is from basic linear algebra, the 2nd step comes from Eq.(10), the 3rd step is because of $\|W\|_{\infty} \leq \operatorname{poly}(n)$ and $\|X\|_{\infty} \leq \operatorname{poly}(n)$.

$\square$

### E.6 PUTTING EVERYTHING TOGETHER

After we have analyzed each $B_i(X)$ term in the previous section, we put them together in this section, to analyze the overall running time and error bound of the gradient of $L(X)$ on $T_i(X)$ in Lemma E.11.

**Lemma E.11** (Fast computation for $\frac{\mathrm{d}L(X)}{\mathrm{d}T_{i-1}(X)}$, formal version of Lemma 4.1). *If we have the below conditions,*

- *Let $L(X)$ be defined as Definition 2.1.*

- *Let $m$ denote the number of self-attention transformer model (see Definition 1.3).*

- *For any $i \in [m]$, let $T_i(X)$ be defined as Definition 2.3.*

- *Let $X \in \mathbb{R}^{n \times d}, W, W_V \in \mathbb{R}^{d \times d}$ be defined in Definition C.3.*

- *Assuming each entry of $X, W, W_V, G_i$ can be re represented using $O(\log(n))$ bits.*

- *Let $G_i \in \mathbb{R}^{n \times d}$ denote the gradient matrix resulting from the application of the chain rule up to the function $g_i$, i.e., $G_i = \frac{\mathrm{d}L(X)}{\mathrm{d}\mathsf{Attn}_i(T_{i-1}(X))}$.*

- *Assume $G_i$ can be computed in $n^{1+o(1)}$ time.*

*We can show that $\frac{\mathrm{d}L(X)}{\mathrm{d}T_{i-1}(X)}$ can be approximated in $n^{1+o(1)}$ time, with $1/\operatorname{poly}(n)$ approximation error. Namely, our algorithm can output $\widetilde{g}_t$ in $n^{1+o(1)}$ time, which satisfies*

$$\|\widetilde{g}_t - \frac{\mathrm{d}L(X)}{\mathrm{d}T_{i-1}(X)}\|_\infty \le 1/\operatorname{poly}(n)$$

*Proof.* By Lemma D.9, we have

$$\frac{\mathrm{d}L(X)}{\mathrm{d}T_{i-1}(X)} = \sum_{i_0=1}^n \sum_{j_0=1}^d \underbrace{G_i(i_0,j_0)}_{1\times 1} \cdot (\underbrace{B_6(X) + B_7(X) + B_8(X)}_{n\times d} + \underbrace{e_{i_0}}_{n\times 1} \underbrace{(B_2(X) + B_4(X))^\top}_{1\times d})$$

$$= \sum_{i\in\{2,4,6,7,8\}} D_i$$

where the 1st step is from Lemma D.9, the 2nd step comes from the definition of $D_6, D_7, D_8, D_2, D_4$.

Then, by Lemma E.3, E.5, E.6, E.8, E.10, we have $D_6, D_7, D_8, D_2, D_4 \in \mathbb{R}^{n\times d}$ can be approximated in $n^{1+o(1)}$ time, with up to $\epsilon/\operatorname{poly}(n)$ error.

Namely, for $i \in \{2,4,6,7,8\}$, let $\widetilde{D}_i \in \mathbb{R}^{n\times d}$ denote the approximated version of $D$, we have

$$\|\widetilde{D}_i - D\|_\infty \le \epsilon/\operatorname{poly}(n)$$

Let $\widetilde{g}_t = \sum_{i\in\{2,4,6,7,8\}} \widetilde{D}_i$.

**Proof of running time.**

The running time for computing $\widetilde{g}_t = \sum_{i\in\{2,4,6,7,8\}} \widetilde{D}_i$ is $O(nd)$.

Therefore, the overall running time for computing $\widetilde{g}_t$ is $n^{1+o(1)}$.

**Proof of error bound.**

We have

$$\|\widetilde{g}_t - \frac{\mathrm{d}L(X)}{\mathrm{d}T_{i-1}(X)}\|_\infty = \|\sum_{i\in\{2,4,6,7,8\}} (\widetilde{D}_i - D_i)\|_\infty$$

$$\le \sum_{i\in\{2,4,6,7,8\}} \|(\widetilde{D}_i - D_i)\|_\infty$$

$$\le \epsilon/\operatorname{poly}(n)$$

where the 1st step is from the definition of $\widetilde{g}_t$ and $\frac{\mathrm{d}L(X)}{\mathrm{d}T_{i-1}(X)}$, the 2nd step comes from basic algebra, the 3rd step is because of $\|\widetilde{D}_i - D\|_\infty \le \epsilon/\operatorname{poly}(n)$.

Then, choose $\epsilon = 1/\operatorname{poly}(n)$, we have

$$\|\widetilde{g}_t - \frac{\mathrm{d}L(X)}{\mathrm{d}T_{i-1}(X)}\|_\infty \le 1/\operatorname{poly}(n)$$

$\square$

# F    FAST COMPUTATION FOR GRADIENT ON $W$

In Section F.1, we introduce some essential notations used in this section. In Section F.2, we offer the gradient of $s(X)$ on $W$, which is equivalent to the gradient of the output of the attention mechanism on $W$. In Section F.3, we illustrate the gradient of $L(X)$ on $W$. In Section F.4, we introduce the almost linear time algorithm for calculating the gradient of $L(X)$ on $W$, along with the error bound analysis.

### F.1 KEY CONCEPTS

**Definition F.1** (Definition of A, (Alman & Song, 2024a)). *Let $A_1, A_2 \in \mathbb{R}^{n \times d}$ be two matrices. Suppose that $\mathsf{A} = A_1 \otimes A_2 \in \mathbb{R}^{n^2 \times d^2}$. We define $\mathsf{A}_{j_0} \in \mathbb{R}^{n \times d^2}$ be a $n \times d^2$ size sub-block from $\mathsf{A}$. Note that there are $n$ such sub-blocks.*

**Remark F.2.** *Note that the $A_1, A_2$ matrices in Definition F.1 is $X$ in our setting. Since in (Alman & Song, 2024a), they consider a more general setting, where $A_1, A_2$ can be difference matrices, while in our problem, we consider self-attention. Therefore, in our paper, we have $A_1 = A_2 = X$.*

### F.2 GRADIENT OF $s(X)$ ON $W$

We begin with introducing the close form of the gradient of $s(X)$.

(Alman & Song, 2024a) proved the close form of the gradient of $c(X) = s(X) - B$ with respect to $W$ for a constant matrix $B$. By chain rule, this is equivalent to the gradient of $s(X)$ with respect to $W$.

**Lemma F.3** (Gradient of $s(X)$ on $W$, Lemma B.1 in (Alman & Song, 2024a)). *If we have the below conditions,*

- *Let $\mathsf{A}$ be defined as Definition F.1. For every $i \in [d^2]$, define $\mathsf{A}_{j_0,i} \in \mathbb{R}^n$ to be the $i$-th column for $\mathsf{A}_{j_0} \in \mathbb{R}^{n \times d^2}$.*

- *Let $f(X), h(X), s(X)$ be defined as Definition C.8, C.9, C.10.*

- *Let $W \in \mathbb{R}^{d \times d}$ be defined as Definition C.3. Let $w \in \mathbb{R}^{d^2}$ denote the vector representation of $W$.*

*Then, for each $i \in [d^2]$, we have For each $j_0 \in [n]$, for every $i_0 \in [d]$*

$$\frac{\mathrm{d}s(X)_{j_0,i_0}}{\mathrm{d}w_i} = \langle \mathsf{A}_{j_0,i} \odot f(X)_{j_0}, h(X)_{i_0} \rangle - \langle f(X)_{j_0}, h(X)_{i_0} \rangle \cdot \langle \mathsf{A}_{j_0,i}, f(X)_{j_0} \rangle$$

### F.3 GRADIENT OF $L(X)$ ON $W$

Differing from the $\ell_2$ loss function used in (Alman & Song, 2024a), our framework supports arbitrary loss functions. Therefore, we use Lemma F.4 to illustrate the gradient of $L(X)$ on $W$.

**Lemma F.4** (Gradient of $L(X)$ on $W$). *If we have the below conditions,*

- *Let $L(X)$ be defined as Definition 2.1.*

- *Let $W \in \mathbb{R}^{d \times d}, X \in \mathbb{R}^{n \times d}$ be Defined as Definition C.3.*

- *Let $p(X)$ be defined as Definition C.12.*

*Then, we can show that*

$$\frac{\mathrm{d}L(X)}{\mathrm{d}W_i} = X^\top \cdot p(X) \cdot X$$

*Proof.* By Lemma F.3, we have, for each $i \in [d^2]$, we have For each $j_0 \in [n]$, for every $i_0 \in [d]$

$$\frac{\mathrm{d}s(X)_{j_0,i_0}}{\mathrm{d}w_i} = \langle \underbrace{\mathsf{A}_{j_0,i}}_{n \times 1} \odot \underbrace{f(X)_{j_0}}_{n \times 1}, \underbrace{h(X)_{i_0}}_{n \times 1} \rangle - \langle \underbrace{f(X)_{j_0}}_{n \times 1}, \underbrace{h(X)_{i_0}}_{n \times 1} \rangle \cdot \langle \underbrace{\mathsf{A}_{j_0,i}}_{n \times 1}, \underbrace{f(X)_{j_0}}_{n \times 1} \rangle \quad (11)$$

By Fact C.1, we have

$$\langle \mathsf{A}_{j_0,i} \odot f(X)_{j_0}, h(X)_{i_0} \rangle = \mathsf{A}_{j_0,i}^\top \operatorname{diag}(f(X)_{j_0}) h(X)_{i_0}$$

and

$$\langle f(X)_{j_0}, h(X)_{i_0} \rangle \cdot \langle f(X)_{j_0}, \mathsf{A}_{j_0,i} \rangle = \mathsf{A}_{j_0,i}^\top f(X)_{j_0} f(X)_{j_0}^\top h(X)_{i_0}$$

By Eq. (11), for each $i \in [d^2]$, we have For each $j_0 \in [n]$, for every $i_0 \in [d]$, we have

$$\frac{\mathrm{d}s(X)_{j_0,i_0}}{\mathrm{d}w_i} = \mathsf{A}_{j_0,i}^{\top}(\mathrm{diag}(f(X)_{j_0}) - f(X)_{j_0}f(X)_{j_0}^{\top})h(X)_{i_0}$$

which implies,

$$\frac{\mathrm{d}s(X)_{j_0,i_0}}{\mathrm{d}W} = \underbrace{\mathsf{A}_{j_0}^{\top}}_{d^2 \times n} \underbrace{(\mathrm{diag}(f(X)_{j_0}) - f(X)_{j_0}f(X)_{j_0}^{\top})}_{n \times n} \underbrace{h(X)_{i_0}}_{n \times 1} \tag{12}$$

By Lemma C.4, for $i \in [m]$, we have

$$\frac{\mathrm{d}L(X)}{\mathrm{d}W_i} = \sum_{i_2=1}^{n} \sum_{j_2=1}^{d} G_i(i_2, j_2) \cdot \frac{\mathrm{d}\mathsf{Attn}_i(T_{i-1}(X))_{i_2,j_2}}{\mathrm{d}W_i}. \tag{13}$$

By the definition of $s(X)$ (Definition C.10), we have

$$s(X) = \mathsf{Attn}_i(T_{i-1}(X))$$

Combining Eq. (12) and Eq. (13), for each $i \in [m]$, we have

$$\frac{\mathrm{d}L(X)}{\mathrm{d}W_i} = \sum_{j_0=1}^{n} \sum_{i_0=1}^{d} \underbrace{G_i(j_0, i_0)}_{1 \times 1} \cdot \underbrace{\mathsf{A}_{j_0}^{\top}}_{d^2 \times n} \underbrace{(\mathrm{diag}(f(X)_{j_0}) - f(X)_{j_0}f(X)_{j_0}^{\top})}_{n \times n} \underbrace{h(X)_{i_0}}_{n \times 1} \tag{14}$$

Recall that we have defined $q(X)$ in Definition C.11,

$$q(X)_{j_0} := \sum_{i_0=1}^{d} G_i(j_0, i_0) \cdot h(X)_{i_0} \tag{15}$$

Recall that $p(x)_{j_0} \in \mathbb{R}^n$ is define as Definition C.12,

$$p(x)_{j_0} := (\mathrm{diag}(f(x)_{j_0}) - f(x)_{j_0}f(x)_{j_0}^{\top})q(x)_{j_0}. \tag{16}$$

Then, we have

$$\frac{\mathrm{d}L(X)}{\mathrm{d}W_i} = \sum_{j_0=1}^{n} \sum_{i_0=1}^{d} \underbrace{G_i(j_0, i_0)}_{1 \times 1} \cdot \underbrace{\mathsf{A}_{j_0}^{\top}}_{d^2 \times n} \underbrace{(\mathrm{diag}(f(X)_{j_0}) - f(X)_{j_0}f(X)_{j_0}^{\top})}_{n \times n} \underbrace{h(X)_{i_0}}_{n \times 1}$$

$$= \sum_{j_0=1}^{n} \underbrace{\mathsf{A}_{j_0}^{\top}}_{d^2 \times n} \underbrace{(\mathrm{diag}(f(X)_{j_0}) - f(X)_{j_0}f(X)_{j_0}^{\top})}_{n \times n} \underbrace{q(X)_{j_0}}_{n \times 1}$$

$$= \sum_{j_0=1}^{n} \mathsf{A}_{j_0}^{\top} p_{j_0}(X)$$

$$= \underbrace{X^{\top}}_{d \times n} \underbrace{p(X)}_{n \times n} \underbrace{X}_{n \times d}$$

where the 1st step is from Eq. (14), the 2nd step comes from Eq. (15), the 3rd step is because of Eq. (16), the 4th step is due to the tensor tricks.

$\square$

### F.4 FAST COMPUTATION

Finally, we introduce the almost linear time algorithm and its error analysis of the gradient of $L(X)$ on $W$ in Lemma F.5.

**Lemma F.5** (Fast computation for $\frac{\mathrm{d}L(X)}{\mathrm{d}W_i}$). *If we have the below conditions,*

- *Let $L(X)$ be defined as Definition 2.1.*

- *Let $m$ denote the number of self-attention transformer layers (see Definition 1.3).*

- *For any $i \in [m]$, let $W_i = W_{Q_i} W_{K_i}^\top$ denote the attention weight in the $i$-th transformer layer.*

*We can show that $\frac{\mathrm{d}L(X)}{\mathrm{d}W_i}$ can be approximated in $n^{1+o(1)}$ time, with $1/\operatorname{poly}(n)$ approximation error. Namely, our algorithm can output $\widetilde{g}_w$ in $n^{1+o(1)}$ time, which satisfies*

$$\|\widetilde{g}_w - \frac{\mathrm{d}L(X)}{\mathrm{d}W_i}\|_\infty \le 1/\operatorname{poly}(n)$$

*Proof.* Recall by Lemma C.15, C.16, we have defined $p_1(X), p_2(X) \in \mathbb{R}^{n \times n}$.

In those Lemmas, we have $p_1(X), p_2(X)$ have low rank approximation $U_3 V_3^\top$ and $U_4 V_4^\top$, respectively.

By the definition of $p(X)$ (Definition C.12), we have

$$p(X) = p_1(X) - p_2(X) \tag{17}$$

Then, by Lemma F.4, we have

$$\frac{\mathrm{d}L(X)}{\mathrm{d}W_i} = X^\top p(X) X$$
$$= X^\top (p_1(X) - p_2(X)) X$$

where the 1st step is from Lemma F.4, the 2nd step comes from Eq. (17).

Let $\widetilde{p}_1(X), \widetilde{p}_2(X)$ denote the low rank approximations for $p_1(X), p_2(X)$, respectively.

**Proof of running time.** We first compute $X^\top \widetilde{p}_1(X) X$ in following order

- Compute $\underbrace{X^\top}_{d \times n} \underbrace{U_3}_{n \times k_3}$, which takes $n^{1+o(1)}$ time.

- Compute $\underbrace{X^\top U_3}_{d \times k_3} \underbrace{V_3^\top}_{k_3 \times n}$, which takes $n^{1+o(1)}$ time.

- Compute $\underbrace{X^\top U_3 V_3^\top}_{d \times n} \underbrace{X}_{n \times d}$, which takes $n^{1+o(1)}$ time.

The overall running time for $X^\top \widetilde{p}_1(X) X$ is $n^{1+o(1)}$.

Similarly, the overall running time for $X^\top \widetilde{p}_2(X) X$ is $n^{1+o(1)}$.

Since $X^\top \widetilde{p}_1(X) X, X^\top \widetilde{p}_2(X) X \in \mathbb{R}^{d \times d}$, the computation time for $X^\top (\widetilde{p}_1(X) - \widetilde{p}_2(X)) X$ is $O(d^2)$.

Therefore, the overall running time for $X^\top (\widetilde{p}_1(X) - \widetilde{p}_2(X)) X$ is $n^{1+o(1)}$.

**Proof of error bound.**

We consider the error for $X^\top \widetilde{p}_1(X) X$ first.

$$
\begin{aligned}
\|X^\top \widetilde{p}_1(X)X - X^\top p_1(X)X\|_\infty &= \|X^\top(\widetilde{p}_1(X) - p_1(X))X\|_\infty \\
&\leq n^2 \|X\|_\infty^2 \|\widetilde{p}_1(X) - p_1(X)\|_\infty \\
&\leq n^2 (\epsilon/\operatorname{poly}(n))\|X\|_\infty^2 \\
&\leq \epsilon/\operatorname{poly}(n)
\end{aligned}
\tag{18}
$$

where the 1st step is from basic algebra, the 2nd step comes from basic linear algebra, the 3rd step is because of $\|\widetilde{p}_1(X) - p_1(X)\|_\infty \leq \epsilon/\operatorname{poly}(n)$, the 4th step is due to $\|X\|_\infty \leq \operatorname{poly}(n)$.

Similarly, we can have

$$
\|X^\top \widetilde{p}_2(X)X - X^\top p_2(X)X\|_\infty \leq \epsilon/\operatorname{poly}(n)
\tag{19}
$$

Therefore, we have

$$
\begin{aligned}
&\|X^\top \widetilde{p}(X)X - X^\top p(X)X\|_\infty \\
&= \|X^\top \widetilde{p}_1(X)X - X^\top p_1(X)X + X^\top \widetilde{p}_2(X)X - X^\top p_2(X)X\|_\infty \\
&\leq \|X^\top \widetilde{p}_1(X)X - X^\top p_1(X)X\|_\infty + \|X^\top \widetilde{p}_2(X)X - X^\top p_2(X)X\|_\infty \\
&\leq (\epsilon/\operatorname{poly}(n)) + (\epsilon/\operatorname{poly}(n)) \\
&= \epsilon/\operatorname{poly}(n)
\end{aligned}
$$

where the 1st step is from basic algebra, the 2nd step comes from triangle inequality, the 3rd step is because of Eq. (18) and Eq. (19), the 4th step is due to basic algebra.

Then, we choose $\epsilon = 1/\operatorname{poly}(n)$, we have

$$
\|\widetilde{g}_w - \frac{\mathrm{d}L(X)}{\mathrm{d}W_i}\|_\infty \leq 1/\operatorname{poly}(n)
$$

$\square$

## G  FAST COMPUTATION FOR GRADIENT ON $W_V$

In Section G.1, we introduce the close form of the gradient of $s(X)$ on $W_V$. In Section G.2, we provide the close form of the gradient of $L(X)$ on $W_V$. In Section G.3, based on the close form calculated in the previous section, we introduce the almost linear time algorithm for computing the gradient of $L(X)$ on $W_V$.

### G.1  GRADIENT OF $s(X)$ ON $W_V$

Since $s(X) = f(X)h(X)$, we begin with considering the gradient of $h(X)$ on $W_V$ in Lemma G.1.

**Lemma G.1** (Gradient of $h(X)$ on $W_V$). *If we have the below conditions,*

- *Let $h(X)$ be defined as Definition C.9.*

- *Let $W_V$ be defined as Definition C.3.*

*Then, for any $i_0 \in [n], j_0 \in [d]$ and any $i_1, j_1 \in [d]$, we have*

$$
\frac{\mathrm{d}h(X)_{i_0,j_0}}{\mathrm{d}(W_V)_{i_1,j_1}} = \begin{cases} X_{i_0,i_1} & j_0 = j_1 \\ 0 & j_0 \neq j_1 \end{cases}
$$

*Proof.* Since $h_{i_0,j_0}$ satisfies

$$
h_{i_0,j_0} = X_{i_0,*}^\top (W_V)_{*,j_0},
$$

we have $h_{i_0,j_0}$ only depends on $(W_V)_{*,j_0}$.

Hence, we have, for $j_0 \neq j_1$,

$$\frac{\mathrm{d}h(X)_{i_0,j_0}}{\mathrm{d}(W_V)_{i_1,j_1}} = 0$$

For $j_0 = j_1$ case, we have

$$\frac{\mathrm{d}h(X)_{i_0,j_0}}{\mathrm{d}(W_V)_{i_1,j_0}} = X_{i_0,i_1}$$

$\square$

Combining the result in the previous Lemma and the chain rule, we can have the gradient of $s(X)$ on $W_V$ in Lemma G.2.

**Lemma G.2** (Gradient of $s(X)$ on $W_V$). *If we have the below conditions,*

- *Let $s(X)$ be defined as Definition C.10.*

- *Let $W_V$ be defined as Definition C.3.*

*Then, for any $i_2 \in [n], j_2 \in [d]$ and any $i_1, j_1 \in [d]$, we have*

- **Part 1.**

$$\frac{\mathrm{d}s(X)_{i_2,j_2}}{\mathrm{d}(W_V)_{i_1,j_1}} = \begin{cases} f(X)_{i_2,*}^\top X_{*,i_1} & j_2 = j_1 \\ 0 & j_2 \neq j_1 \end{cases}$$

- **Part 2.**

$$\underbrace{\frac{\mathrm{d}s(X)_{i_2,j_2}}{\mathrm{d}W_V}}_{d \times d} = \underbrace{X^\top}_{d \times n} \underbrace{f(X)_{i_2,*}}_{n \times 1} \underbrace{e_{j_2}^\top}_{1 \times d}$$

*Proof.* **Proof of Part 1.**

By Definition C.10, we have

$$s(X)_{i_2,j_2} := f(X)_{i_2,*}^\top h(X)_{*,j_2} \tag{20}$$

Therefore, $s(X)_{i_2,j_2}$ is only depends on $h(X)_{*,j_2}$, which further means $s(X)_{i_2,j_2}$ is only depends on $(W_V)_{*,j_2}$.

Hence, for $j_1 \neq j_2$, we have

$$\frac{\mathrm{d}s(X)_{i_2,j_2}}{\mathrm{d}(W_V)_{i_1,j_2}} = 0$$

We consider $j_1 = j_2$ case.

By, Eq. (20), we can derive that

$$\frac{\mathrm{d}s(X)_{i_2,j_2}}{\mathrm{d}h(X)_{i_3,j_2}} = f(X)_{i_2,i_3} \tag{21}$$

By chain rule, we have

$$\frac{\mathrm{d}s(X)_{i_2,j_2}}{\mathrm{d}(W_V)_{i_1,j_2}} = \sum_{i_3=1}^d \frac{\mathrm{d}s(X)_{i_2,j_2}}{\mathrm{d}h(X)_{i_3,j_2}} \frac{\mathrm{d}h(X)_{i_3,j_2}}{\mathrm{d}(W_V)_{i_1,j_2}}$$

$$= \sum_{i_3=1}^d f(X)_{i_2,i_3} \frac{\mathrm{d}h(X)_{i_3,j_2}}{\mathrm{d}(W_V)_{i_1,j_2}}$$

$$= \sum_{i_3=1}^{d} f(X)_{i_2,i_3} X_{i_3,i_1}$$

$$= f(X)_{i_2,*}^{\top} X_{*,i_1} \tag{22}$$

where the 1st step is from chain rule, the 2nd step comes from Eq. (21), the 3rd step is because of Lemma G.1, the 4th step is due to basic linear algebra.

**Proof of Part 2.**

By Eq (22), we have

$$\underbrace{\frac{\mathrm{d}s(X)_{i_2,j_2}}{\mathrm{d}(W_V)_{*,j_2}}}_{d \times 1} = \underbrace{X^{\top}}_{d \times n} \underbrace{f(X)_{i_2,*}}_{n \times 1}$$

which implies

$$\underbrace{\frac{\mathrm{d}s(X)_{i_2,j_2}}{\mathrm{d}W_V}}_{d \times d} = \underbrace{X^{\top}}_{d \times n} \underbrace{f(X)_{i_2,*}}_{n \times 1} \underbrace{e_{j_2}^{\top}}_{1 \times d}$$

$\square$

### G.2 GRADIENT OF $L(X)$ ON $W_V$

Since we have already got the close form of the gradient of $s(X)$ on $W_V$, we can easily extend it and get the close form of the gradient of $L(X)$ on $W_V$ in Lemma G.3.

**Lemma G.3** (Gradient of $L(X)$ on $W_V$). *If we have the below conditions,*

- *Let $L(X)$ be defined as Definition 2.1.*

- *Let $W_V$ be defined as Definition C.3.*

*Then, we can show that*

$$\underbrace{\frac{\mathrm{d}L(X)}{\mathrm{d}W_{V_i}}}_{d \times d} = \underbrace{X^{\top}}_{d \times n} \underbrace{f(X)}_{n \times n} \underbrace{G_i}_{n \times d}$$

*Proof.* We slightly abuse the notation, using $W_V$ to represent $V_i$ in Lemma G.1, G.2.

By Lemma G.2, we have

$$\underbrace{\frac{\mathrm{d}s(X)_{i_2,j_2}}{\mathrm{d}W_V}}_{d \times d} = \underbrace{X^{\top}}_{d \times n} \underbrace{f(X)_{i_2,*}}_{n \times 1} \underbrace{e_{j_2}^{\top}}_{1 \times d} \tag{23}$$

By Lemma C.4, we have

$$\frac{\mathrm{d}L(X)}{\mathrm{d}W_{V_i}} = \sum_{i_2=1}^{n} \sum_{j_2=1}^{d} G_i(i_2, j_2) \cdot \frac{\mathrm{d}\mathsf{Attn}_i(T_{i-1}(X))_{i_2,j_2}}{\mathrm{d}W_{V_i}}. \tag{24}$$

By Definition C.10 and Definition C.3, we have

$$s(X) = \mathsf{Attn}_i(T_{i-1}(X))$$

Therefore, combining Eq. (23) and Eq. (24), we have

$$\frac{\mathrm{d}L(X)}{\mathrm{d}W_{V_i}}$$

$$= \sum_{i_2=1}^{n} \sum_{j_2=1}^{d} \underbrace{G_i(i_2, j_2)}_{1 \times 1} \underbrace{X^\top}_{d \times n} \underbrace{f(X)_{i_2,*}}_{n \times 1} \underbrace{e_{j_2}^\top}_{1 \times d}$$

$$= \sum_{i_2=1}^{n} \underbrace{X^\top}_{d \times n} \underbrace{f(X)_{i_2,*}}_{n \times 1} \sum_{j_2=1}^{d} \underbrace{G_i(i_2, j_2)}_{1 \times 1} \underbrace{e_{j_2}^\top}_{1 \times d}$$

$$= \sum_{i_2=1}^{n} \underbrace{X^\top}_{d \times n} \underbrace{f(X)_{i_2,*}}_{n \times 1} \underbrace{G_i(i_2, *)^\top}_{1 \times d}$$

$$= \underbrace{X^\top}_{d \times n} \underbrace{f(X)}_{n \times n} \underbrace{G_i}_{n \times d}$$

where the 1st step is from Eq. (23) and Eq. (24), the 2nd step comes from basic algebra, the 3rd step is because of basic linear algebra, the 4th step is due to basic linear algebra.

$\square$

### G.3 FAST COMPUTATION

Finally, we can introduce our almost linear time algorithm for computing the $L(X)$ gradient on $W_V$.

**Lemma G.4** (Fast computation for $\frac{dL(X)}{d(W_V)_i}$, formal version of Lemma 4.1). *If we have the below conditions,*

- *Let $L(X)$ be defined as Definition 2.1.*

- *Let $m$ denote the number of self-attention transformer layers (see Definition 1.3).*

- *For any $i \in [m]$, let $W_{V_i} \in \mathbb{R}^{d \times d}$ denote the attention weight in the $i$-th transformer layer.*

*We can show that $\frac{dL(X)}{dW_{V_i}}$ can be approximated in $n^{1+o(1)}$ time, with $1/\operatorname{poly}(n)$ approximation error. Namely, our algorithm can output $\widetilde{g}_v$ in $n^{1+o(1)}$ time, which satisfies*

$$\|\widetilde{g}_v - \frac{dL(X)}{dW_{V_i}}\|_\infty \le 1/\operatorname{poly}(n)$$

*Proof.* Recall in Lemma C.13, $U_1 V_1^\top$ is the low rank approximation of $f(X)$.

Let $\widetilde{f}(X) := U_1 V_1^\top$ denote the low rank approximation of $f(X)$.

Recall in Lemma G.3, we have

$$\frac{dL(X)}{dW_{V_i}} = \underbrace{X^\top}_{d \times n} \underbrace{f(X)}_{n \times n} \underbrace{G_i}_{n \times d}$$
$$\underbrace{\phantom{\frac{dL(X)}{dW_{V_i}}}}_{d \times d}$$

**Proof of running time.**

We compute $X^\top \widetilde{f}(X) G_i$ in following order

- Compute $\underbrace{X^\top}_{d \times n} \cdot \underbrace{U_1}_{n \times k_1}$, which takes $n^{1+o(1)}$ time.

- Compute $\underbrace{X^\top \cdot U_1}_{d \times k_1} \cdot \underbrace{V_1^\top}_{k_1 \times n}$, which takes $n^{1+o(1)}$ time.

- Compute $\underbrace{X^\top \cdot U_1 \cdot V_1^\top}_{d \times n} \cdot \underbrace{G_i}_{n \times d}$, which takes $d^2 \cdot n$ time.

The overall running time is $n^{1+o(1)}$.

**Proof of error bound.**

We have

$$\|X^\top \cdot f(X) \cdot G_i - X^\top \cdot \widetilde{f}(X) \cdot G_i\|_\infty = \|X^\top \cdot (f(X) - \widetilde{f}(X)) \cdot G_i\|_\infty$$

$$\leq n^2 \|X\|_\infty \|f(X) - \widetilde{f}(X)\|_\infty \|G_i\|_\infty$$

$$\leq n^2 (\epsilon/\operatorname{poly}(n)) \|X\|_\infty \|G_i\|_\infty$$

$$\leq \epsilon/\operatorname{poly}(n)$$

where the 1st step is from basic algebra, the 2nd step comes from basic linear algebra, the 3rd step is because of $\|f(X) - \widetilde{f}(X)\|_\infty \leq \epsilon/\operatorname{poly}(n)$, the 4th step is due to $\|X\|_\infty \leq \operatorname{poly}(n)$ and $\|G_i\|_\infty \leq \operatorname{poly}(n)$.

Let $\widetilde{g}_v = X^\top \cdot \widetilde{f}(X) \cdot G_i$.

We choose $\epsilon = 1/\operatorname{poly}(n)$. Then, we have

$$\|\widetilde{g}_v - \frac{\mathrm{d}L(X)}{\mathrm{d}W_{V_i}}\|_\infty \leq 1/\operatorname{poly}(n)$$

$\square$

## H    GRADIENT APPROXIMATION FOR ENTIRE MODEL

In Section H.1, we introduce the close form of $G_i$ and argue that $G_i$ can be computed in almost linear time $n^{1+o(1)}$. In Section H.2, we provide the almost linear time algorithm for gradient computing on a single-layer transformer. In Section H.3, with the help of math induction, we introduce the almost linear time algorithm for computing the gradient of the multi-layer transformer, along with its approximation error.

### H.1    COMPUTATION TIME FOR $G_i$

Here we consider $g_i$ in Definition 1.3 as a linear layer with an arbitrary non-linear activation $\phi$. Since $g_i$ can be viewed as a composition of an MLP and an activation function, we begin with analyzing the $T_i$ gradient on $\mathsf{Attn}_i$.

**Lemma H.1** (Gradient of $T_i$ on $\mathsf{Attn}_i$ )**.** *If we have the below conditions,*

- *Let $T_i(X)$ be defined as Definition 2.3.*

- *Assuming for any $Z \in \mathbb{R}^{n \times d}$, we have $g_i(Z) \in \mathbb{R}^{n \times d}$, and $g_i(Z) = \phi(ZW_g)$, where $W_g \in \mathbb{R}^{d \times d}$ and $\phi : \mathbb{R} \to \mathbb{R}$ denotes any element-wise activation function. Let $\phi'$ denote the derivative of $\phi$.*

- *We simplify the notation, using $T_i$ and $\mathsf{Attn}_i$ to represent $T_i(X)$ and $\mathsf{Attn}_i(T_{i-1}(X))$, respectively.*

- *For any matrix $Z \in \mathbb{R}^{n \times d}$, we use $Z(i, j)$ to denote the $(i, j)$-th entry of $Z$.*

*Then, we can show that, for any $i_4, i_5 \in [n], j_4, j_5 \in [d]$,*

- **Part 1.**

$$\frac{\mathrm{d}T_i(i_4, j_4)}{\mathrm{d}\mathsf{Attn}_i(i_5, j_5)} = \begin{cases} \underbrace{\phi'(\mathsf{Attn}_i(i_4, *)^\top W_g(*, j_4))}_{1 \times 1} \underbrace{W_g(j_5, j_4)}_{1 \times 1} & i_4 = i_5 \\ 0 & i_4 \neq i_5 \end{cases}$$

- **Part 2.**

$$\underbrace{\frac{\mathrm{d}T_i(i_4, j_4)}{\mathrm{d}\mathsf{Attn}_i}}_{n \times d} = \underbrace{\phi'(\mathsf{Attn}_i(i_4, *)^\top W_g(*, j_4))}_{1 \times 1} \underbrace{e_{i_4}}_{n \times 1} \underbrace{W_g(*, j_4)^\top}_{1 \times d}$$

*Proof.* **Proof of Part 1.**

By the definition of $T_i$ (Definition 2.3), for $i_4 \in [d], j_4 \in [n]$, we have

$$T_i(i_4, j_4) = \phi(\mathsf{Attn}_i(i_4, *)^\top W_g(*, j_4))$$

Therefore, for any $i_5 \neq i_4$, we have

$$\frac{\mathrm{d}T_i(i_4, j_4)}{\mathrm{d}\mathsf{Attn}_i(i_5, j_5)} = 0$$

Then, we consider $i_4 = i_5$ case.

By basic calculus, we have

$$\frac{\mathrm{d}T_i(i_4, j_4)}{\mathrm{d}\mathsf{Attn}_i(i_4, j_5)} = \underbrace{\phi'(\mathsf{Attn}_i(i_4, *)^\top W_g(*, j_4))}_{1 \times 1} \underbrace{W_g(j_5, j_4)}_{1 \times 1}$$

Combining two equations mentioned above, we have the result for **Part 1**.

**Proof of Part 2.**

By result of **Part 1**, for $i_5 = i_4$, we have

$$\frac{\mathrm{d}T_i(i_4, j_4)}{\mathrm{d}\mathsf{Attn}_i(i_4, j_5)} = \underbrace{\phi'(\mathsf{Attn}_i(i_4, *)^\top W_g(*, j_4))}_{1 \times 1} \underbrace{W_g(j_5, j_4)}_{1 \times 1}$$

which implies

$$\frac{\mathrm{d}T_i(i_4, j_4)}{\mathrm{d}\mathsf{Attn}_i(i_4, *)} = \underbrace{\phi'(\mathsf{Attn}_i(i_4, *)^\top W_g(*, j_4))}_{1 \times 1} \underbrace{W_g(*, j_4)}_{d \times 1}$$

By result of **Part 1**, for $i_5 \neq i_4$, we have

$$\frac{\mathrm{d}T_i(i_4, j_4)}{\mathrm{d}\mathsf{Attn}_i(i_5, *)} = 0$$

By basic linear algebra, combining the two equations mentioned above, we have

$$\frac{\mathrm{d}T_i(i_4, j_4)}{\mathrm{d}\mathsf{Attn}_i} = \underbrace{\phi'(\mathsf{Attn}_i(i_4, *)^\top W_g(*, j_4))}_{1 \times 1} \underbrace{e_{i_4}}_{n \times 1} \underbrace{W_g(*, j_4)^\top}_{1 \times d}$$

$\square$

Then, we can argue that the computation for $G_i$ can be done in almost linear time $n^{1+o(1)}$.

**Lemma H.2** (Computation time for $G_i$, formal version of Lemma 4.4). *If we have the below conditions,*

- *Let $G_i \in \mathbb{R}^{n \times d}$ denote the gradient matrix resulting from the application of the chain rule up to the function $g_i$, i.e., $G_i = \frac{\mathrm{d}L(X)}{\mathrm{d}\mathsf{Attn}_i(T_{i-1}(X))}$.*

- *Assuming we already have $\frac{\mathrm{d}L(X)}{\mathrm{d}T_i(X)}$.*

- *Assuming for any $Z \in \mathbb{R}^{n \times d}$, we have $g_i(Z) \in \mathbb{R}^{n \times d}$, and $g_i(Z) = \phi(ZW_g)$, where $W_g \in \mathbb{R}^{d \times d}$ and $\phi : \mathbb{R} \to \mathbb{R}$ denotes any element-wise activation function. Let $\phi'$ denote the derivative of $\phi$.*

- *We simplify the notation, using $T_i$ and $\mathsf{Attn}_i$ to represent $T_i(X)$ and $\mathsf{Attn}_i(T_{i-1}(X))$, respectively.*

- *For any matrix $Z \in \mathbb{R}^{n \times d}$, we use $Z(i, j)$ to denote the $(i, j)$-th entry of $Z$.*

*Then, we can show that $G_i$ can be computed in $n^{1+o(1)}$ time.*

*Proof.* Let $g_{T_i} := \frac{\mathrm{d}L(X)}{\mathrm{d}T_i}$, and for any $i_4 \in [n], j_4 \in [d]$, let $g_{T_i}(i_4, j_4)$ denote the $(i_4, j_4)$-th entry of $g_{T_i}$.

Similarly, for any $i_5 \in [n], j_5 \in [d]$, let $T_i(i_5, j_5)$ denote the $(i_5, j_5)$-th entry of $T_i$.

We can have

$$
\begin{aligned}
G_i &= \frac{\mathrm{d}L(X)}{\mathrm{d}\mathsf{Attn}_i} \\
&= \frac{\mathrm{d}L(X)}{\mathrm{d}T_i} \cdot \frac{\mathrm{d}T_i}{\mathrm{d}\mathsf{Attn}_i} \\
&= g_{T_i} \cdot \frac{\mathrm{d}T_i}{\mathrm{d}\mathsf{Attn}_i} \\
&= \sum_{i_4=1}^{n} \sum_{j_4=1}^{d} g_{T_i}(i_4, j_4) \cdot \frac{\mathrm{d}T_i(i_4, j_4)}{\mathrm{d}\mathsf{Attn}_i}
\end{aligned}
$$

where the 1st step is from the definition of $G_i$, the 2nd step comes from chain rule, the 3rd step is because of the definition of $g_{T_i}$, the 4th step is due to chain rule.

$$
\begin{aligned}
&\sum_{i_4=1}^{n} \sum_{j_4=1}^{d} g_{T_i}(i_4, j_4) \cdot \frac{\mathrm{d}T_i(i_4, j_4)}{\mathrm{d}\mathsf{Attn}_i} \\
&= \sum_{i_4=1}^{n} \sum_{j_4=1}^{d} \underbrace{g_{T_i}(i_4, j_4)}_{1 \times 1} \underbrace{\phi'(\mathsf{Attn}_i(i_4, *)^\top W_g(*, j_4))}_{1 \times 1} \underbrace{e_{i_4}}_{n \times 1} \underbrace{W_g(*, j_4)^\top}_{1 \times d} \\
&= \sum_{i_4=1}^{n} \underbrace{e_{i_4}}_{n \times 1} \sum_{j_4=1}^{d} \underbrace{g_{T_i}(i_4, j_4)}_{1 \times 1} \underbrace{\phi'(\mathsf{Attn}_i(i_4, *)^\top W_g(*, j_4))}_{1 \times 1} \underbrace{W_g(*, j_4)^\top}_{1 \times d} \\
&= \sum_{i_4=1}^{n} \underbrace{e_{i_4}}_{n \times 1} (\underbrace{W_g}_{d \times d} (\underbrace{g_{T_i}(i_4, *)}_{d \times 1} \odot \underbrace{\phi'(\mathsf{Attn}_i(i_4, *)^\top W_g)}_{d \times 1}))^\top \\
&= \underbrace{(g_{T_i} \odot \phi'(\mathsf{Attn}_i W_g))}_{n \times d} \underbrace{W_g^\top}_{d \times d}
\end{aligned} \tag{25}
$$

where the 1st step is from Lemma H.1, the 2nd step comes from basic algebra, the 3rd step is because of basic linear algebra, the 4th step is due to basic linear algebra.

By Eq. (25), we have the close form of $G_i$.

We can compute $G_i$ in the following order

- Compute $\underbrace{(g_{T_i} \odot \phi'(\mathsf{Attn}_i W_g))}_{n \times d}$, which takes $n \cdot d$ time.

- Compute $\underbrace{(g_{T_i} \odot \phi'(\mathsf{Attn}_i W_g))}_{n \times d} \underbrace{W_g^\top}_{d \times d}$, which takes $d^2 \cdot n$ time.

Therefore, the overall running time for $G_i$ is $n^{1+o(1)}$.

$\square$

## H.2   FAST COMPUTATION FOR SINGLE-LAYER TRANSFORMER

In this section, we dive into the computation time and approximation error of the gradient of a single-layer transformer. We demonstrate in the following Lemma that the gradient of a single-layer transformer can be computed in almost linear time $n^{1+o(1)}$, and its error can be bounded by $1/\operatorname{poly}(n)$.

**Lemma H.3** (Single-layer transformer gradient approximation). *If we have the below conditions,*

- *Let $L(X)$ be defined as Definition 2.1.*

- *Let $X$ be defined as Definition C.3.*

- *Let the gradient matrix $G_i \in \mathbb{R}^{n \times d}$ be defined as $G_i = \frac{\mathrm{d}L(X)}{\mathrm{d}\mathsf{Attn}_i(T_{i-1}(X))}$.*

- *For $i_2 \in [n], j_2 \in [d]$, let $G_i(i_2, j_2)$ denote the $(i_2, j_2)$-th entry of $G_i$.*

- *Assuming for any $Z \in \mathbb{R}^{n \times d}$, we have $g_i(Z) \in \mathbb{R}^{n \times d}$, and $g_i(Z) = \phi(Z \cdot W_g)$, where $W_g \in \mathbb{R}^{d \times d}$ and $\phi : \mathbb{R} \to \mathbb{R}$ denotes any element-wise activation function. Let $\phi'$ denote the derivative of $\phi$.*

- *Suppose we have a single-layer transformer (see Definition 1.3).*

*Then, we can show that,*

- **Part 1: running time.** *Our algorithm can approximate $\frac{\mathrm{d}L(X)}{\mathrm{d}X}$ in $n^{1+o(1)}$ time.*

- **Part 2: error bound.** *The approximation error of the single-layer transformer can be bounded by $1/\operatorname{poly}(n)$. Namely, our algorithm output $\widetilde{g}_1$ satisfies*

$$\|\widetilde{g}_1 - \frac{\mathrm{d}L(X)}{\mathrm{d}X}\|_\infty \leq 1/\operatorname{poly}(n)$$

*Proof.* By Definition 1.3, a single-layer transformer has following structure:

$$g_1 \circ \mathsf{Attn}_1 \circ g_0(X)$$

By the definition of $G_i$, we have

$$
\begin{aligned}
G_1 &= \frac{\mathrm{d}L(X)}{\mathrm{d}\mathsf{Attn}_1(T_0(X))} \\
&= \frac{\mathrm{d}L(X)}{\mathrm{d}T_1(X)} \cdot \frac{\mathrm{d}T_1(X)}{\mathrm{d}\mathsf{Attn}_1(T_0(X))}
\end{aligned}
\tag{26}
$$

By Lemma H.2, we have $G_1$ can be computed in $n^{1+o(1)}$ time.

**Proof of Part 1: running time.**

For less confusion, in this part of the proof, we ignore the approximation error temporarily.

Since we have got $G_1$, we use methods mentioned in Lemma E.11, F.5, G.4 to compute $\frac{\mathrm{d}L(X)}{\mathrm{d}T_0(X)}, \frac{\mathrm{d}L(X)}{\mathrm{d}W_1}, \frac{\mathrm{d}L(X)}{\mathrm{d}W_{V_1}}$, respectively, which takes $n^{1+o(1)}$ time for each.

Then, since we have $\frac{\mathrm{d}L(X)}{\mathrm{d}T_0(X)}$, again by Lemma H.2, we have $\frac{\mathrm{d}L(X)}{\mathrm{d}X}$ can be computed in $n^{1+o(1)}$ time.

Therefore, the overall running time is $n^{1+o(1)}$.

**Proof of Part 2: error bound.**

Then, we move on to the error bound.

By Lemma H.2 and Eq. (26), there is no approximation error when computing $G_1$.

By Lemma E.11, F.5, G.4, we have there is $1/\operatorname{poly}(n)$ approximation error on $\frac{\mathrm{d}L(X)}{\mathrm{d}T_0(X)}, \frac{\mathrm{d}L(X)}{\mathrm{d}W_1}, \frac{\mathrm{d}L(X)}{\mathrm{d}W_{V_1}}$, respectively.

Let $\widetilde{g}_{t_0}, \widetilde{g}_{w_1}, \widetilde{g}_{v_1}$ denote the approximation results of $\frac{\mathrm{d}L(X)}{\mathrm{d}T_0(X)}, \frac{\mathrm{d}L(X)}{\mathrm{d}W_1}, \frac{\mathrm{d}L(X)}{\mathrm{d}W_{V_1}}$, respectively.

We have

$$\|\widetilde{g}_{t_0} - \frac{\mathrm{d}L(X)}{\mathrm{d}T_0(X)}\|_\infty \leq 1/\operatorname{poly}(n) \tag{27}$$

and

$$\|\widetilde{g}_{w_1} - \frac{\mathrm{d}L(X)}{\mathrm{d}W_1}\|_\infty \leq 1/\operatorname{poly}(n)$$

and

$$\|\widetilde{g}_{v_1} - \frac{\mathrm{d}L(X)}{\mathrm{d}W_{V_1}}\|_\infty \leq 1/\operatorname{poly}(n)$$

Let $\widetilde{G}_0 = \widetilde{g}_{t_0} \cdot \frac{\mathrm{d}T_0(X)}{\mathrm{d}X}$ denote the approximated version of $G_0$.

We have

$$\|\widetilde{G}_0 - G_0\|_\infty$$
$$= \|(\widetilde{g}_{t_0} - \frac{\mathrm{d}L(X)}{\mathrm{d}T_0(X)}) \cdot \frac{\mathrm{d}T_0(X)}{\mathrm{d}X}\|_\infty$$
$$\leq n \cdot d\|\widetilde{g}_{t_0} - \frac{\mathrm{d}L(X)}{\mathrm{d}T_0(X)}\|_\infty \|\frac{\mathrm{d}T_0(X)}{\mathrm{d}X}\|_\infty$$
$$\leq n \cdot d(1/\operatorname{poly}(n))\|\frac{\mathrm{d}T_0(X)}{\mathrm{d}X}\|_\infty$$
$$\leq 1/\operatorname{poly}(n)$$

where the 1st step is from the definition of $\widetilde{G}_0$, the 2nd step comes from basic linear algebra, the 3rd step is because of Eq. (27), the 4th step is due to each entry can be written by $O(\log n)$ bits.

Let $\widetilde{g}_1 = \widetilde{G}_0$.

Therefore, we have

$$\|\widetilde{g}_1 - \frac{\mathrm{d}L(X)}{\mathrm{d}X}\|_\infty \leq 1/\operatorname{poly}(n)$$

$\square$

### H.3 FAST COMPUTATION FOR MULTI-LAYER TRANSFORMER

Since we have already demonstrated that almost linear time gradient computation can be applied to a single-layer transformer, with the help of math induction, we can easily generalize that result to the multi-layer transformer. In the following Lemma, we display that the gradient of the multi-layer transformer can be computed in almost linear time, and its approximation error can be bounded by $1/\operatorname{poly}(n)$.

**Lemma H.4** (Multi-layer transformer gradient approximation, formal version of Lemma 4.5). *If we have the below conditions,*

- *Let $L(X)$ be defined as Definition 2.1.*

- *Let $X$ be defined as Definition C.3.*

- *Let $G_i \in \mathbb{R}^{n \times d}$ denote the gradient matrix resulting from the application of the chain rule up to the function $g_i$, i.e., $G_i = \frac{\mathrm{d}L(X)}{\mathrm{d}\mathrm{Attn}_i(T_{i-1}(X))}$.*

- *For $i_2 \in [n], j_2 \in [d]$, let $G_i(i_2, j_2)$ denote the $(i_2, j_2)$-th entry of $G_i$.*

- *Let gradient components for each layer be computed according to Lemma E.11, F.5, G.4.*

- *Assuming for any $Z \in \mathbb{R}^{n \times d}$, we have $g_i(Z) \in \mathbb{R}^{n \times d}$, and $g_i(Z) = \phi(Z \cdot W_g)$, where $W_g \in \mathbb{R}^{d \times d}$ and $\phi : \mathbb{R} \to \mathbb{R}$ denotes any element-wise activation function. Let $\phi'$ denote the derivative of $\phi$.*

- *Suppose we have a $m$-layer transformer (see Definition 1.3).*

*Then, we can show that,*

- **Part 1: running time.** *Our algorithm can approximate $\frac{\mathrm{d}L(X)}{\mathrm{d}X}$ in $n^{1+o(1)}$ time.*

- **Part 2: error bound.** *The approximation error of the multi-layer transformer can be bounded by $1/\mathrm{poly}(n)$. Namely, our algorithm output $\widetilde{g}$ satisfies*

$$\|\widetilde{g} - \frac{\mathrm{d}L(X)}{\mathrm{d}X}\|_\infty \leq 1/\mathrm{poly}(n)$$

*Proof.* We use math induction to prove this Lemma.

**Step 1: Proof of a single-layer transformer.**

Firstly, by Lemma H.3, we have that for one-layer transformer, our conclusion is established.

**Step 2: Assumption for $k$-layer transformer.**

Secondly, we assume for any $k$, for $k$-layer transformer model, we have

- Our algorithm can approximate $\frac{\mathrm{d}L(X)}{\mathrm{d}X}$ in $n^{1+o(1)}$ time.

- The approximation error of the $k$-layer transformer can be bounded by $1/\mathrm{poly}(n)$. Namely, our algorithm output $\widetilde{g}$ satisfies

$$\|\widetilde{g} - \frac{\mathrm{d}L(X)}{\mathrm{d}X}\|_\infty \leq 1/\mathrm{poly}(n)$$

**Step 3: Proof of $(k+1)$-layer transformer.**

Thirdly, we consider the $(k + 1)$-layer transformer model.

Without loss of generality, we assume that the additional transformer layer is added at the beginning of the model.

Namely, let $\mathsf{F}_k$ denote a $k$-layer transformer model. We have

$$\mathsf{F}_k(X) = g_k \circ \mathsf{Attn}_k \circ \cdots \circ g_1 \circ \mathsf{Attn}_1 \circ g_0(X)$$

Let the $(k + 1)$-layer transformer model have the following structure:

$$\mathsf{F}_{k+1}(X) = \mathsf{F}_k \circ \mathsf{Attn} \circ g(X) \tag{28}$$

Let $T_0 := g(X)$.

By assumption, we have

- $\frac{\mathrm{d}L(X)}{\mathrm{d}\mathsf{Attn}(T_0)}$ can be approximated in $n^{1+o(1)}$ time.

- Let $\widetilde{g}_k$ denote the approximated version of $\frac{\mathrm{d}L(X)}{\mathrm{d}\mathsf{Attn}(T_0)}$. We have

$$\|\widetilde{g}_k - \frac{\mathrm{d}L(X)}{\mathrm{d}\mathsf{Attn}(T_0)}\|_\infty \leq 1/\mathrm{poly}(n) \tag{29}$$

**Step 3.1: Proof of the running time for $(k + 1)$-layer transformer**

For less confusion, in this part of the proof, we ignore the approximation error temporarily.

By the assumption, we have $\frac{\mathrm{d}L(X)}{\mathrm{d}\mathsf{Attn}(T_0)}$ can be approximated in $n^{1+o(1)}$ time.

We compute $\frac{\mathrm{d}L(X)}{\mathrm{d}X}$ in following order:

- Since we already have $\frac{\mathrm{d}L(X)}{\mathrm{d}\mathsf{Attn}(T_0)}$, by Lemma E.11, the computation time for $\frac{\mathrm{d}L(X)}{\mathrm{d}T_0}$ is $n^{1+o(1)}$.

- Since we have $\frac{\mathrm{d}L(X)}{\mathrm{d}T_0}$, by Lemma H.2, the computation time for $\frac{\mathrm{d}L(X)}{\mathrm{d}X}$ is $n^{1+o(1)}$.

Therefore, for $(k + 1)$-layer transformer, the overall running time for $\frac{\mathrm{d}L(X)}{\mathrm{d}X}$ is $n^{1+o(1)}$.

**Step 3.2: Proof of the error bound for $(k + 1)$-layer transformer**

By Lemma E.11, during the process of solving the approximated version of $\frac{\mathrm{d}L(X)}{\mathrm{d}g(X)}$, the approximation error will not be magnified by more than $\mathrm{poly}(n)$.

Let $\widetilde{g}_{t_0}$ denote the approximated version of $\frac{\mathrm{d}L(X)}{\mathrm{d}g(X)}$, we have

$$\|\widetilde{g}_{t_0} - \frac{\mathrm{d}L(X)}{\mathrm{d}g(X)}\|_\infty \le \mathrm{poly}(n)\|\widetilde{g}_k - \frac{\mathrm{d}L(X)}{\mathrm{d}T(X)}\|_\infty$$
$$\le 1/\mathrm{poly}(n) \tag{30}$$

where the 1st step is from the above statement, the 2nd step comes from Eq. (29), the 3rd step is because of basic algebra.

Then, we consider

$$\frac{\mathrm{d}L(X)}{\mathrm{d}X} = \frac{\mathrm{d}L(X)}{\mathrm{d}g(X)} \cdot \frac{\mathrm{d}g(X)}{\mathrm{d}X} \tag{31}$$

Recall that we have $\widetilde{g} = \frac{\mathrm{d}L(X)}{\mathrm{d}X}$. Then, we have

$$\|\widetilde{g} - \frac{\mathrm{d}L(X)}{\mathrm{d}X}\|_\infty = \|(\widetilde{g}_{t_0} - \frac{\mathrm{d}L(X)}{\mathrm{d}g(X)}) \cdot \frac{\mathrm{d}g(X)}{\mathrm{d}X}\|_\infty$$
$$\le n \cdot d\|\widetilde{g}_{t_0} - \frac{\mathrm{d}L(X)}{\mathrm{d}g(X)}\|_\infty \|\frac{\mathrm{d}g(X)}{\mathrm{d}X}\|_\infty$$
$$\le n \cdot d(1/\mathrm{poly}(n))\|\frac{\mathrm{d}g(X)}{\mathrm{d}X}\|_\infty$$
$$\le 1/\mathrm{poly}(n)$$

where the 1st step is from Eq. (31), the 2nd step comes from basic linear algebra, the 3rd step is because of Eq. (30), the 4th step is due to each entry can be written by $O(\log n)$ bits.

**Step 4: Use math induction.**

So far, with the assumption that our statement holds under $k$-layer transformer, we have proved that our statement still holds under $(k + 1)$-layer transformer.

Therefore, by math induction, our statement holds for any $m$-layer transformer.

$\square$

## I  CAUSAL ATTENTION MASK

This section will discuss how to combine the causal attention mask with our framework. We argue that even with the causal attention mask, we can also achieve almost linear time gradient computing for the multi-layer transformer.

In Section I.1, we introduce essential tools from literature to deal with the causal mask added on the attention matrix. In Section I.2, we show that with the addition of causal mask, our framework can still achieve almost linear time gradient computation.

### I.1    TOOLS FROM PREVIOUS WORK

Firstly, we restate a classical low-rank approximation method in the literature.

**Lemma I.1** (Low-rank approximation, (Alman & Song, 2023)). *Suppose* $Q, K \in \mathbb{R}^{n \times d}$, *with* $\|Q\|_\infty \leq R$, *and* $\|K\|_\infty \leq R$. *Let* $A := \exp(QK^\top/d) \in \mathbb{R}^{n \times n}$. *For accuracy parameter* $\epsilon \in (0, 1)$, *there is a positive integer $g$ bounded above by*

$$g = O\Big( \max \Big\{ \frac{\log(1/\epsilon)}{\log(\log(1/\epsilon)/R)}, R^2 \Big\} \Big),$$

*and a positive integer $r$ bounded above by*

$$r \leq \binom{2(g+d)}{2g}$$

*such that: There is a matrix* $\widetilde{A} \in \mathbb{R}^{n \times n}$ *that is an $(\epsilon, r)$-approximation of* $A \in \mathbb{R}^{n \times n}$. *Furthermore, the matrices $U_0$ and $V_0$ defining $\widetilde{A}$ can be computed in $O(n \cdot r)$ time.*

Then, we provide the formal definition for the causal attention mask.

**Definition I.2** (Causal attention mask, (Liang et al., 2024a)). *We define the causal attention mask as* $M \in \{0, 1\}^{n \times n}$, *where* $M_{i,j} = 1$ *if* $i \geq j$ *and* $M_{i,j} = 0$ *otherwise.*

---

**Algorithm 2** Causal attention mask algorithm, Algorithm 4 in (Liang et al., 2024a)

1: **procedure** CAUSALMASK($U_0 \in \mathbb{R}^{n \times k}, V_0 \in \mathbb{R}^{n \times k}, v \in \mathbb{R}^n$)                    ▷ Lemma I.3
2:        $c_0 \leftarrow \mathbf{0}_k$
3:        **for** $j = 1 \rightarrow n$ **do**
4:            $b_j \leftarrow \underbrace{(V_0^\top)_j}_{k \times 1} \underbrace{v_j}_{\text{scalar}}$                    ▷ Let $(V_0^\top)_j$ denote the $j$-th row of $V_0 \in \mathbb{R}^{n \times k}$
5:            $c_j \leftarrow \underbrace{c_{j-1}}_{k \times 1} + \underbrace{b_j}_{k \times 1}$
6:        **end for**
7:        **for** $j = 1 \rightarrow n$ **do**
8:            $Y_j \leftarrow \langle \underbrace{(U_0^\top)_j}_{k \times 1}, \underbrace{c_j}_{k \times 1} \rangle$
9:        **end for**
10: **return** $Y$                    ▷ $Y \in \mathbb{R}^n$
11: **end procedure**

---

In previous work (Liang et al., 2024a), they point out there exists an algorithm (Algorithm 2) that can calculate low-rank matrices (with the causal attention mask) multiplication with any vector $v$ in almost linear time. We restate their results in Lemma I.3.

**Lemma I.3** (Fast computation for causal attention mask on tensor, (Liang et al., 2024a)). *Let* $M \in \{0, 1\}^{n \times n}$ *be a causal attention mask defined in Definition I.2. Let* $U_0, V_0 \in \mathbb{R}^{n \times k}$. *Let* $v \in \mathbb{R}^n$. *Then, there exists an algorithm (see Algorithm 2) whose output satisfies that*

$$Y = (M \odot (U_0 V_0^\top))v,$$

*which takes $O(nk)$ time.*

We extend their results to the multiplication of matrix with $n^{o(1)}$ columns.

**Lemma I.4** (Fast computation for causal attention mask on matrix). *If we have the below conditions,*

- *Let* $M \in \{0, 1\}^{n \times n}$ *be a causal attention mask defined in Definition I.2.*

- *Let $U_0, V_0 \in \mathbb{R}^{n \times k}$ where $k = n^{o(1)}$.*

- *Let $H \in \mathbb{R}^{n \times k_H}$ where $k_H = n^{o(1)}$.*

*Then, there exists an algorithm, whose output satisfies that*

$$Z = (M \odot (U_0 V_0^\top))H,$$

*which takes $n^{1+o(1)}$ time.*

*Proof.* For $j \in [k_H]$, let $H_{*,j} \in \mathbb{R}^n$ denote the $j$-th column of $H$.

By Lemma I.3, we can compute $(M \odot (U_0 V_0^\top))H_{*,j}$ in $O(nk)$ time.

There are $k_H$ columns in total. Therefore, the overall running time is $O(nkk_H) = O(n \cdot n^{o(1)} \cdot n^{o(1)}) = n^{1+o(1)}$. $\qquad\square$

## I.2  FAST COMPUTATION WITH CAUSAL MASK

We can easily change all low-rank matrices multiplication to the algorithm mentioned in Lemma I.4. Then, our framework can support the causal attention mask and still achieves almost linear time gradient computing for the multi-layer transformer.

The causal mask directly affects the attention matrix, so it's necessary to define the attention matrix with the causal mask applied.

**Definition I.5.** *Let $M \in \{0,1\}^{n \times n}$ be a causal attention mask defined in Definition I.2. We define attention matrix with causal mask as:*

$$\widehat{f}(X) := D^{-1}(M \odot A)$$

*where $A := \exp(XWX^\top/d)$ and $D := \mathrm{diag}((M \odot A) \cdot \mathbf{1}_n)$.*

After analyzing the components of gradients on $T_i(X), W_i, W_{V_i}$ in Section E, F and G, we categorize them into two groups: one involving the dot product and the other involving the Hadamard product of the attention matrix. Then, we can show $\widehat{f}(X)H$ and $(\widehat{f}(X) \odot (UV^\top))H$ for low rank matrices $U, V, H$ can be approximated in almost linear time.

**Lemma I.6.** *If we have the below conditions,*

- *Let $\widehat{f}(X)$ be defined in Definition I.5.*

- *Let $U, V \in \mathbb{R}^{n \times k}$ where $k = n^{o(1)}$.*

- *Let $H \in \mathbb{R}^{n \times k_H}$ where $k_H = n^{o(1)}$.*

*Then, approximating the following takes $n^{1+o(1)}$ time:*

- *Part 1. $\widehat{f}(X)H$*

- *Part 2. $(\widehat{f}(X) \odot (UV^\top))H$*

*Proof.* From Definition I.5, we know

$$\widehat{f}(X) := D^{-1}(M \odot A)$$

where $D := \mathrm{diag}((M \odot A) \cdot \mathbf{1}_n)$.

By Lemma I.1, $U_0 V_0^\top$ is a good approximation for $A$. Then, we can approximate $\widehat{f}(X)$ by:

$$D^{-1}(M \odot (U_0 V_0^\top))$$

where $D := \mathrm{diag}((M \odot (U_0 V_0^\top)) \cdot \mathbf{1}_n)$.

Using Lemma I.3, we know $(M \odot (U_0 V_0^\top)) \cdot v$ for any vector $v \in \mathbb{R}^n$ can be computed in almost linear time.

We begin by examining the normalization matrix $D^{-1}$. Calling Lemma I.3, we compute $(M \odot (U_0 V_0^\top)) \cdot \mathbf{1}_n$ in almost linear time. Then, it takes $O(n)$ time to make $(M \odot (U_0 V_0^\top)) \cdot \mathbf{1}_n$ diagonal. Given that $D$ is diagonal, its inverse $D^{-1}$ can be determined in $O(n)$ time. Thus, we can compute $D^{-1}$ in almost linear time.

**Proof of Part 1.** $H$ can be viewed as a combination of $k_H$ vectors, each of size $n$. Calling Lemma I.4, we can compute $(M \odot (U_0 V_0^\top))H$ in $n^{1+o(1)}$ time.

Finally, we compute $\underbrace{D^{-1}}_{n \times n} \underbrace{(M \odot (U_0 V_0^\top))H}_{n \times k_H}$, which takes $n^{1+o(1)}$ time since $D^{-1}$ is diagonal. The

overall gradient computation remains $n^{1+o(1)}$ time.

**Proof of Part 2.** The proof for this part involves Fact C.2. We can show

$$
\begin{aligned}
((D^{-1}(M \odot (U_0 V_0^\top))) \odot (UV^\top))H &= ((M \odot (D^{-1}U_0 V_0^\top)) \odot (UV^\top))H \\
&= (M \odot ((D^{-1}U_0 V_0^\top) \odot (UV^\top)))H \\
&= (M \odot ((D^{-1}U_0) \oslash U)(V_0 \oslash V)^\top)H
\end{aligned}
$$

where the 1st step is from $D(A \odot B) = (DA) \odot B = A \odot (DB)$ for diagonal matrix $D \in \mathbb{R}^{m \times m}$ and $A, B \in \mathbb{R}^{m \times n}$, the 2nd step comes from $(A \odot B) \odot C = A \odot (B \odot C)$ for $A, B, C \in \mathbb{R}^{m \times n}$, and the last step follows from Fact C.2.

Let $U_M := (D^{-1}U_0) \oslash U$ and $V_M := V_0 \oslash V$.

For $U_M$, we compute $\underbrace{D^{-1}}_{n \times n} \underbrace{U_0}_{n \times k}$ which takes $nk$ time. We then compute $\underbrace{(D^{-1}U_0)}_{n \times k} \oslash \underbrace{U}_{n \times k}$ which takes $O(nk^2)$ time.

For $V_M$, we compute $\underbrace{V_0}_{n \times k} \oslash \underbrace{V}_{n \times k}$ which takes $O(nk^2)$ time.

We now have $(M \odot (U_M V_M^\top))H$. Calling Lemma I.4, we finish the proof. $\square$

We now prove for gradient components that have dot product.

**Lemma I.7** (Components for dot product). *If we have the below conditions,*

- *Let $\widehat{f}(X)$ be defined in Definition I.5.*

- *Let $G_i \in \mathbb{R}^{n \times d}$ denote the gradient matrix resulting from the application of the chain rule up to the function $g_i$, i.e., $G_i = \frac{\mathrm{d}L(X)}{\mathrm{d}\mathsf{Attn}_i(T_{i-1}(X))}$.*

- *Let $D_6 = -f(X) \operatorname{diag}(K) XW^\top$ be defined in Lemma D.17.*

- *Let $D_2 = -\operatorname{diag}(K) f(X) XW$ be defined in Lemma D.17.*

- *Let $D_8 = f(X) G_i W_V^\top$ be defined in Lemma D.17.*

- *Let $g_v := X^\top f(X) G_i$ be the gradient on $W_{V_i}$ and defined in Lemma G.3.*

*Then, we can show the following can be approximated in almost linear time:*

- *Part 1. $\widehat{D}_6 = -\widehat{f}(X) \operatorname{diag}(K) XW^\top$*

- *Part 2. $\widehat{D}_2 = -\operatorname{diag}(K) \widehat{f}(X) XW$*

- *Part 3. $\widehat{D}_8 = \widehat{f}(X) G_i W_V^\top$*

- *Part 4. $\widehat{g}_v := X^\top \widehat{f}(X) G_i$*

*Proof.* **Proof of Part 1.** For $\widehat{D}_6$, we compute $\underbrace{\text{diag}(K)}_{n \times n} \underbrace{X}_{n \times d}$ first, which takes $nd$ time.

Then, we compute $\underbrace{\widehat{f}(X)}_{n \times n} \underbrace{\text{diag}(K)X}_{n \times d}$ using **Part 1.** of Lemma I.6, which takes $n^{1+o(1)}$ time.

Finally, we compute $\underbrace{\widehat{f}(X)\text{diag}(K)X}_{n \times d} \underbrace{W^\top}_{d \times d}$, which takes $n^{1+o(1)}$ time.

**Proof of Part 2.** For $\widehat{D}_2$, we compute $\underbrace{\widehat{f}(X)}_{n \times n} \underbrace{X}_{n \times d}$ using **Part 1.** of Lemma I.6, which takes $n^{1+o(1)}$ time.

Then, we compute $\underbrace{\text{diag}(K)}_{n \times n} \underbrace{\widehat{f}(X)X}_{n \times d}$, which takes $nd$ time.

After that, we compute $\underbrace{\text{diag}(K)\widehat{f}(X)X}_{n \times d} \underbrace{W}_{d \times d}$, which takes $n^{1+o(1)}$ time.

**Proof of Part 3.** For $\widehat{D}_8$, we compute in the following steps:

We compute $\underbrace{\widehat{f}(X)}_{n \times n} \underbrace{G_i}_{n \times d}$ using **Part 1.** of Lemma I.6, which takes $n^{1+o(1)}$ time.

Then, we compute $\underbrace{\widehat{f}(X)G_i}_{n \times d} \underbrace{W_V^\top}_{d \times d}$, which takes $n \cdot d^2$ time.

**Proof of Part 4.** For $\widehat{g}_v$, we compute in the following steps:

We compute $\underbrace{\widehat{f}(X)}_{n \times n} \underbrace{G_i}_{n \times d}$ using **Part 1.** of Lemma I.6, which takes $n^{1+o(1)}$ time.

Then, we compute $\underbrace{X^\top}_{d \times n} \underbrace{\widehat{f}(X)G_i}_{n \times d}$, which takes $n \cdot d^2$ time. $\qquad\square$

We then prove for gradient components that have Hadamard product.

**Lemma I.8** (Components for Hadamard product)**.** *If we have the below conditions,*

- *Let $\widehat{f}(X)$ be defined in Definition I.5.*

- *Let $G_i \in \mathbb{R}^{n \times d}$ denote the gradient matrix resulting from the application of the chain rule up to the function $g_i$, i.e., $G_i = \frac{\text{d}L(X)}{\text{dAttn}_i(T_{i-1}(X))}$.*

- *Let $D_7 = (f(X) \odot (h(X)G_i^\top))XW^\top$ be defined in Lemma D.17.*

- *Let $D_4 = (f(X) \odot (G_i h(X)^\top))XW$ be defined in Lemma D.17.*

- *Let $g_w := X^\top p(X)X = X^\top(p_1(X) - p_2(X))X$ be the gradient on $W_i$ and defined in Definition C.12 and Lemma F.5 where $p_1(X) = f(X) \odot q(X)$ and $p_2(X) = \text{diag}(p_1(X) \cdot \mathbf{1}_n)f(X)$.*

*Then, we can show the following can be approximated in almost linear time:*

- *Part 1. $\widehat{D}_7 = (\widehat{f}(X) \odot (h(X)G_i^\top))XW^\top$*

- *Part 2. $\widehat{D}_4 = (\widehat{f}(X) \odot (G_i h(X)^\top))XW$*

- *Part 3. $\widehat{g}_w := X^\top(\widehat{p}_1(X) - \widehat{p}_2(X))X$ where $\widehat{p}_1(X) = \widehat{f}(X) \odot q(X)$ and $p_2(X) = \text{diag}(\widehat{p}_1(X) \cdot \mathbf{1}_n)\widehat{f}(X)$.*

*Proof.* **Proof of Part 1.** For $\widehat{D}_7$, we can compute $\underbrace{(\widehat{f}(X) \odot (h(X)G_i^\top))}_{n \times n} \underbrace{X}_{n \times d}$ using **Part 2.** of Lemma I.6, which takes $n^{1+o(1)}$ time.

We then compute $\underbrace{(\widehat{f}(X) \odot (h(X)G_i^\top))X}_{n \times d} \underbrace{W^\top}_{d \times d}$, which takes $nd^2$ time.

**Proof of Part 2.** For $\widehat{D}_7$, we can compute $\underbrace{(\widehat{f}(X) \odot (G_i h(X)^\top))}_{n \times n} \underbrace{X}_{n \times d}$ using **Part 2.** of Lemma I.6, which takes $n^{1+o(1)}$ time.

We then compute $\underbrace{(\widehat{f}(X) \odot (G_i h(X)^\top))X}_{n \times d} \underbrace{W}_{d \times d}$, which takes $nd^2$ time.

**Proof of Part 3.** For $\widehat{g}_w$, we consider $X^\top \widehat{p}_1(X)X$ first. Based on Definition C.11, we have $\widehat{p}_1(X) = \widehat{f}(X) \odot q(X) = \widehat{f}(X) \odot (G_i h(X)^\top)$. We then compute $(\widehat{f}(X) \odot (G_i h(X)^\top))X$ using **Part 2.** of Lemma I.6, which takes $n^{1+o(1)}$ time. After that, we compute $\underbrace{X^\top}_{d \times n} \underbrace{(\widehat{f}(X) \odot (G_i h(X)^\top))X}_{n \times d}$, which takes $nd^2$ time.

Now we consider $X^\top \widehat{p}_2(X)X$. By definition, $\widehat{p}_2(X) = \mathrm{diag}(\widehat{p}_1(X) \cdot \mathbf{1}_n)\widehat{f}(X)$. We first compute $\widehat{p}_1(X) \cdot \mathbf{1}_n = (\widehat{f}(X) \odot (G_i h(X)^\top)) \cdot \mathbf{1}_n$ using **Part 2.** of Lemma I.6, which takes $n^{1+o(1)}$ time. Meanwhile, we compute $\widehat{f}(X)X$ using **Part 1.** of Lemma I.6, which takes $n^{1+o(1)}$ time. We then have $\underbrace{\mathrm{diag}(\widehat{p}_1(X) \cdot \mathbf{1}_n)}_{n \times n} \underbrace{\widehat{f}(X)X}_{n \times d}$, which takes $nd$ time. Finally, we compute $\underbrace{X^\top}_{d \times n} \underbrace{\mathrm{diag}(\widehat{p}_1(X) \cdot \mathbf{1}_n)\widehat{f}(X)X}_{n \times d}$, which takes $nd^2$ time.

Together, $\underbrace{X^\top \widehat{p}_1(X)X}_{d \times d} - \underbrace{X^\top \widehat{p}_2(X)X}_{d \times d}$ takes $d^2$ time. $\qquad\square$

Thus, we show that our framework can support causal attention masks.

## J  RESIDUAL CONNECTION

In this section, we discuss how to adapt our framework to the attention mechanism with the residual connection.

In Section J.1, we provide a formalized definition of the two residual connections used in the attention mechanism. In Section J.2, we argue that with the addition of the residual connection, the gradient over the attention mechanism can be computed in almost linear time $n^{1+o(1)}$ and the approximation error can be bound by $1/\mathrm{poly}(n)$. In Section J.3, we use math induction to show that the gradient over the entire transformer with the residual connection can also be computed in almost linear time $n^{1+o(1)}$.

### J.1  KEY CONCEPTS

Recall that in Definition 2.3, we have defined $T_i(X) \in \mathbb{R}^{n \times d}$ as the intermediate variable output by the $i$-th transformer layer. For simplicity, we use $T_i$ to represent $T_i(X)$ in the rest part of this section. Namely, we have

$$T_i = (g_i \circ \mathsf{Attn}_i)(T_{i-1})$$

Then, we consider adding the residual connection to our framework. Note that there are two residual connection operations in one transformer layer. We first define the residual connection over the $\mathsf{Attn}_i$ in Definition J.1.

**Definition J.1** (Residual connection over $\mathsf{Attn}_i$). *If we have the below conditions,*

- *Let $T_i$ be defined as Definition 2.3.*

- *Let $\mathsf{Attn}_i$ be defined as Definition C.3.*

*We define $Z_i \in \mathbb{R}^{n \times d}$ as the output with the residual connection of $\mathsf{Attn}_i$. Namely, we have*

$$Z_i = T_{i-1} + \mathsf{Attn}_i(T_{i-1})$$

Then, we consider the second residual connection over the MLP layer $g_i$, where we have the formal definition for this in Definition J.2.

**Definition J.2** (Residual connection over $g_i$). *If we have the below conditions,*

- *Let the multi-layer transformer be defined as Definition 1.3.*

- *Let the intermediate variable $T_i$ be defined as Definition 2.3.*

- *Let $g_i$ denote the components other than self-attention in the $i$-th transformer layer.*

- *Let $Z_i \in \mathbb{R}^{n \times d}$ be defined as Definition J.1.*

*Then $T_i$, the output of $i$-th layer transformer with the residual connection, should have the following form:*

$$T_i = Z_i + g_i(Z_i)$$

### J.2 ANALYSIS OF THE RESIDUAL CONNECTION

In the previous section, we have defined the two residual connection operations.

In this section, we argue that if the gradient computation can be done in almost linear time without the residual connection, then with the addition of the residual connection, the gradient computation can also be completed in almost linear time.

**Lemma J.3** (Analysis of the residual connection). *If we have the below conditions,*

- *Let $L(X)$ be defined as Definition 2.1.*

- *Let $Y_R \in \mathbb{R}^{n \times d}$ and $X_R \in \mathbb{R}^{n \times d}$ denote the output and input of the residual connection, respectively.*

- *Let $\mathsf{H} : \mathbb{R}^{n \times d} \to \mathbb{R}^{n \times d}$ denote some layer in the transformer, such as MLP, $\mathsf{Attn}$, etc.*

- *Suppose the residual connection can be written as*

$$Y_R = X_R + \mathsf{H}(X_R).$$

- *Assuming we have $\frac{\mathrm{d}L(X)}{\mathrm{d}Y_R} \in \mathbb{R}^{n \times d}$, then we can calculate $\frac{\mathrm{d}L(X)}{\mathrm{d}Y_R} \frac{\mathrm{d}\mathsf{H}(X_R)}{\mathrm{d}X_R}$ in almost linear time $n^{1+o(1)}$.*

*Then, we can show that,*

- $\frac{\mathrm{d}L(X)}{\mathrm{d}X_R}$ *can be calculated in almost linear time $n^{1+o(1)}$.*

- *If $\frac{\mathrm{d}L(X)}{\mathrm{d}Y_R}$ has $1/\mathrm{poly}(n)$ approximation error, then the approximation error on $\frac{\mathrm{d}L(X)}{\mathrm{d}X_R}$ is still $1/\mathrm{poly}(n)$.*

*Proof.* By the chain rule, we have

$$\frac{\mathrm{d}L(X)}{\mathrm{d}X_R} = \frac{\mathrm{d}L(X)}{\mathrm{d}Y_R} \frac{\mathrm{d}Y_R}{\mathrm{d}X_R}$$

$$= \frac{\mathrm{d}L(X)}{\mathrm{d}Y_R}(I + \frac{\mathrm{d}\mathsf{H}(X_R)}{\mathrm{d}X_R})$$

$$= \frac{\mathrm{d}L(X)}{\mathrm{d}Y_R} + \frac{\mathrm{d}L(X)}{\mathrm{d}Y_R}\frac{\mathrm{d}\mathsf{H}(X_R)}{\mathrm{d}X_R} \tag{32}$$

where the 1st step is from the chain rule, the 2nd step comes from basic calculus, the 3rd step is because of basic algebra.

By the assumption, we already have $\frac{\mathrm{d}L(X)}{\mathrm{d}Y_R}$, and $\frac{\mathrm{d}L(X)}{\mathrm{d}Y_R}\frac{\mathrm{d}\mathsf{H}(X_R)}{\mathrm{d}X_R}$ can be computed in almost linear time $n^{1+o(1)}$.

The addition operation between $\frac{\mathrm{d}L(X)}{\mathrm{d}Y_R}$ and $\frac{\mathrm{d}L(X)}{\mathrm{d}Y_R}\frac{\mathrm{d}\mathsf{H}(X_R)}{\mathrm{d}X_R}$ takes $n \cdot d$ time.

Therefore, the overall running time for $\frac{\mathrm{d}L(X)}{\mathrm{d}X_R}$ is $n^{1+o(1)}$.

Then, we consider the approximation error.

By Eq. (32) and basic linear algebra, the approximation error will not be magnified by more than $(n \cdot d \operatorname{poly}(n) + 1)$. Since $(n \cdot d \operatorname{poly}(n) + 1)(1/\operatorname{poly}(n)) = \operatorname{poly}(n)$, the approximation error on $\frac{\mathrm{d}L(X)}{\mathrm{d}X_R}$ can be bounded by $1/\operatorname{poly}(n)$.

$\square$

### J.3 ANALYSIS FOR THE ENTIRE MODEL WITH THE RESIDUAL CONNECTION

In the previous section, we have shown that, with the addition of the residual connection on a single component, the gradient computation time can still be done in almost linear time. We will apply this finding to the entire model.

We begin by single layer proof.

**Lemma J.4** (Fast gradient computation for single-layer transformer with residual connection)**.** *If we have the below conditions,*

- *Let $L(X)$ be defined as Definition 2.1.*

- *Let $X \in \mathbb{R}^{n \times d}$ be defined as Definition C.3.*

- *Suppose we have a single-layer transformer (see Definition 1.3).*

- *Let the residual connection be defined as Definition J.1 and J.2.*

*Then, we can show that,*

- **Part 1: running time.** *Our algorithm can approximate $\frac{\mathrm{d}L(X)}{\mathrm{d}X}$ in $n^{1+o(1)}$ time.*

- **Part 2: error bound.** *The approximation error of the single-layer transformer with the residual connection can be bounded by $1/\operatorname{poly}(n)$. Namely, our algorithm output $\widetilde{g}_{r_1}$ satisfies*

$$\|\widetilde{g}_{r_1} - \frac{\mathrm{d}L(X)}{\mathrm{d}X}\|_\infty \le 1/\operatorname{poly}(n)$$

*Proof.* We use $T_i$ to represent $T_i(X)$ for simplicity. By the definition of $T_i$ (see also Definition 2.3), we have the following equations

$$T_0 = g_0(X)$$

Follow Definition J.1 and J.2, we have

$$Z_1 = T_0 + \mathsf{Attn}_1(T_0)$$

and

$$T_1 = Z_1 + g_1(Z_1)$$

Then we calculate the gradient by the following steps:

- **Step 1: Calculate** $\frac{\mathrm{d}L(X)}{\mathrm{d}T_1}$. By the definition of $L(X)$ (see also Definition 2.1), we have $\frac{\mathrm{d}L(X)}{\mathrm{d}T_1}$ can be computed in $n \cdot d$ time.

- **Step 2: Calculate** $\frac{\mathrm{d}L(X)}{\mathrm{d}Z_1}$. By Lemma H.2, the assumption in Lemma J.3 is satisfied. Therefore, we have $\frac{\mathrm{d}L(X)}{\mathrm{d}Z_1}$ can be computed in almost linear time $n^{1+o(1)}$.

- **Step 3: Calculate** $\frac{\mathrm{d}L(X)}{\mathrm{d}T_0}$. By Lemma E.11, the assumption in Lemma J.3 is satisfied. Hence, $\frac{\mathrm{d}L(X)}{\mathrm{d}T_0}$ can be computed in almost linear time. By Lemma E.11, the approximation error is $1/\operatorname{poly}(n)$.

- **Step 4: Calculate** $\frac{\mathrm{d}L(X)}{\mathrm{d}X}$. By Lemma H.2, $\frac{\mathrm{d}L(X)}{\mathrm{d}X}$ can be computed in $n^{1+o(1)}$. The approximation error is $(n \cdot d)(1/\operatorname{poly}(n)) = (1/\operatorname{poly}(n))$.

To sum up, we can show that the overall running time for $\frac{\mathrm{d}L(X)}{\mathrm{d}X}$ is $n^{1+o(1)}$ and the approximation error is $1/\operatorname{poly}(n)$.

Let $\widetilde{g}_{r_1}$ be the output of **Step 4**. Then we are done.

$\square$

We now prove for multi-layer.

**Lemma J.5** (Fast gradient computation for multi-layer transformer with residual connection). *If we have the below conditions,*

- *Let $L(X)$ be defined as Definition 2.1.*

- *Let $X \in \mathbb{R}^{n \times d}$ be defined as Definition C.3.*

- *Let the residual connection be defined as Definition J.1 and J.2.*

- *Suppose we have a $m$-layer transformer (see Definition 1.3).*

*Then, we can show that,*

- **Part 1: running time.** *Our algorithm can approximate $\frac{\mathrm{d}L(X)}{\mathrm{d}X}$ in $n^{1+o(1)}$ time.*

- **Part 2: error bound.** *The approximation error of the $m$-layer transformer with the residual connection can be bounded by $1/\operatorname{poly}(n)$. Namely, our algorithm output $\widetilde{g}_r$ satisfies*

$$\|\widetilde{g}_r - \frac{\mathrm{d}L(X)}{\mathrm{d}X}\|_\infty \le 1/\operatorname{poly}(n)$$

*Proof.* We use math induction in this proof.

**Step 1: Proof of a single-layer transformer.**

Firstly, by Lemma J.4, we have the statement holds for a single-layer transformer.

**Step 2: Assumption for $k$-layer transformer.**

Secondly, we assume for any $k$, for $k$-layer transformer model, we have

- **Part 1: running time.** Our algorithm can approximate $\frac{\mathrm{d}L(X)}{\mathrm{d}X}$ in $O(n^{1+o(1)})$ time.

- **Part 2: error bound.** The approximation error of the $k$-layer transformer can be bounded by $1/\operatorname{poly}(n)$. Namely, our algorithm output $\widetilde{g}$ satisfies

$$\|\widetilde{g} - \frac{\mathrm{d}L(X)}{\mathrm{d}X}\|_\infty \le 1/\operatorname{poly}(n)$$

**Step 3: Proof of $(k+1)$-layer transformer.**

Thirdly, we consider the $(k+1)$-layer transformer model.

Let $\mathsf{F}_k$ denote a $k$-layer transformer with the residual connection.

Then, the entire model can be written as

$$(\mathsf{F}_k \circ g_0)(X)$$

By the definition of $T_i$, we have

$$T_0 = g_0(X)$$

Then, by definition of $Z_i$ (see also Definition J.1), we have

$$Z_1 = T_0 + \mathsf{Attn}_1(T_0)$$

By Definition J.2, we have

$$T_1 = Z_1 + g_1(Z_1)$$

Without loss of generality, we assume that the additional transformer layer is added at the beginning of the model. Then, the $(k+1)$-layer transformer model has the following structure:

$$\mathsf{F}_{k+1}(X) = \mathsf{F}_k(T_1)$$

By the assumption for $k$-layer transformer, we have $\frac{\mathrm{d}L(X)}{\mathrm{d}T_1}$ can be computed in almost linear time $n^{1+o(1)}$ and the approximation error can be bounded by $1/\operatorname{poly}(n)$.

We apply similar proof of Lemma J.4, then we can show that, we can compute $\frac{\mathrm{d}L(X)}{\mathrm{d}X}$ in almost linear time $n^{1+o(1)}$ and the approximation error can be bounded by $1/\operatorname{poly}(n)$.

$\square$

## K  MULTI-HEAD ATTENTION

Following the notation used in Section B.1, we use $h$ to denote the number of heads, and $d_h = d/h$ to denote the dimension of each head.

**Definition K.1** (Multi-head attention). *If we have the below conditions,*

- *Let $h$ denote the number of heads.*

- *Let $d$ denote the hidden dimension. Let $d_h = d/h$ denote the dimension of each attention head.*

- *Let $Q, K, V \in \mathbb{R}^{n \times d}$ be defined as Definition C.3.*

- *Let $f(X)$ be defined as Definition C.8.*

- *Let $s(X)$ be defined as Definition C.10.*

*The multi-head attention can be formalized as follows:*

- **Step 1.** *Split the hidden dimension $d$ of $Q, K, V \in \mathbb{R}^{n \times d}$ into $h$ parts. Then, for each $l \in [h]$, we have $Q_l, K_l, V_l \in \mathbb{R}^{n \times d_h}$.*

- **Step 2.** *For each $l \in [h]$, calculate the attention matrix $f_l := \mathsf{Softmax}(Q_l K_l^\top / d_h) \in \mathbb{R}^{n \times n}$, and calculate the corresponding attention result $s_l := f_l V_l \in \mathbb{R}^{n \times d_h}$.*

- **Step 3.** *Concatenate $s_l \in \mathbb{R}^{n \times d_h}$ together, then we have the final multi-head attention output $s \in \mathbb{R}^{n \times d}$.*

Then, we dive into the analysis of the gradient computation process over the attention mechanism with multi-head attention.

**Lemma K.2** (Analysis of the multi-head attention). *If we have the below conditions,*

- *Let* $\mathsf{Attn}(X)$ *be defined as Definition C.3.*

- *Let multi-head attention mechanism be defined as Definition K.1.*

- *Let* $Y_m, X_m \in \mathbb{R}^{n \times d}$ *denote the output and input of the multi-head attention, respectively.*

*Then, we can show that,*

- $\frac{\mathrm{d}L(X)}{\mathrm{d}X_m}$ *can be calculated in almost linear time* $n^{1+o(1)}$.

- *If* $\frac{\mathrm{d}L(X)}{\mathrm{d}Y_m}$ *has* $1/\operatorname{poly}(n)$ *approximation error, then the approximation error on* $\frac{\mathrm{d}L(X)}{\mathrm{d}X_m}$ *is still* $1/\operatorname{poly}(n)$.

*Proof.* Following the notations used in Definition K.1, for $l \in [h]$, we use $s_l \in \mathbb{R}^{n \times d_h}$ to denote the output by each attention head. And we use $s \in \mathbb{R}^{n \times d}$ to denote the concatenated version of the output of the multi-head attention.

By the chain rule and the definition of $L(X)$ (see also Definition 2.1), we have

$$\frac{\mathrm{d}L(X)}{\mathrm{d}X_m} = \frac{\mathrm{d}L(X)}{\mathrm{d}Y_m} \cdot \frac{\mathrm{d}Y_m}{\mathrm{d}s} \frac{\mathrm{d}s}{\mathrm{d}X_m}$$

$$= \frac{\mathrm{d}L(X)}{\mathrm{d}Y_m} \cdot \frac{\mathrm{d}Y_m}{\mathrm{d}s} \sum_{l=1}^{h} \frac{\mathrm{d}s_l}{\mathrm{d}X_m}$$

where the 1st step is from the chain rule, the 2nd step comes from $s \in \mathbb{R}^{n \times d}$ is the concatenated version of $s_l \in \mathbb{R}^{n \times d_h}$.

We calculate the gradient in the following steps:

- **Step 1: Calculate** $\frac{\mathrm{d}L(X)}{\mathrm{d}Y_m}$. By the definition of $L(X)$ (Definition 2.1), we have that $\frac{\mathrm{d}L(X)}{\mathrm{d}Y_m}$ can be calculated in $n \cdot d$ time.

- **Step 2: Calculate** $\frac{\mathrm{d}L(X)}{\mathrm{d}Y_m} \cdot \frac{\mathrm{d}Y_m}{\mathrm{d}s}$. Since we already have $\frac{\mathrm{d}L(X)}{\mathrm{d}Y_m}$, by Lemma H.2, we have $\frac{\mathrm{d}L(X)}{\mathrm{d}Y_m} \cdot \frac{\mathrm{d}Y_m}{\mathrm{d}s}$ can be computed in almost linear time $n^{1+o(1)}$.

- **Step 3: Calculate** $\frac{\mathrm{d}L(X)}{\mathrm{d}Y_m} \cdot \frac{\mathrm{d}Y_m}{\mathrm{d}s} \sum_{l=1}^{h} \frac{\mathrm{d}s_l}{\mathrm{d}X_m}$. For each $l \in [h]$, by Lemma E.11, $\frac{\mathrm{d}L(X)}{\mathrm{d}Y_m} \cdot \frac{\mathrm{d}Y_m}{\mathrm{d}s} \cdot \frac{\mathrm{d}s_l}{\mathrm{d}X_m}$ can be computed in $n^{1+o(1)}$. Since the number of heads $h$ can be viewed as a constant here, it takes $n^{1+o(1)}$ time to compute the gradients on $h$ heads.

Therefore, the overall running time for $\frac{\mathrm{d}L(X)}{\mathrm{d}X_m}$ is $n^{1+o(1)}$.

Then, we consider the error bound.

By assumption, there is $1/\operatorname{poly}(n)$ approximation error on $\frac{\mathrm{d}L(X)}{\mathrm{d}Y_m}$. For each $l \in [h]$, the approximation error will not be magnified by more than $n^2 \cdot d \cdot d_h \cdot \operatorname{poly}(n)$ on $\frac{\mathrm{d}L(X)}{\mathrm{d}Y_m} \cdot \frac{\mathrm{d}Y_m}{\mathrm{d}s} \cdot \frac{\mathrm{d}s_l}{\mathrm{d}X_m}$.

Then, since there is total $h$ heads, the approximation error on $\frac{\mathrm{d}L(X)}{\mathrm{d}X_m}$ can be bound by

$$h \cdot n^2 \cdot d \cdot d_h \cdot \operatorname{poly}(n) \cdot (1/\operatorname{poly}(n)) = 1/\operatorname{poly}(n)$$

$\square$

Similar to the proof of Lemma H.3 and H.4, we apply Lemma K.2 to deal with the multi-head attention in each transformer layer. Then, we can show that $\frac{\mathrm{d}L(X)}{\mathrm{d}X}$ can be computed in almost linear time $n^{1+o(1)}$ and the approximation error can be bounded by $1/\operatorname{poly}(n)$.

## LLM USAGE DISCLOSURE

LLMs were used only to polish language, such as grammar and wording. These models did not contribute to idea creation or writing, and the authors take full responsibility for this paper's content.

