# OpenReview forum: "Training Multi-Layer Transformers in Almost Linear Time"
_ICLR.cc/2026/Conference — ICLR 2026 Conference Withdrawn Submission_

### Official Review · Reviewer_Cryn · 2025-10-26

**Soundness:** 3
**Presentation:** 3
**Contribution:** 3
**Rating:** 4
**Confidence:** 3

**Summary:**

This paper provides a theoretical framework and algorithm for approximating gradient computation in multi-layer transformer architectures with almost linear time complexity, $n^{1+o(1)}$, and polynomially small approximation error $1/poly(n)$. The approach extends earlier results on single-layer acceleration to the backward pass of full multi-layer transformers.

**Strengths:**

- This paper provides important theoritical contributions, extending single-layer to module-wise gradient computation, which allows the gradient of each self-attention layer to be approximated in almost linear time.
- The paper is well-structured, with clear definitions, lemmas, and algorithmic pseudocode.
- This paper potentially can have significant impact, if empirically validated. And it can dramatically reduce the training cost of long-context models.

**Weaknesses:**

The paper’s assumptions constrain real-world applicability.
- The results assume a fixed number of layers m=O(1) so that error propagation stays bounded. Real LLMs contain tens or hundreds of layers, where accumulated approximation errors may become non-negligible.
- The hidden dimension is assumed to grow only as d=O(logn), which simplifies proofs but does not reflect actual transformer configurations.
- The claim of polynomially small error lacks clarifing the constants and polynomial degrees, and the true error magnitude and runtime overhead for real-world is unknown.
- The paper provides no experiments, even on toy setups, to validate gradient accuracy, runtime scaling, or convergence behavior.

**Questions:**

- Although the abstract claims “numerical experiments,” the paper contains no explicit experimental section, quantitative tables, or runtime evaluations?
- What is the per-layer error accumulation?

---

> ### Author Response · Authors · 2025-12-01
>
> Thank you for your thoughtful feedback. Your comments are very helpful and much appreciated. We will address these in the next version.

---

### Official Review · Reviewer_rmB9 · 2025-11-01

**Soundness:** 2
**Presentation:** 1
**Contribution:** 1
**Rating:** 2
**Confidence:** 4

**Summary:**

This paper tries to prove the theoretical possibility of training a transformer model in almost linear time, more specifically, calculating both the forward and backward passes of the attention layer in almost linear time, while having a small error of the order of 1/poly(n), where n represents the context length (or sequence length of the model). To do this, they use results from Alman & Song (2023,2024) as well as reordering the gradient calculations to stay in almost linear time.

Alman, Josh, and Zhao Song. "Fast attention requires bounded entries." Advances in Neural Information Processing Systems 36 (2023): 63117-63135.

Alman, Josh, and Zhao Song. "The fine-grained complexity of gradient computation for training large language models." Advances in Neural Information Processing Systems 37 (2024): 61419-61454.

**Strengths:**

- The paper extends the results from Alman & Song to the full transformer (including multihead, MLP layers, and residual connections) to attempt to show that it is possible to train a transformer in almost linear time.

**Weaknesses:**

- The paper is not very well-written and at times hard to follow or read. Additionally, the paper is difficult to understand without reading a substantial portion of the appendix. All proofs of the paper are relegated to the appendix, and the informal versions of the theorems in the paper state that they show that different parts of the model can be calculated in near-linear time without any intuition of strong reasoning as to why. It is hard to say whether the paper is the main paper or the appendix itself.
- The paper claims multiple times that the results either "transcend" or represent a significant leap forward; however, most of the paper seems to be an application of two papers, which they cite and a reordering of the gradient calculations such that the operations stay near-linear.
- In the abstract, the authors claim that they have done numerical experiments to show that their theory works; however, these are nowhere to be found. In addition, the two papers that reduce the calculation of attention to near linear time (forward and backwards) state that this could be applicable to highly quantised models (4 bits or less), but in non-quantised the U and V matrices would be infeasibly large (the $k_1$, would be in the order of millions, or billions, which would make them slower than quadratic calculations when n = 128k.
- In multiple proofs in the appendix, the authors claim poly(n) / poly(n) <= 1/poly(n) (the poly(n)s are not derived from the same place and are in practice different) because the matrices can be written in log(n) bits; this assumption is very strong and not justified. Without it, the paper does not show that the error does not grow.
- The last two weaknesses can be explained by a perceived misunderstanding of the related works, the central part of getting the attention operation to be near linear time with a small error is that the weights of the weight matrices can be bounded by $o(\sqrt{log(n)})$ this however never shows in this paper (outside of lemma C.13).

**Questions:**

- Were any practical experiments done, as claimed in the abstract?

---

> ### Author Response · Authors · 2025-12-01
>
> Thank you for your thoughtful feedback. Your comments are very helpful and much appreciated. We will address these in the next version.

---

### Official Review · Reviewer_1C18 · 2025-11-01

**Soundness:** 3
**Presentation:** 3
**Contribution:** 3
**Rating:** 4
**Confidence:** 4

**Summary:**

This paper proposes a theoretical framework and algorithm to train multi-layer Transformers in almost linear time reducing the usual quadratic complexity of attention to near-linear. It extends prior work on fast single-layer attention to handle multi-layer backprop, including practical components like multi-head attention, residual connections, and causal masking. The authors provide formal proofs with provable approximation error but offer no experiment validation.

**Strengths:**

- The paper presents a highly ambitious and conceptually original approach to reducing Transformer training complexity. It extends theoretical results from single-layer attention to full multi-layer Transformer architectures, which is a non-trivial and technically deep contribution.
- The theoretical framework is comprehensive and covers practically important architectural components, including multi-head attention, residual connections, and causal masking. This level of completeness makes the theory relevant to real Transformer models rather than being confined to simplified abstractions.
- The proofs and algorithmic structure are clearly presented and mathematically rigorous. The authors successfully derive polynomially bounded errors while maintaining almost-linear time complexity, showing careful control of approximation quality across layers.
- The potential long-term impact of this work is very high. If the approach can be made practical, it could drastically lower the cost of training large models, make long-context training feasible, and influence how future large-scale sequence models are built.

**Weaknesses:**

- The paper lacks any empirical validation or implementation evidence. There are no runtime benchmarks (GPU, TPU test) , or experiments demonstrating that the proposed algorithm actually accelerates Transformer training on real hardware.
- The asymptotic claims hide important constants and assumptions. The requirement that the embedding dimension or number of heads grows only logarithmically with sequence length is unrealistic for large models, and the polynomial factors may dominate at practical scales.
- The algorithm’s reliance on polynomial kernel approximations and low-rank projections raises serious concerns about hardware efficiency. These operations may not map well to GPUs or TPUs, where dense matrix multiplications dominate performance.
- Numerical stability could be problematic. High-degree polynomial expansions are known to suffer from rounding errors under FP16 or BF16 arithmetic, which could make the approach unstable during training.

**Questions:**

1. Have the authors implemented or simulated the proposed algorithm, even at small scale, to verify that it produces correct gradients and leads to measurable runtime improvements over standard attention? Without any empirical results, how can we be confident the asymptotic gains translate into practical speedups?
2. Could the authors make the hidden factors in the $n^{1+o(1)}$ complexity explicit, particularly the dependence on embedding dimension d, number of heads h, and number of layers L? These constants are crucial to assess whether the claimed efficiency holds for real-world Transformer sizes.
3. How do the authors envision mapping the polynomial kernel and low-rank computations onto GPU/TPU hardware, which favors dense matrix multiplications? Are there steps in the algorithm that become sequential or memory-bound, and how would those affect parallelism and wall-clock speed?

---

> ### Author Response · Authors · 2025-12-01
>
> Thank you for your thoughtful feedback. Your comments are very helpful and much appreciated. We will address these in the next version.

---

### Official Review · Reviewer_cVbo · 2025-11-03

**Soundness:** 2
**Presentation:** 1
**Contribution:** 2
**Rating:** 0
**Confidence:** 3

**Summary:**

The authors propose to study gradient descent approximation of self-attention, such that per-step gradient update of transformers can be done in time almost linear in the context length. The authors highlight the importance of low rank approximation in achieving such a speed-up in training.

**Strengths:**

The paper argues that training can approach linear complexity in context length, with minimal error accumulation. However, the work currently lacks empirical support for this claim. In practice, the real-world value of the proposed method depends critically on whether the theoretical savings translate into meaningful wall-clock speedups and whether the approximation error remains negligible for real world training of models. Including controlled experiments that quantify these effects would significantly strengthen the paper.

**Weaknesses:**

This paper’s presentation is extremely weak and not currently suitable for a conference submission. The authors repeatedly defer key ideas to the appendix, referencing terms and components without providing intuition for why they matter or how they fit together. As written, the reader has no clear narrative path to understand the algorithm or proofs, only a sequence of lemmas with no guiding intuition. For a theoretical paper claiming impactful practical consequences, this is a major issue.

There is a clever idea here (low-rank approximation in gradient computation), but it is buried. Even one well-motivated proof sketch in the main text (e.g., for Lemma 4.1) explaining why the decomposition arises and how low-rank structure is exploited would help readers understand what’s going on. At present, the paper reads as if the appendix contains the “real” content and the main text is only a table of contents to it. This is not appropriate for ICLR.

**Concerns About Theoretical Claims**

Lemma 4.5 states that the approximation error scales as $n^m/poly(n)$. $m$ is assumed to be a constant. But current 7B models have $m$ as $32$. This means the error explodes as $n^{32}/poly(n)$. This isn't a small and reasonable error accumulation!


**“We further validate our approach through numerical experiments…”**

However, there are no experiments in the paper. No figures, no tables, no empirical section. This is misleading. If this was a draft produced hastily with LLMs, it should have been cleaned and verified before submission.

Please respect reviewer time. If you promise experiments, they must actually be present.

**Questions:**

I don't have any questions, as the paper has been very poorly presented.

---

> ### Author Response · Authors · 2025-12-01
>
> Thank you for your thoughtful feedback. Your comments are very helpful and much appreciated. We will address these in the next version.

---

### Note · Authors · 2025-12-01

I have read and agree with the venue's withdrawal policy on behalf of myself and my co-authors.